# Accurate and Scalable Estimation of Epistemic Uncertainty for Graph Neural Networks

**Puja Trivedi**[1]*, **Mark Heimann**[2], **Rushil Anirudh**[2,3], **Danai Koutra**[1], **Jayaraman J. Thiagarajan**[2]
[1]University of Michigan, CSE Department, [2]Lawrence Livermore National Laboratory, [3]Amazon

## Abstract

While graph neural networks (GNNs) are widely used for node and graph representation learning tasks, the reliability of GNN uncertainty estimates under distribution shifts remains relatively under-explored. Indeed, while post-hoc calibration strategies can be used to improve in-distribution calibration, they need not also improve calibration under distribution shift. However, techniques which produce GNNs with better intrinsic uncertainty estimates are particularly valuable, as they can always be combined with post-hoc strategies later. Therefore, in this work, we propose G-$\Delta$UQ, a novel training framework designed to improve intrinsic GNN uncertainty estimates. Our framework adapts the principle of stochastic data centering to graph data through novel graph anchoring strategies, and is able to support partially stochastic GNNs. While, the prevalent wisdom is that fully stochastic networks are necessary to obtain reliable estimates, we find that the functional diversity induced by our anchoring strategies when sampling hypotheses renders this unnecessary and allows us to support G-$\Delta$UQ on pretrained models. Indeed, through extensive evaluation under covariate, concept and graph size shifts, we show that G-$\Delta$UQ leads to better calibrated GNNs for node and graph classification. Further, it also improves performance on the uncertainty-based tasks of out-of-distribution detection and generalization gap estimation. Overall, our work provides insights into uncertainty estimation for GNNs, and demonstrates the utility of G-$\Delta$UQ in obtaining reliable estimates.

## 1 Introduction

As graph neural networks (GNNs) are increasingly deployed in critical applications with test-time distribution shifts (Zhang & Chen, 2018; Gaudelet et al., 2020; Yang et al., 2018; Yan et al., 2019; Zhu et al., 2022), it becomes necessary to expand model evaluation to include safety-centric metrics, such as calibration errors (Guo et al., 2017), out-of-distribution (OOD) rejection rates (Hendrycks & Gimpel, 2017), and generalization error predictions (GEP) (Jiang et al., 2019), to holistically understand model performance in such shifted regimes (Hendrycks et al., 2022b; Trivedi et al., 2023b). Notably, improving on these additional metrics often requires reliable uncertainty estimates, such as maximum softmax or predictive entropy, which can be derived from prediction probabilities. Although there is a clear understanding in the computer vision literature that the quality of uncertainty estimates can noticeably deteriorate under distribution shifts (Wiles et al., 2022; Ovadia et al., 2019), the impact of such shifts on graph neural networks (GNNs) remains relatively under-explored.

Post-hoc calibration methods (Guo et al., 2017; Gupta et al., 2021; Kull et al., 2019; Zhang et al., 2020), which use validation datasets to rescale logits to obtain better calibrated models, are an effective, accuracy-preserving strategy for improving uncertainty estimates and model trust-worthiness. Indeed, several post-hoc calibration strategies (Hsu et al., 2022; Wang et al., 2021) have been recently proposed to explicitly account for the non-IID nature of node-classification datasets. However, while these methods are effective at improving uncertainty estimate reliability on in-distribution (ID) data, they have not been evaluated on OOD data, where they may become unreliable. To this end, training strategies which produce models with better intrinsic uncertainty estimates are valuable as they will provide better out-of-the-box ID and OOD estimates, which can then be further combined with post-hoc calibration strategies if desired.

---

*Correspondence to `pujat@umich.edu`.

The $\Delta$-UQ training framework (Thiagarajan et al., 2022) was recently proposed as a scalable, single model alternative for vision models ensembles and has achieved state-of-the-art performance on calibration and OOD detection tasks. Central to $\Delta$-UQ's success is the concept of *anchored* training, where models are trained on stochastic, relative representations of input samples in order to simulate sampling from different functional modes at test time (Sec. 2.) While, on the surface, $\Delta$-UQ also appears as a potentially attractive framework for obtaining reliable, intrinsic uncertainty estimates on graph-based tasks, there are several challenges that arise from the structured, discrete, and variable-sized nature of graph data that must be resolved first. Namely, the anchoring procedure used by $\Delta$-UQ is not applicable for graph datasets, and it is unclear how to design alternative anchoring strategies such that sufficiently diverse functional modes are sampled at inference to provide reliable epistemic uncertainty estimates.

**Proposed Work.** Thus, our work proposes G-$\Delta$UQ, a novel training paradigm which provides better intrinsic uncertainty estimates for both graph and node classification tasks through the use of newly introduced graph-specific, anchoring strategies. Our contributions can be summarized as follows:

• **(Partially) Stochastic Anchoring for GNNs.** We propose G-$\Delta$UQ, a novel training paradigm that improves the reliability of uncertainty estimates on GNN-based tasks. Our novel graph-anchoring strategies support partial stochasticity GNNs as well as training with pretrained models. (Sec. 3).

• **Evaluating Uncertainty-Modulated CIs under Distribution Shifts.** Across covariate, concept and graph-size shifts, we demonstrate that G-$\Delta$UQ leads to better calibration. Moreover, G-$\Delta$UQ's performance is further improved when combined with post-hoc calibration strategies on several node and graph-level tasks, including new safety-critical tasks (Sec. 5).

• **Fine-Grained Analysis of G-$\Delta$UQ.** We study the calibration of architectures of varying expressivity and G-$\Delta$UQ 's ability to improve them under varying distribution shift. We further demonstrate its utility as a lightweight strategy for improving the calibration of pretrained GNNs (Sec. 6).

## 2   RELATED WORK & BACKGROUND

While uncertainty estimates are useful for a variety of safety-critical tasks (Hendrycks & Gimpel, 2017; Jiang et al., 2019; Guo et al., 2017), DNNs are well-known to provide poor uncertainty estimates directly out of the box (Guo et al., 2017). To this end, there has been considerable interest in building calibrated models, where the confidence of a prediction matches the probability of the prediction being correct. Notably, since GEP and OOD detection methods often rely upon transformations of a model's logits, improving calibration can in turn improve performance on these tasks as well. Due to their accuracy-preserving properties, post-hoc calibration strategies, which rescale confidences after training using a validation dataset, are particularly popular. Indeed, several methods (Guo et al., 2017; Gupta et al., 2021; Kull et al., 2019; Zhang et al., 2020) have been proposed for DNNs in general and, more recently, dedicated node-classifier calibration methods (Hsu et al., 2022; Wang et al., 2021) have also been proposed to accommodate the non-IID nature of graph data. (See App. A.9 for more details.) Notably, however, such post-hoc methods do not lead to reliable estimates under distribution shifts, as enforcing calibration on ID validation data does not directly lead to reliable estimates on OOD data (Ovadia et al., 2019; Wiles et al., 2022; Hendrycks et al., 2019).

Alternatively, Bayesian methods have been proposed for DNNs (Hernández-Lobato & Adams, 2015; Blundell et al., 2015), and more recently GNNs (Zhang et al., 2019; Hasanzadeh et al., 2020), as inherently "uncertainty-aware" strategies. However, not only do such methods often lead to performance loss, require complicated architectures and additional training time, they often struggle to outperform the simple Deep Ensembles (DEns) baseline (Lakshminarayanan et al., 2017). By training a collection of independent models, DEns is able to sample different functional modes of the hypothesis space, and thus, capture epistemic variability to perform uncertainty quantification (Wilson & Izmailov, 2020). Given that DEns requires training and storing multiple models, the SoTA $\Delta$-UQ framework (Thiagarajan et al., 2022) was recently proposed to sample different functional modes using only a single model, based on the principle of *anchoring*.

**Background on Anchoring.** Conceptually, anchoring is the process of creating a relative representation for an input sample in terms of a random "anchor." By randomizing anchors throughout training (e.g., stochastically centering samples with respect to different anchors), $\Delta$-UQ emulates the process of sampling and learning different solutions from the hypothesis space.

In detail, let $\mathcal{D}_{train}$ be the training distribution, $\mathcal{D}_{test}$ be the testing distribution, and $\mathcal{D}_{anchor} := \mathcal{D}_{train}$ be the anchoring distribution. Existing research on stochastic centering has focused on vision models (CNNs, ResNets, ViT) and used input space transformations to construct anchored representations. Specifically, given an image sample with corresponding label, $(\mathbf{I}, y)$, and anchor $\mathbf{C} \in \mathcal{D}_{anchor}$, anchored samples were created by subtracting and then channel-wise concatenating two images: $[\mathbf{I} - \mathbf{C} || \mathbf{C}]$[1]. Given the anchored representation, a corresponding stochastically centered model can be defined as $f_\theta : [\mathbf{I} - \mathbf{C} || \mathbf{C}] \to \hat{y}$, and can be trained as shown in Fig. 1. At inference, similar to ensembles, predictions and uncertainties are aggregated over different hypotheses. Namely, given $K$ random anchors, the mean target class prediction, $\boldsymbol{\mu}(y|\mathbf{I})$, and the corresponding variance, $\boldsymbol{\sigma}(y|\mathbf{I})$ are computed as: $\boldsymbol{\mu}(y|\mathbf{I}) = \frac{1}{K} \sum_{k=1}^{K} f_\theta([\mathbf{I} - \mathbf{C}_k, \mathbf{C}_k])$ and $\boldsymbol{\sigma}(y|\mathbf{I}) = \sqrt{\frac{1}{K-1} \sum_{k=1}^{K} (f_\theta([\mathbf{I} - \mathbf{C}_k, \mathbf{C}_k]) - \boldsymbol{\mu})^2}$. Since the variance over $K$ anchors captures epistemic uncertainty by sampling different hypotheses, these estimates can be used to modulate the predictions: $\boldsymbol{\mu}_{\text{calib.}} = \boldsymbol{\mu}(1 - \boldsymbol{\sigma})$. Notably, the rescaled logits and uncertainty estimates have led to state-of-the-art performance on image outlier rejection, calibration, and extrapolation (Anirudh & Thiagarajan, 2022; Netanyahu et al., 2023).

## 3 GRAPH-$\Delta$UQ: UNCERTAINTY-AWARE PREDICTIONS

Given $\Delta$-UQ's success in improving calibration and generalization (Netanyahu et al., 2023) under distribution shifts on computer vision tasks and the limitations of existing post-hoc strategies, stochastic centering appears as a potentially attractive framework for obtaining reliable uncertainty estimates when performing GNN-based classification tasks. However, there are several challenges that must be addressed before to applying it to graph data. Namely, while input space transformations, which induce fully stochastic models, were sufficient for sampling diverse functional hypotheses from vision models, it is (i) non-trivial to define such transformations when working with variable sized, discrete graph data and (ii) unclear whether full stochasticity is in fact needed when working with message passing models. Below, we explore these issues through novel graph anchoring strategies. However, we begin with a conceptual discussion of the role of anchoring strategies in generating reliable uncertainty estimates.

```
#training
for (I,y) in trainloader:
    C = create_anchors(n=BatchSize)
    I_Anc = CONCAT([I-C,C],dim=1)
    preds = model(I_Anc)
    loss = criterion(preds,Y)
#inference
anc_preds = []
testAnc = create_anchors(n=K)
for A in testAncs:
    I_anc = CONCAT([I-A,A],dim=1)
    preds = model(I_anc)
    anc_preds.append(preds)
P = CONCAT(anc_preds,dim=0)
mu = MEAN(P,dim=0)
var = STDDEV(P,dim=0)
```

Figure 1: **Training/Inference with Anchoring.**

**What are the goals of anchoring?:** As discussed in Sec. 2, epistemic uncertainty can be estimated by aggregating the variability over different functional hypotheses (Hüllermeier & Waegeman, 2021). Indeed, the prevalent wisdom behind the success of DeepEns is its ability to sample *diverse* functional hypotheses. Since these hypotheses are more likely to differ on OOD inputs, aggregating them can lead to better generalization and uncertainity estimates. Insofar as stochastic centering seeks to simulate an ensemble through a single model, a key goal of the anchoring distribution/strategy is then to ensure that sampled hypotheses are also diverse. Thiagarajan et al. (2022) obtained sufficient diversity by using input space anchoring to sample a fully stochastic network. However, in the context of Bayesian neural networks (BNNs), it was recently shown that partial stochasticity can perform equally well with respect to fully stochastic BNNs at significantly less cost (Sharma et al., 2023). This suggests that in addition to the "amount" of diversity, the "effective" or functional diversity is also important for performance. However, in practice, it is difficult to control this balance, so existing methods default to heuristics that only promote diverse hypotheses. For example, DeepEns uses different random seeds or shuffles the batch order when creating ensemble members, and $\Delta$-UQ relies upon fully stochastic models. To this end, we propose three different anchoring strategies that only handle the difficulties of working with graph data and GNNs, but also induce different scales of the aforementioned balance. At a high-level, our strategies trade-off the amount of stochasticity (i.e., amount of diversity) and the semantic expressivity of the anchoring distribution to accomplish this.

**Notations.** Let $\mathcal{G} = (\mathbf{X}^0, \mathbf{A}, Y)$ be a graph with node features $\mathbf{X}^0 \in \mathbb{R}^{N \times d}$, adjacency matrix $\mathbf{A} \in [0, 1]^{N \times N}$ and labels $Y$, where $N, d, q$ denote the number of nodes, feature dimension and number of classes, respectively. When performing graph classification, $Y \in \{0, 1\}^q$; for node classification, let $Y \in \{0, 1\}^{N \times q}$. We define a graph classification GNN consisting of $\ell$ message

---

[1]For example, channel wise concatenating two RGB images creates a 6 channel sample.

passing layers (MPNN), a graph-level readout function (READOUT), and classifier head (MLP) as follows: $\mathbf{X}^{\ell+1} = \text{MPNN}^{\ell+1}\left(\mathbf{X}^\ell, \mathbf{A}\right)$, $\mathbf{G} = \text{READOUT}\left(\mathbf{X}^{\ell+1}\right)$, and $\hat{Y} = \text{MLP}\left(\mathbf{G}\right)$ where $\mathbf{X}^{\ell+1} \in \mathbb{R}^{N \times d_\ell}$ is the intermediate node representation at layer $\ell+1$, $\mathbf{G} \in \mathbb{R}^{1 \times d_{\ell+1}}$ is the graph representation, and $\hat{Y} \in \{0,1\}^q$ is the predicted label. When performing node classification, we do not include the READOUT layer, and instead output node-level predictions: $\hat{Y} = \text{MLP}\left(\mathbf{X}^{\ell+1}\right)$. We use subscript $_i$ to indicate indexing and $||$ to indicate concatenation.

## 3.1 Node Feature Anchoring

We begin by introducing a graph anchoring strategy for inducing fully stochastic GNNs. Due to size variability and discreteness, performing a structural residual operation by subtracting two adjacency matrices would be ineffective at inducing an anchored GNN. Indeed, such a transform would introduce artificial edge weights and connectivity artifacts. Likewise, when performing graph classification, we cannot directly anchor over node features, since graphs are different sizes. Taking arbitrary subsets of node features is also inadvisable as node features cannot be considered IID. Further, due to iterative message passing, the network may not be able to converge after aggregating $l$ hops of stochastic node representations (see A.15 for details). Furthermore, there is a risk of exploding stochasticity when anchoring MPNNs. Namely, after $l$ rounds of message passing, a node's representations will have aggregated information from its $l$ hop neighborhood. However, since anchors are unique to individual nodes, these representations are not only stochastic due to their own anchors but also those of their neighbors.

To address both these challenges, we instead fit a $d$-dimensional Gaussian distribution over the training dataset's input node features which is then used as the anchoring distribution (see Fig. 2). While a simple solution, the fitted distribution allows us to easily sample anchors for arbitrarily sized graphs, and helps manage stochasticity by reducing the complexity of the anchoring distribution, ensuring that overall stochasticity is manageable, even after aggregating the $l$-hop neighbhorhood. (See A.15 for details.) We emphasize that this distribution is only used for anchoring and does not assume that the dataset's node features are normally distributed. During training, we randomly sample a unique anchor for each node. Mathematically, given anchors $\mathbf{C}^{N \times d} \sim \mathcal{N}(\mu, \sigma)$, we create the anchored node features as: $[\mathbf{X}^0 - \mathbf{C}||\mathbf{X}^0]$. During inference, we sample a fixed set of $K$ anchors and compute residuals for all nodes with respect to the *same* anchor after performing appropriate broadcasting, e.g., $\mathbf{c}^{1 \times d} \sim \mathcal{N}(\mu, \sigma)$, where $\mathbf{C} := \text{REPEAT}(\mathbf{c}, N)$ and $[\mathbf{X}^0 - \mathbf{C}_k||\mathbf{X}^0]$ is the $k$th anchored sample. For datasets with categorical node features, anchoring can be performed after embedding the node features into a continuous space. If node features are not available, anchoring can still be performed via positional encodings (Wang et al., 2022b), which are known to improve the expressivity and performance of GNNs (Dwivedi et al., 2022a). Lastly, note that performing node feature anchoring (NFA) is the most analogous extension of $\Delta$-UQ to graphs as it results in fully stochastic GNNs. This is particularly true on node classification tasks where each node can be viewed as an individual sample, similar to a image sample original $\Delta$-UQ formulation.

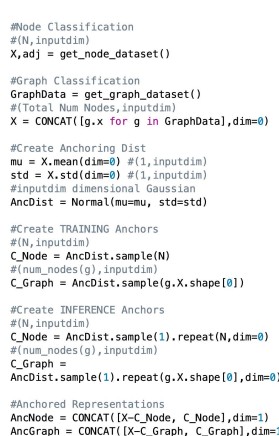

```
#Node Classification
#(N,inputdim)
X,adj = get_node_dataset()

#Graph Classification
GraphData = get_graph_dataset()
#(Total Num Nodes,inputdim)
X = CONCAT([g.x for g in GraphData],dim=0)

#Create Anchoring Dist
mu = X.mean(dim=0) #(1,inputdim)
std = X.std(dim=0) #(1,inputdim)
#inputdim dimensional Gaussian
AncDist = Normal(mu=mu, std=std)

#Create TRAINING Anchors
#(N,inputdim)
C_Node = AncDist.sample(N)
#(num_nodes(g),inputdim)
C_Graph = AncDist.sample(g.X.shape[0])

#Create INFERENCE Anchors
#(N,inputdim)
C_Node = AncDist.sample(1).repeat(N,dim=0)
#(num_nodes(g),inputdim)
C_Graph =
AncDist.sample(1).repeat(g.X.shape[0],dim=0)

#Anchored Representations
AncNode = CONCAT([X-C_Node, C_Node],dim=1)
AncGraph = CONCAT([X-C_Graph, C_Graph],dim=1)
```

Figure 2: **Node Feature Anchoring Pseudocode.**

## 3.2 Hidden Layer Anchoring for Graph Classification

While NFA can conceptually be used for graph classification tasks, there are several nuances that may limit its effectiveness. Notably, since each sample (and label) is at a graph-level, NFA not only effectively induces multiple anchors per sample, it also ignores structural information that may be useful in sampling more *functionally diverse* hypotheses, e.g., hypotheses which capture functional modes that rely upon different high-level semantic, non-linear features. To improve the quality of hypothesis sampling, we introduce hidden layer anchoring below, which incorporates structural information into anchors at the expense of full stochasticity in the network (See Fig. 1):

*Hidden Layer and Readout Anchoring:* Given a GNN containing $\ell$ MPNN layers, let $2 \leq r \leq \ell$ be the layer at which we perform anchoring. Then, given the intermediate node representations $\mathbf{X}^{r-1} = \text{MPNN}^{r-1}(\mathbf{X}^{r-2}, \mathbf{A})$, we randomly shuffle the node features over the entire batch, $(\mathbf{C} =$

SHUFFLE($\mathbf{X}^{r-1}$, dim $= 0$)), concatenate the residuals ($[\mathbf{X}^{r-1} - C||C]$), and proceed with the READOUT and MLP layers as usual. (See A.1 for corresponding pseudocode.) Note the gradients of the query sample are not considered when updating parameters, and MPNN$^r$ is modified to accept inputs of dimension $d_r \times 2$ (to take in anchored representations as inputs). At inference, we subtract a single anchor from all node representations using broadcasting. Hidden layer anchoring induces the following GNN: $\mathbf{X}^{r-1} = \text{MPNN}^{r-1}(\mathbf{X}^{r-2}, \mathbf{A})$, $\mathbf{X}^r = \text{MPNN}^r\left([\mathbf{X}^{r-1} - \mathbf{C}||\mathbf{C}], \mathbf{A}\right)$, and $\mathbf{X}^{\ell+1} = \text{MPNN}^{r+1...\ell}\left(\mathbf{X}^r, \mathbf{A}\right)$, and $\hat{Y} = \text{MLP}(\text{READOUT}\left(\mathbf{X}^{\ell+1}\right))$ .

Not only do hidden layer anchors aggregate structural information over $r$ hops, they induce a GNN that is now partially stochastic, as layers $1 \ldots r$ are deterministic. Indeed, by reducing network stochasticity, it is naturally expected that hidden layer anchoring will reduce the diversity of the hypotheses, but by sampling more *functionally diverse* hypotheses through deeper, semantically expressive anchors, it is possible that *naively* maximizing diversity is in fact not required for reliable uncertainty estimation. To validate this hypothesis, we thus propose the final variant, READOUT anchoring for graph classification tasks. While conceptually similar to hidden layer anchoring, here, we simultaneously minimize GNN stochasticity (only the classifier is stochastic) and maximize anchor expressivity (anchors are graph representations pooled after $\ell$ rounds of message passing). Notably, READOUT anchoring is also compatible with pretrained GNN backbones, as the final MLP layer of a pretrained model is discarded (if necessary), and reinitialized to accommodate query/anchor pairs. Given the frozen MPNN backbone, only the anchored classifier head is trained.

In Sec. 5, we empirically verify the effectiveness of our proposed G-ΔUQ variants and demonstrate that fully stochastic GNNs are, in fact, unnecessary to obtain highly generalizable solutions, meaningful uncertainties and improved calibration on graph classification tasks.

## 4 NODE CLASSIFICATION EXPERIMENTS: G-ΔUQ IMPROVES CALIBRATION

In this section, we demonstrate that G-ΔUQ improves uncertainty estimation in GNNs, particularly when evaluating *node classifiers* under distribution shifts. To the best of our knowledge, GNN calibration has not been extensively evaluated under this challenging setting, where uncertainty estimates are known to be unreliable (Ovadia et al., 2019). We demonstrate that G-ΔUQ not only directly provides better estimates, but also that combining G-ΔUQ with existing post-hoc calibration methods further improves performance.

**Experimental Setup.** We use the concept and covariate shifts for WebKB, Cora and CBAS datasets provided by Gui et al. (2022), and follow the recommended hyperparameters for training. In our implementation of node feature anchoring, we use 10 random anchors to obtain predictions with G-ΔUQ. All our results are averaged over 5 seeds and post-hoc calibration methods (described further in App. A.9) are fitted on the in-distribution validation dataset. The expected calibration error and accuracy on the unobserved "OOD test" split are reported.

**Results.** From Table 1 (and expanded in Table. 12), we observe that across 4 datasets and 2 shifts that G-ΔUQ, *without* any post-hoc calibration (✗), is superior to the vanilla model on nearly every benchmark for better or same accuracy (8/8 benchmarks) and better calibration error (7/8), often with a significant gain in calibration performance. Moreover, we note that combining G-ΔUQ with a particular posthoc calibration method improves performance relative to using the same posthoc method with a vanilla model. Indeed, on WebKB, across 9 posthoc strategies, "G-ΔUQ +<calibration method>" improves or maintains the calibration performance of the corresponding "no G-ΔUQ +<calibration method>" in 7/9 (concept) and 6/9 (covariate) cases. (See App. A.8 for more discussion.) Overall, across post hoc methods and evaluation sets, G-ΔUQ variants are very performant achieving (best accuracy: 8/8), best calibration (6/8) or second best calibration (2/8).

## 5 GRAPH CLASSIFICATION UNCERTAINTY EXPERIMENTS WITH G-ΔUQ

While applying G-ΔUQ to node classification tasks was relatively straightforward, performing stochastic centering with graph classification tasks is more nuanced. As discussed in Sec. 3, different anchoring strategies can introduce varying levels of stochasticity, and it is unknown how these strategies affect uncertainty estimate reliability. Therefore, we begin by demonstrating that fully stochastic GNNs are not necessary for producing reliable estimates (Sec. 5.1). We then extensively evaluate the calibration of partially stochastic GNNs on covariate and concept shifts with and without post-hoc calibration strategies (Sec. 5.2), as well as for different UQ tasks (Sec. 5.3).

Figure 3: **Effect of Anchoring Layer.** Anchoring at different layers (L1, L2, L3) induces different hypotheses spaces. Variations of stochastic anchoring outperform models without it, and the lightweight `READOUT` anchoring in particular generally performs well across datasets and architectures.

Lastly, we demonstrate that G-$\Delta$UQ's uncertainty estimates remain reliable when used with different architectures and pretrained backbones (Sec. 6).

## 5.1 IS FULL STOCHASTICITY NECESSARY FOR G-$\Delta$UQ?

By changing the anchoring strategy and intermediate anchoring layer, we can induce varying levels of stochasticity in the resulting GNNs. As discussed in Sec. 3, we hypothesize that the decreased stochasticity incurred by performing anchoring at deeper network layers will lead to more functionally diverse hypotheses, and consequently more reliable uncertainty estimates. We verify this hypothesis here, by studying the effect of anchoring layer on calibration under graph-size distribution shift. Namely, we find that `READOUT` anchoring sufficiently balances stochasticity and functional diversity.

**Experimental Setup.** We study the effect of different anchoring strategies on graph classification calibration under graph-size shift. Following the procedure of (Buffelli et al., 2022; Yehudai et al., 2021), we create a size distribution shift by taking the smallest 50%-quantile of graph size for the training set, and evaluate on the largest 10% quantile. Following (Buffelli et al., 2022), we apply this splitting procedure to NCI1, NCI09, and PROTEINS (Morris et al., 2020), consider 3 GNN backbones (GCN (Kipf & Welling, 2017), GIN (Xu et al., 2019), and PNA (Corso et al., 2020)) and use the same architectures/parameters. (See Appendix A.6 for dataset statistics.) The accuracy and expected calibration error over 10 seeds on the largest-graph test set are reported for models trained with and without stochastic anchoring.

**Results.** We compare the performance of anchoring at different layers in Fig. 3. While there is no clear winner across datasets and architectures for which *layer* to perform anchoring, we find there is consistent trend across all datasets and architectures the best accuracy and ECE is obtained by a G-$\Delta$UQ variant. Overall, our results clearly indicate that partial stochasticity can yield substantial benefits when estimating uncertainty (though suboptimal layers selections are generally not too harmful). Insofar, as we are the first to focus on partially stochastic anchored GNNs, automatically selecting the anchoring layer is an interesting direction of future work. However, in subsequent experiments, we use `READOUT` anchoring, unless otherwise noted, as it is faster to train (see App. A.13), and allow our methods to support pretrained models. Indeed, `READOUT` anchoring (L3) yields top performance for some datasets and architectures such as PNA on PROTEINS, compared to earlier (L1, L2) and, as we discuss below, is very performative on a variety of tasks and shifts.

## 5.2 CALIBRATION UNDER CONCEPT AND COVARIATE SHIFTS

Next, we assess the ability of G-$\Delta$UQ to produce well-calibrated models under covariate and concept shift in graph classification tasks. We find that G-$\Delta$UQ not only provides better calibration out of the box, its performance is further improved when combined with post-hoc calibration techniques.

**Experimental Setup.** We use three different datasets (GOODCMNIST, GOODMotif-basis, GOODSST2) with their corresponding splits and shifts from the recently proposed Graph Out-Of Distribution (GOOD) benchmark (Gui et al., 2022). The architectures and hyperparameters suggested by the benchmark are used for training. G-$\Delta$UQ uses `READOUT` anchoring and 10 random anchors (see App. A.7 for more details). We report accuracy and expected calibration error for the OOD test dataset, taken over three seeds.

**Results.** As shown in Table 1, we observe that G-$\Delta$UQ leads to inherently better calibrated models, as the ECE from G-$\Delta$UQ without additional post-hoc calibration (✗) is better than the vanilla ("No G-$\Delta$UQ") counterparts on 5/6 datasets. Moreover, we find that combining G-$\Delta$UQ with a

Table 1: **Calibration under Covariate and Concept shifts.** G-ΔUQ leads to better calibrated models for node-(GOODCora) and graph-level prediction tasks under different kinds of distribution shifts. Notably, G-ΔUQ can be combined with post-hoc calibration techniques to further improve calibration. The expected calibration error (ECE) is reported. Best, Second.

| | | | Shift: Concept | | | | Shift: Covariate | | | |
| | | | Accuracy (↑) | | ECE (↓) | | Accuracy (↑) | | ECE (↓) | |
| Dataset | Domain | Calibration | No G-ΔUQ | G-ΔUQ | No G-ΔUQ | G-ΔUQ | No G-ΔUQ | G-ΔUQ | No G-ΔUQ | G-ΔUQ |
|---|---|---|---|---|---|---|---|---|---|---|
| GOODCora | Degree | × | 0.581±0.003 | 0.595±0.003 | 0.307±0.009 | 0.13±0.011 | 0.47±0.002 | 0.518±0.014 | 0.348±0.032 | 0.141±0.008 |
| | | CAGCN | 0.581±0.003 | 0.597±0.002 | 0.135±0.009 | 0.128±0.025 | 0.47±0.002 | 0.522±0.025 | 0.256±0.08 | 0.231±0.025 |
| | | Dirichlet | 0.534±0.007 | 0.551±0.004 | 0.12±0.004 | 0.196±0.003 | 0.414±0.007 | 0.449±0.01 | 0.163±0.002 | 0.356±0.01 |
| | | ETS | 0.581±0.003 | 0.596±0.004 | 0.301±0.009 | 0.116±0.018 | 0.47±0.002 | 0.523±0.003 | 0.31±0.077 | 0.141±0.003 |
| | | GATS | 0.581±0.003 | 0.596±0.004 | 0.185±0.018 | 0.229±0.039 | 0.47±0.002 | 0.521±0.011 | 0.211±0.004 | 0.308±0.011 |
| | | IRM | 0.582±0.002 | 0.597±0.002 | 0.125±0.001 | 0.102±0.002 | 0.469±0.001 | 0.522±0.004 | 0.194±0.005 | 0.13±0.004 |
| | | Orderinvariant | 0.581±0.003 | 0.592±0.002 | 0.226±0.024 | 0.213±0.049 | 0.47±0.002 | 0.498±0.027 | 0.318±0.042 | 0.196±0.027 |
| | | Spline | 0.571±0.003 | 0.595±0.003 | 0.080±0.004 | 0.068±0.004 | 0.459±0.003 | 0.52±0.004 | 0.158±0.01 | 0.098±0.004 |
| | | VS | 0.581±0.003 | 0.596±0.004 | 0.306±0.004 | 0.127±0.002 | 0.47±0.001 | 0.522±0.005 | 0.345±0.005 | 0.146±0.005 |
| GOODCMNIST | Color | × | 0.499±0.003 | 0.497±0.002 | 0.439±0.078 | 0.334±0.066 | 0.348±0.009 | 0.355±0.034 | 0.551±0.147 | 0.423±0.172 |
| | | Dirichlet | 0.495±0.009 | 0.510±0.008 | 0.303±0.012 | 0.304±0.007 | 0.350±0.053 | 0.335±0.059 | 0.542±0.091 | 0.406±0.076 |
| | | ETS | 0.499±0.011 | 0.500±0.013 | 0.433±0.014 | 0.359±0.013 | 0.348±0.037 | 0.336±0.067 | 0.538±0.077 | 0.467±0.088 |
| | | IRM | 0.499±0.006 | 0.500±0.010 | 0.285±0.004 | 0.283±0.008 | 0.348±0.049 | 0.336±0.071 | 0.416±0.084 | 0.425±0.093 |
| | | Orderinvariant | 0.499±0.030 | 0.500±0.028 | 0.379±0.050 | 0.386±0.042 | 0.348±0.036 | 0.337±0.059 | 0.475±0.077 | 0.542±0.104 |
| | | Spline | 0.495±0.008 | 0.497±0.010 | 0.29±0.007 | 0.291±0.008 | 0.346±0.051 | 0.335±0.071 | 0.414±0.085 | 0.425±0.093 |
| | | VS | 0.499±0.007 | 0.500±0.012 | 0.439±0.006 | 0.377±0.009 | 0.349±0.037 | 0.336±0.067 | 0.549±0.071 | 0.468±0.089 |
| | | Ensembling | 0.505±0.001 | 0.509±0.004 | 0.437±0.082 | 0.343±0.004 | 0.397±0.005 | 0.408±0.006 | 0.423±0.017 | 0.327±0.013 |
| GOODMotif | Basis | × | 0.925±0.001 | 0.925±0.003 | 0.095±0.014 | 0.078±0.007 | 0.691±0.001 | 0.689±0.002 | 0.329±0.274 | 0.342±0.266 |
| | | Dirichlet | 0.925±0.011 | 0.923±0.010 | 0.081±0.015 | 0.103±0.007 | 0.686±0.009 | 0.681±0.009 | 0.337±0.067 | 0.316±0.047 |
| | | ETS | 0.925±0.009 | 0.927±0.012 | 0.095±0.010 | 0.096±0.013 | 0.691±0.011 | 0.699±0.016 | 0.314±0.041 | 0.304±0.049 |
| | | IRM | 0.925±0.014 | 0.93±0.013 | 0.087±0.018 | 0.097±0.010 | 0.691±0.011 | 0.698±0.016 | 0.316±0.051 | 0.305±0.045 |
| | | Orderinvariant | 0.925±0.010 | 0.928±0.011 | 0.091±0.009 | 0.093±0.007 | 0.691±0.011 | 0.690±0.011 | 0.321±0.050 | 0.319±0.041 |
| | | Spline | 0.925±0.010 | 0.927±0.011 | 0.091±0.008 | 0.089±0.012 | 0.691±0.010 | 0.689±0.016 | 0.324±0.055 | 0.313±0.051 |
| | | VS | 0.925±0.009 | 0.927±0.012 | 0.095±0.010 | 0.095±0.013 | 0.683±0.013 | 0.680±0.018 | 0.326±0.057 | 0.311±0.059 |
| | | Ensembling | 0.932±0.002 | 0.943±0.006 | 0.086±0.016 | 0.047±0.003 | 0.714±0.012 | 0.699±0.009 | 0.298±0.383 | 0.321±0.196 |
| GOODSST2 | Length | × | 0.694±0.002 | 0.693±0.001 | 0.288±0.017 | 0.277±0.011 | 0.826±0.002 | 0.828±0.004 | 0.159±0.027 | 0.154±0.039 |
| | | Dirichlet | 0.686±0.02 | 0.683±0.001 | 0.15±0.021 | 0.138±0.015 | 0.793±0.005 | 0.8±0.012 | 0.15±0.02 | 0.131±0.007 |
| | | ETS | 0.685±0.02 | 0.683±0.001 | 0.21±0.009 | 0.211±0.003 | 0.794±0.005 | 0.8±0.011 | 0.287±0.007 | 0.296±0.014 |
| | | IRM | 0.685±0.019 | 0.682±0.002 | 0.239±0.002 | 0.231±0.006 | 0.796±0.006 | 0.801±0.011 | 0.26±0.005 | 0.265±0.011 |
| | | Orderinvariant | 0.685±0.02 | 0.683±0.001 | 0.225±0.002 | 0.222±0.003 | 0.794±0.005 | 0.8±0.011 | 0.226±0.003 | 0.224±0.007 |
| | | Spline | 0.684±0.02 | 0.683±0.002 | 0.233±0.005 | 0.23±0.005 | 0.79±0.004 | 0.794±0.016 | 0.259±0.005 | 0.263±0.012 |
| | | VS | 0.685±0.019 | 0.683±0 | 0.334±0.044 | 0.374±0.002 | 0.787±0.008 | 0.8±0.013 | 0.307±0.116 | 0.32±0.011 |
| | | Ensembling | 0.705±0.002 | 0.709±0.004 | 0.276±0.038 | 0.248±0.022 | 0.838±0.001 | 0.842±0.006 | 0.154±0.032 | 0.132±0.019 |

particular post-hoc calibration methods further elevates its performance relative to combining the same strategy with vanilla models. Indeed, for a fixed post-hoc calibration strategy, G-ΔUQ improves the calibration, while maintaining comparable if not better accuracy on the vast majority of the methods and datasets. There are some settings where combining G-ΔUQ or the vanilla model with a post-hoc method leads decreases performance (for example, GOODSST2, covariate, ETS, calibration) but we emphasize that this is not a short-coming of G-ΔUQ. Posthoc strategies, which rely upon ID calibration datasets, may not be effective on shifted data. This further emphasizes the importance of our OOD evaluation and G-ΔUQ as an intrinsic method for improving uncertainty estimation.

## 5.3 USING CONFIDENCE ESTIMATES IN SAFETY-CRITICAL TASKS

While post-hoc calibration strategies rely upon an additional calibration dataset to provide meaningful uncertainty estimates, such calibration datasets are not always available and may not necessarily improve OOD performance (Ovadia et al., 2019). Thus, we also evaluate the quality of the uncertainty estimates directly provided by G-ΔUQ on two additional UQ-based, safety-critical tasks (Hendrycks et al., 2022b; 2021; Trivedi et al., 2023b): (i) OOD detection (Hendrycks et al., 2019), which attempts to classify samples as in- or out-of-distribution, and (ii) generalization error prediction (GEP) (Jiang et al., 2019), which attempts to predict the generalization on unlabeled test datasets (to the best of our knowledge, we are the first to study GEP of graph classifiers). In the interest of space, we present the results on GEP in the appendix.

**OOD Detection Experimental Setup**. By reliably detecting OOD samples and abstaining from making predictions on them, models can avoid over-extrapolating to irrelevant distributions. While many scores have been proposed for detection (Hendrycks et al., 2019; 2022a; Lee et al., 2018; Wang et al., 2022a; Liu et al., 2020), popular scores, such as maximum softmax probability and predictive entropy (Hendrycks & Gimpel, 2017), are derived from uncertainty estimates. Here, we report the

Table 2: **GOOD-Datasets, OOD Detection Performance.** The AUROC of the binary classification task of classifying OOD samples is reported. G-ΔUQ variants outperform the vanilla models on 6/8 datasets. [We further note that end-to-end G-ΔUQ does in fact lose performance relative to the vanilla model on 4 datasets. Investigating why pretrained G-ΔUQ is able to increase performance on those datasets is an interesting direction of future work. It does not appear that a particular shift is more difficult for this task: concept shift is easier for GOODCMNIST and GOODMotif(Basis) while covariate shift is easier for GOODMotif(Size) and GOODSST2. Combining G-ΔUQ with more sophisticated, uncertainty or confidence based OOD scores may further improve performance.]

| Method | CMNIST (Color) Concept(↑) | Covariate(↑) | MotifLPE (Basis) Concept(↑) | Covariate(↑) | MotifLPE (Size) Concept(↑) | Covariate(↑) | SST2 Concept(↑) | Covariate(↑) |
|---|---|---|---|---|---|---|---|---|
| Vanilla | $0.759 \pm 0.006$ | $0.468 \pm 0.092$ | $0.736 \pm 0.021$ | $\mathbf{0.466 \pm 0.001}$ | $0.680 \pm 0.003$ | $0.755 \pm 0.074$ | $\mathbf{0.350 \pm 0.014}$ | $0.345 \pm 0.066$ |
| G-ΔUQ | $0.771 \pm 0.002$ | $0.470 \pm 0.043$ | $0.758 \pm 0.006$ | $0.328 \pm 0.022$ | $0.677 \pm 0.005$ | $0.691 \pm 0.067$ | $0.338 \pm 0.023$ | $0.351 \pm 0.042$ |
| Pretr. G-ΔUQ | $\mathbf{0.774 \pm 0.016}$ | $\mathbf{0.543 \pm 0.152}$ | $\mathbf{0.769 \pm 0.029}$ | $0.272 \pm 0.025$ | $\mathbf{0.686 \pm 0.004}$ | $\mathbf{0.829 \pm 0.113}$ | $0.324 \pm 0.055$ | $\mathbf{0.446 \pm 0.049}$ |

Table 3: **RotMNIST-Calibration.** Here, we report expanded results (calibration) on the Rotated MNIST dataset, including a variant that combines G-ΔUQ with Deep Ens. Notably, we see that anchored ensembles outperform basic ensembles in both accuracy and calibration.

| Architecture | LPE? | G-ΔUQ | Calibration | Avg.ECE (↓) | ECE (10) (↓) | ECE (15) (↓) | ECE (25) (↓) | ECE (35) (↓) | ECE (40) (↓) |
|---|---|---|---|---|---|---|---|---|---|
| GatedGCN | ✗ | ✗ | ✗ | $0.038 \pm 0.001$ | $0.059 \pm 0.001$ | $0.068 \pm 0.340$ | $0.126 \pm 0.008$ | $0.195 \pm 0.012$ | $0.245 \pm 0.011$ |
| | ✗ | ✅ | ✗ | $\mathbf{0.018} \pm 0.008$ | $\underline{0.029} \pm 0.013$ | $\mathbf{0.033} \pm 0.164$ | $\underline{0.069} \pm 0.033$ | $\underline{0.117} \pm 0.048$ | $\underline{0.162} \pm 0.067$ |
| | ✗ | ✗ | Ensembling | $0.026 \pm 0.000$ | $0.038 \pm 0.001$ | $0.042 \pm 0.001$ | $0.084 \pm 0.002$ | $0.135 \pm 0.001$ | $0.185 \pm 0.003$ |
| | ✗ | ✅ | Ensembling | $\mathbf{0.014} \pm 0.003$ | $0.018 \pm 0.005$ | $0.021 \pm 0.005$ | $\underline{0.036} \pm 0.012$ | $\underline{0.069} \pm 0.032$ | $\underline{0.114} \pm 0.056$ |
| GatedGCN | ✅ | ✗ | ✗ | $0.036 \pm 0.003$ | $0.059 \pm 0.002$ | $0.068 \pm 0.340$ | $0.125 \pm 0.006$ | $0.191 \pm 0.007$ | $0.240 \pm 0.008$ |
| | ✅ | ✅ | ✗ | $0.022 \pm 0.007$ | $\mathbf{0.028} \pm 0.014$ | $\underline{0.034} \pm 0.169$ | $\mathbf{0.062} \pm 0.022$ | $\mathbf{0.109} \pm 0.019$ | $\mathbf{0.141} \pm 0.019$ |
| | ✅ | ✗ | Ensembling | $0.024 \pm 0.001$ | $0.038 \pm 0.001$ | $0.043 \pm 0.002$ | $0.083 \pm 0.001$ | $0.139 \pm 0.004$ | $0.181 \pm 0.002$ |
| | ✅ | ✅ | Ensembling | $0.017 \pm 0.002$ | $0.024 \pm 0.005$ | $\underline{0.027} \pm 0.008$ | $\mathbf{0.030} \pm 0.004$ | $\mathbf{0.036} \pm 0.012$ | $\mathbf{0.059} \pm 0.033$ |
| GPS | ✅ | ✗ | ✗ | $0.026 \pm 0.001$ | $0.044 \pm 0.001$ | $0.052 \pm 0.156$ | $0.108 \pm 0.006$ | $0.197 \pm 0.012$ | $0.273 \pm 0.008$ |
| | ✅ | ✅ | ✗ | $0.022 \pm 0.001$ | $0.037 \pm 0.005$ | $0.044 \pm 0.133$ | $0.091 \pm 0.008$ | $0.165 \pm 0.018$ | $0.239 \pm 0.018$ |
| | ✅ | ✗ | Ensembling | $\underline{0.016} \pm 0.001$ | $0.026 \pm 0.002$ | $0.030 \pm 0.000$ | $0.066 \pm 0.000$ | $0.123 \pm 0.000$ | $0.195 \pm 0.000$ |
| | ✅ | ✅ | Ensembling | $\mathbf{0.014} \pm 0.000$ | $\underline{0.023} \pm 0.002$ | $\underline{0.027} \pm 0.003$ | $0.055 \pm 0.004$ | $0.103 \pm 0.006$ | $0.164 \pm 0.006$ |

AUROC for the binary classification task of detecting OOD samples using the maximum softmax probability as the score (Kirchheim et al., 2022).

**OOD Detection Results.** As shown in Table 2, we observe that G-ΔUQ variants improve OOD detection performance over the vanilla baseline on 6/8 datasets, where pretrained G-ΔUQ obtains the best overall performance on 6/8 datasets. G-ΔUQ performs comparably on GOODSST2(concept shift), but does lose some performance on GOODMotif(Covariate). We note that vanilla models provided by the original benchmark generalized poorly on this particular dataset (increased training time/accuracy did not improve performance), and this behavior was reflected in our experiments. We suspect that poor generalization coupled with stochasticity may explain G-ΔUQ's performance here.

# 6 FINE GRAINED ANALYSIS OF G-ΔUQ

Given that the previous sections extensively verified the effectiveness of G-ΔUQ on a variety of covariate and concept shifts across several tasks, we seek a more fine-grained understanding of G-ΔUQ's behavior with respect to different architectures and training strategies. In particular, we demonstrate that G-ΔUQ continues to improve calibration with expressive graph transformer architectures, and that using READOUT anchoring with pretrained GNNs is an effective lightweight strategy for improving calibration of frozen GNN models.

## 6.1 CALIBRATION UNDER CONTROLLED SHIFTS

Recently, it was shown that modern, non-convolutional architectures (Minderer et al., 2021) are not only more performant but also more calibrated than older, convolutional architectures (Guo et al., 2017) under vision distribution shifts. Here, we study an analogous question: are more expressive GNN architectures better calibrated under distribution shift, and how does G-ΔUQ impact their calibration? Surprisingly, we find that more expressive architectures are not considerably better calibrated than their MPNN counterparts, and ensembles of MPNNs outperform ensembles of GTrans. Notably, G-ΔUQ continues to improve calibration with respect to these architectures as well.

**Experimental Setup.** *(1) Models.* While improving the expressivity of GNNs is an active area of research, positional encodings (PEs) and graph-transformer (GTran) architectures (Müller et al., 2023) are popular strategies due to their effectiveness and flexibility. GTrans not only help mitigate over-smoothing and over-squashing (Alon & Yahav, 2021; Topping et al., 2022) but they also better capture long-range dependencies (Dwivedi et al., 2022b).

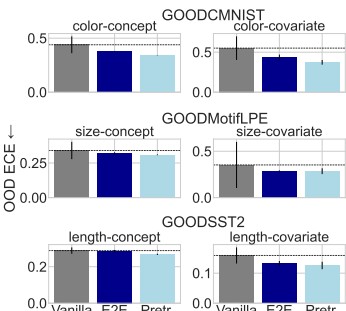

Meanwhile, graph PEs help improve expressivity by differentiating isomorphic nodes, and capturing structural vs. proximity information (Dwivedi et al., 2022a). Here, we ask if these enhancements translate to improved calibration under distribution shift by comparing architectures with/without PEs and transformer vs. MPNN models. We use equivariant and stable PEs (Wang et al., 2022b), the state-of-the-art, "general, powerful, scalable" (GPS) framework with a GatedGCN backbone for the GTran, GatedGCN for the vanilla MPNN, and perform `READOUT` anchoring with 10 random anchors. *(2) Data.* In order to understand calibration behavior as distribution shifts become progressively more severe, we create structurally distorted but valid graphs by rotating MNIST images by a fixed number of degrees (Ding et al., 2021) and then creating the corresponding super-pixel graphs (Dwivedi et al., 2020; Knyazev et al., 2019; Velickovic et al., 2018). (See Appendix, Fig. 6.) Since superpixel segmentation on these rotated images will yield different superpixel $k$-nn graphs but leave class information unharmed, we can emulate different severities of label-preserving structural distortion shifts. We note that models are trained only using the original (0° rotation) graphs. Accuracy (see appendix) and ECE over 3 seeds are reported for the rotated graphs.

Figure 4: Out-of-distribution calibration error from applying G-$\Delta$UQ in end-to-end training vs. to a pretrained model, which is a simple yet effective way to use stochastic anchoring.

**Results.** In Table 3, we present the OOD calibration results, with results of more variants and metrics in the supplementary Table 5 and 8. First, we observe that PEs have minimal effects on both calibration and accuracy by comparing GatedGCN with and without LPEs. This suggests that while PEs may enhance expressivity, they do not directly induce better calibration. Next, we find that while vanilla GPS is better calibrated when the distribution shift is not severe (10, 15, 25 degrees), it is less calibrated (but more performant) than GatedGCN at more severe distribution shifts (35, 40 degrees). This is in contrast to known findings about vision transformers. Lastly, we see that G-$\Delta$UQ continues to improve calibration across all considered architectural variants, with minimal accuracy loss. *Surprisingly, however, we observe that ensembles of G-$\Delta$UQ models not only effectively resolve any performance drops, they also cause MPNNs to be better calibrated than their GTran counterparts.*

## 6.2 How does G-$\Delta$UQ perform with pretrained models?

As large-scale pretrained models become the norm, it is beneficial to be able to perform lightweight training that leads to safer models. Thus, we investigate if `READOUT` anchoring is such a viable strategy when working with pretrained GNN backbones, as it only requires training a stochastically centered classifier on top of a frozen backbone. (Below, we discuss results on GOODDataset, but please see A.4 for results on RotMNIST and A.12 for additional discussion.)

**Results.** From Fig. 4 (and expanded in Fig. 8), we observe that across datasets, pretraining (PT) yields competitive (often superior) OOD calibration with respect to end-to-end (E2E) G-$\Delta$UQ. With the exception of GOODMotif (basis) dataset, PT G-$\Delta$UQ improves the OOD ECE over both vanilla and E2E G-$\Delta$UQ models at comparable or improved OOD accuracy (6/8 datasets). Furthermore, PT G-$\Delta$UQ also improves the ID ECE on all but the GOODMotif(size) (6/8), where it performs comparably to the vanilla model, and maintains the ID accuracy. Notably, as only an anchored classifier is trained, PT G-$\Delta$UQ substantially reduces training time relative to E2E G-$\Delta$UQ and vanilla models (see App. A.13), highlighting its strengths as a light-weight, effective strategy for improving uncertainty estimation.

## 7 Conclusion

We propose G-$\Delta$UQ, a novel training approach that adapts stochastic data centering for GNNs through newly introduced graph-specific anchoring strategies. Our extensive experiments demonstrate G-$\Delta$UQ improves calibration and uncertainty estimates of GNNs under distribution shifts.

## 8 ACKNOWLEDGEMENTS

This work was performed under the auspices of the U.S. Department of Energy by the Lawrence Livermore National Laboratory under Contract No. DE-AC52-07NA27344, Lawrence Livermore National Security, LLC. and is partially supported by the LLNL-LDRD Program under Project No. 22-SI-004 with IM release number LLNL-CONF-850978. This work is also partially supported by the National Science Foundation under CAREER Grant No. IIS 1845491, Army Young Investigator Award No. W9-11NF1810397, and Adobe, Amazon, Facebook, and Google faculty awards. Any opinions, findings, and conclusions or recommendations expressed here are those of the author(s) and do not reflect the views of funding parties. RA completed this work while at Lawrence Livermore National Security, LLC. PT thanks Ekdeep Singh Lubana and Vivek Sivaraman Narayanaswamy for useful discussions during the course of this project.

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

# A   APPENDIX

- **PseudoCode** (Sec. A.1)
- **Ethics** (Sec. A.2)
- **Reproducibility** (Sec. A.3)
- **Details and Expanded Results for Super-pixel Graph Experiments**(Sec. A.4)
- **Stochastic Centering on the Empirical NTK of Graph Neural Networks** (Sec. A.5)
- **Size-Generalization Dataset Statistics** (Sec. A.6)
- **GOOD Dataset Statistics and Expanded Results** (Sec. A.7)
- **Alternative Anchoring Strategies on GOOD Datasets** (Sec. A.8)
- **Discussion of Post-hoc Calibration Strategies** (Sec. A.9)
- **Details of Generalization Gap Experiments** (Sec. A.10)
- **Expanded Pretraining Results** (Sec. A.12)
- **Runtimes** (Sec. A.13)
- **Mean and Variance of Node Feature Anchoring Distributions** (Sec. A.14)
- **Discussion on Anchoring Design Choices** (Sec. A.15)

## A.1 PSEUDOCODE FOR G-ΔUQ

```python
class GNN:
    def __init__():
        self.make_layers()
        self.MPNN1 = MPNN(InputDim,HDim1)
        self.MPNN2 = MPNN(HDim1,HDim2)
        self.MPNN3 = MPNN(HDim2, HDim3)
        self.READOUT = READOUTLayer()
        self.classifier = Linear(HDim3, NumClass)

    def forward(self, X,Adj):
        h = self.MPNN1(X,Adj) # Out: (B,N,HDim1)
        h = self.MPNN2(h,Adj) # Out: (B, N,HDim2)
        h = self.MPNN3(h,Adj) # Out: (B,N,HDim3)
        g = self.READOUT(h,Batch) # Out: (B, HDim3)
        pred = self.classifier(g) # Out: (B, NumClass)
        return pred
```

(a) Vanilla GNN

```python
class GNN_NodeFeatureAnc:
    def __init__(GNN):
        self.MPNN1 = MPNN(InputDim * 2,HDim1)
        self.MPNN2 = MPNN(HDim1,HDim2)
        self.MPNN3 = MPNN(HDim2, HDim3)
        self.READOUT = READOUTLayer()
        self.classifier = Linear(HDim3, NumClass)
        self.make_anchor_dist(AggNodeFeats)

    def make_anchor_dist(NodeFeats):
        mu = MEAN(AggNodeFeats,dim=0)
        std = STD(AggNodeFeats,dim=0)
        #d-dim Gaussian
        self.AnchorDist = Normal(mean=mu, std=std)

    def forward(X,Adj):
        # (N,InputDim)
        C = self.AnchorDist.sample(X.shape[0])
        # (N,InputDim x 2)
        AncRep = CONCAT([X - C, C], dim=1)
        h = self.MPNN1(AncRep,Adj) #Out: (N,HDim1)
        h = self.MPNN2(h,Adj)
        h = self.MPNN3(h,Adj)
        g = self.READOUT(h,Batch)
        pred = self.classifier(g)
        return pred
```

(b) G-ΔUQ with Node Feature Anchoring

```python
class GNN_NodeFeatureAnc:
    def __init__(GNN):
        self.MPNN1 = MPNN(InputDim * 2,HDim1)
        self.MPNN2 = MPNN(HDim1,HDim2)
        self.MPNN3 = MPNN(HDim2, HDim3)
        self.READOUT = READOUTLayer()
        self.classifier = Linear(HDim3, NumClass)
        self.make_anchor_dist(AggNodeFeats)

    def make_anchor_dist(NodeFeats):
        mu = MEAN(AggNodeFeats,dim=0)
        std = STD(AggNodeFeats,dim=0)
        #d-dim Gaussian
        self.AnchorDist = Normal(mean=mu, std=std)

    def forward(X,Adj):
        # (N,InputDim)
        C = self.AnchorDist.sample(X.shape[0])
        # (N,InputDim x 2)
        AncRep = CONCAT([X - C, C], dim=1)
        h = self.MPNN1(AncRep,Adj) #Out: (N,HDim1)
        h = self.MPNN2(h,Adj)
        h = self.MPNN3(h,Adj)
        g = self.READOUT(h,Batch)
        pred = self.classifier(g)
        return pred
```

(c) G-ΔUQ with Hidden Rep Anchoring

```python
class GNN_ReadOutAnc:
    def __init__(GNN):
        self.MPNN1 = MPNN(InputDim,HDim1)
        self.MPNN2 = MPNN(HDim1, HDim2)
        self.MPNN3 = MPNN(HDim2, HDim3)
        self.READOUT = READOUTLayer()
        #Corres. to GDUQLayer!
        self.classifier = Linear(HDim3 * 2, NumClass)

    def forward(X,Adj):
        h = self.MPNN1(X,Adj)
        h = self.MPNN2(h,Adj)
        h = self.MPNN3(h,Adj)
        g = self.READOUT(h,Batch) #OUT: (B, HDim3)
        C = SHUFFLE(g, dim=0) #(B, HDim3)
        gAnc = CONCAT([g-C,C],dim=1)#(B, HDim3 x 2)
        pred = self.classifier(gAnc) #Out: (B, NumClass)
        return pred
```

(d) G-ΔUQ with READOUT Anchoring

Figure 5: **PseudoCode for G-ΔUQ**. We provide simplified pseudo-code to demonstrate how anchoring can be performed. We assume PyTorchGeometric style mini-batching. Changes with respect to the vanilla GNN are shown in bold. Unchanged lines are grayed out.

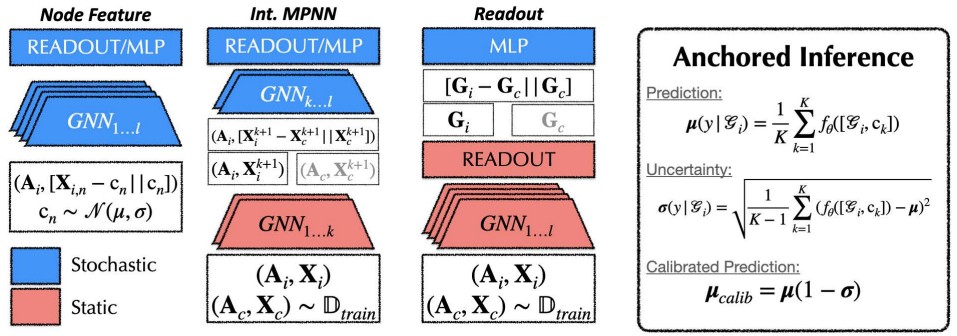

**Overview of G-ΔUQ models.** Here, we present a conceptual overview of how G-ΔUQ induces partially stochastic models. This figure is complementary to 5.

## A.2 ETHICS STATEMENT

This work proposes a method to improve uncertainty estimation in graph neural networks, which has potential broader societal impacts. As graph learning models are increasingly deployed in real-world applications like healthcare, finance, and transportation, it becomes crucial to ensure these models make reliable predictions and know when they may be wrong. Unreliable models can lead to harmful outcomes if deployed carelessly. By improving uncertainty quantification, our work contributes towards trustworthy graph AI systems.

We also consider several additional safety-critical tasks, including generalization gap prediction for graph classification (to the best of our knowledge, we are the first to report results on this task) and OOD detection. We hope our work will encourage further study in these important areas.

However, there are some limitations. Our method requires (modest) additional computation during training and inference, which increases resource usage. Although G-$\Delta$UQ, unlike post-hoc methods, does not need to be fit on a validation dataset, evaluation of its benefits also also relies on having some out-of-distribution or shifted data available, which may not always be feasible. We have seen in Table 1 that there are tasks for which G-$\Delta$UQfails to improve accuracy and/or calibration of some post-hoc methods, further emphasizing the need to perform appropriate model selection and the risks if shifted validation data is not available. Finally, there are open questions around how much enhancement in uncertainty calibration translates to real-world safety and performance gains.

Looking ahead, we believe improving uncertainty estimates is an important direction for graph neural networks and deep learning more broadly. This will enable the development safe, reliable AI that benefits society. We hope our work inspires more research in the graph domain that focuses on uncertainty quantification and techniques that provide guarantees about model behavior, especially for safety-critical applications. Continued progress will require interdisciplinary collaboration between graph machine learning researchers and domain experts in areas where models are deployed.

## A.3 REPRODUCIBILITY

For reproducing our experiments, we have made our code available at this anonymous repository. In the remainder of this appendix (specifically App. A.6, A.7), and A.10), we also provide additional details about the benchmarks and experimental setup.

A.4   DETAILS ON SUPER-PIXEL EXPERIMENTS

We provide an example of the rotated images and corresponding super-pixel graphs in Fig. 6. (Note that classes "6" and "9" may be confused under severe distribution shift, i.e. 90 degrees rotation or more. Hence, to avoid harming class information, our experiments only consider distribution shift from rotation up to 40 degrees.)

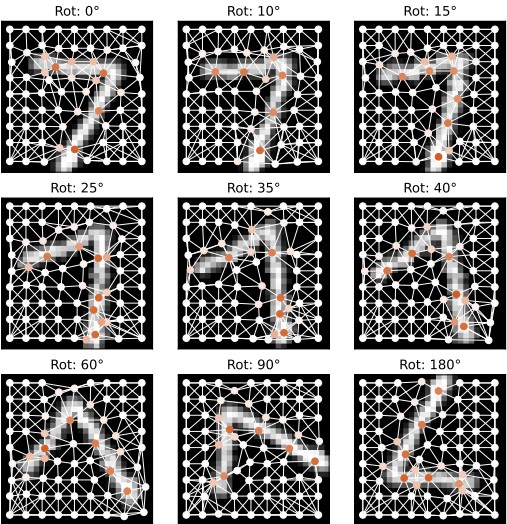

Figure 6: **Rotated Super-pixel MNIST.** Rotating images prior to creating super-pixels to leads to some structural distortion (Ding et al., 2021). However, we can see that the class-discriminative information is preserved, despite rotation. This allows for simulating different levels of graph structure distribution shifts, while still ensuring that samples are valid.

Tables 4 and 5 provided expanded results on the rotated image super-pixel graph classification task, discussed in Sec. 6.1.

In Table 7 we focus on the calibration results on this task for GPS variants alone. Across all levels of distribution shift, the best method is our strategy for applying G-ΔUQ  to a pretrained model–demonstrating that this is not just a practical choice when it is infeasible to retrain a model, but can lead to powerful performance by any measure. Second-best on all datasets is applying G-ΔUQ  during training, further highlighting the benefits of stochastic anchoring.

In addition to the structural distribution shifts we get by rotating the images before constructing super-pixel graphs, we also simulate feature distribution shifts by adding Gaussian noise with different standard deviations to the pixel value node features in the super-pixel graphs. In Table 8, we report accuracy and calibration results for varying levels of distribution shift (represented by the size of the standard deviation of the Gaussian noise). Across different levels of feature distribution shift, we also see that G-ΔUQ  results in superior calibration, while maintaining competitive or in many cases superior accuracy.

Table 4: **RotMNIST-Accuracy.** Here, we report expanded results (accuracy) on the Rotated MNIST dataset, including a variant that combines G-ΔUQ with Deep Ens. Notably, we see that anchored ensembles outperform basic ensembles in both accuracy and calibration. (Best results for models using Deep Ens. and those not using it marked separately.)

| MODEL | G-ΔUQ? | LPE? | Avg. Test (↑) | Acc. (10) (↑) | Acc. (15) (↑) | Acc. (25) (↑) | Acc. (35) (↑) | Acc. (40) (↑) |
|---|---|---|---|---|---|---|---|---|
| GatedGCN | ✗ | ✗ | 0.947 ±0.002 | 0.918 ±0.002 | 0.904 ±0.005 | 0.828 ±0.009 | 0.738 ±0.009 | 0.679 ±0.007 |
|  | ✅ | ✗ | 0.933 ±0.015 | 0.894 ±0.019 | 0.878 ±0.020 | 0.794 ±0.032 | 0.698 ±0.036 | 0.636 ±0.048 |
|  | ✗ | ✅ | 0.949 ±0.002 | 0.917 ±0.004 | 0.904 ±0.005 | 0.829 ±0.007 | 0.744 ±0.007 | 0.685 ±0.006 |
|  | ✅ | ✅ | 0.915 ±0.032 | 0.872 ±0.038 | 0.852 ±0.0414 | 0.776 ±0.039 | 0.680 ±0.037 | 0.631 ±0.033 |
| GPS | ✗ | ✅ | **0.970** ±0.001 | **0.948** ±0.001 | **0.938** ±0.001 | **0.873** ±0.006 | **0.770** ±0.013 | **0.688** ±0.009 |
|  | ✅ | ✅ | 0.969 ±0.001 | 0.946 ±0.003 | 0.937 ±0.003 | 0.869 ±0.003 | 0.769 ±0.012 | 0.679 ±0.014 |
| GPS (Pretrained) | ✅ | ✅ | 0.967 ±0.002 | 0.945 ±0.004 | 0.934 ±0.005 | 0.864 ±0.009 | 0.759 ±0.010 | 0.674 ±0.002 |
| GatedGCN-DENS | ✗ | ✗ | 0.963 ±0.0002 | 0.943 ±0.001 | 0.933 ±0.001 | 0.874 ±0.002 | 0.794 ±0.002 | 0.731 ±0.002 |
|  | ✅ | ✗ | 0.949 ±0.008 | 0.922 ±0.008 | 0.907 ±0.011 | 0.828 ±0.020 | 0.733 ±0.032 | 0.662 ±0.046 |
|  | ✗ | ✅ | 0.965 ±0.001 | 0.943 ±0.001 | 0.933 ±0.001 | 0.873 ±0.001 | 0.792 ±0.004 | 0.736 ±0.003 |
|  | ✅ | ✅ | 0.954 ±0.005 | 0.930 ±0.010 | 0.917 ±0.011 | 0.850 ±0.023 | 0.759 ±0.025 | 0.696 ±0.032 |
| GPS-DENS | ✗ | ✅ | **0.980** ±0.000 | **0.969** ±0.000 | **0.961** ±0.000 | **0.913** ±0.000 | **0.834** ±0.000 | **0.750** ±0.000 |
|  | ✅ | ✅ | 0.978 ±0.001 | 0.963 ±0.000 | 0.953 ±0.001 | 0.905 ±0.000 | 0.822 ±0.002 | 0.736 ±0.003 |

Table 5: **RotMNIST-Calibration.** Here, we report expanded results (calibration) on the Rotated MNIST dataset, including a variant that combines G-ΔUQ with Deep Ens. Notably, we see that anchored ensembles outperform basic ensembles in both accuracy and calibration. (Best results for models using Deep Ens. and those not using it marked separately.)

| MODEL | G-ΔUQ | LPE? | Avg.ECE (↓) | ECE (10) (↓) | ECE (15) (↓) | ECE (25) (↓) | ECE (35) (↓) | ECE (40) (↓) |
|---|---|---|---|---|---|---|---|---|
| GatedGCN-TS | ✗ | ✗ | 0.035 ±0.001 | 0.054 ±0.002 | 0.062 ±0.003 | 0.118 ±0.007 | 0.185 ±0.006 | 0.233 ±0.008 |
|  | ✗ | ✅ | 0.033 ±0.002 | 0.053 ±0.002 | 0.061 ±0.004 | 0.116 ±0.005 | 0.179 ±0.006 | 0.225 ±0.005 |
| GatedGCN | ✗ | ✗ | 0.038 ±0.001 | 0.059 ±0.001 | 0.068 ±0.340 | 0.126 ±0.008 | 0.195 ±0.012 | 0.245 ±0.011 |
|  | ✅ | ✗ | **0.018** ±0.008 | 0.029 ±0.013 | **0.033** ±0.164 | 0.069 ±0.033 | 0.117 ±0.048 | 0.162 ±0.067 |
|  | ✗ | ✅ | 0.036 ±0.003 | 0.059 ±0.002 | 0.068 ±0.340 | 0.125 ±0.006 | 0.191 ±0.007 | 0.240 ±0.008 |
|  | ✅ | ✅ | 0.022 ±0.007 | **0.028** ±0.014 | 0.034 ±0.169 | **0.062** ±0.022 | **0.109** ±0.019 | **0.141** ±0.019 |
| GPS-TS | ✗ | ✅ | 0.024 ±0.001 | 0.041 ±0.001 | 0.049 ±0.001 | 0.102 ±0.006 | 0.188 ±0.012 | 0.261 ±0.008 |
| GPS | ✗ | ✅ | 0.026 ±0.001 | 0.044 ±0.001 | 0.052 ±0.156 | 0.108 ±0.006 | 0.197 ±0.012 | 0.273 ±0.008 |
|  | ✅ | ✅ | 0.022 ±0.001 | 0.037 ±0.005 | 0.044 ±0.133 | 0.091 ±0.008 | 0.165 ±0.018 | 0.239 ±0.018 |
| GPS (Pretrained) | ✅ | ✅ | 0.021 ±0.001 | 0.032 ±0.003 | 0.039 ±0.116 | 0.083 ±0.002 | 0.153 ±0.007 | 0.217 ±0.012 |
| GatedGCN-DENS | ✗ | ✗ | 0.026 ±0.000 | 0.038 ±0.001 | 0.042 ±0.001 | 0.084 ±0.002 | 0.135 ±0.001 | 0.185 ±0.003 |
|  | ✅ | ✗ | **0.014** ±0.003 | **0.018** ±0.005 | **0.021** ±0.005 | 0.036 ±0.012 | 0.069 ±0.032 | 0.114 ±0.056 |
|  | ✗ | ✅ | 0.024 ±0.001 | 0.038 ±0.001 | 0.043 ±0.002 | 0.083 ±0.001 | 0.139 ±0.004 | 0.181 ±0.002 |
|  | ✅ | ✅ | 0.017 ±0.002 | 0.024 ±0.005 | 0.027 ±0.008 | **0.030** ±0.004 | **0.036** ±0.012 | **0.059** ±0.033 |
| GPS-DENS | ✗ | ✅ | 0.016 ±0.001 | 0.026 ±0.002 | 0.030 ±0.000 | 0.066 ±0.000 | 0.123 ±0.000 | 0.195 ±0.000 |
|  | ✅ | ✅ | **0.014** ±0.000 | 0.023 ±0.002 | 0.027 ±0.003 | 0.055 ±0.004 | 0.103 ±0.006 | 0.164 ±0.006 |

Table 6: **Accuracy of GPS Variants on RotatedMNIST.** We focus on the accuracy results for GPS variants on rotated MNIST dataset. Using G-ΔUQ (with or without pretraining) remains close in accuracy to foregoing it, generally within the range of the standard deviation of the results.

| MODEL | G-ΔUQ? | Avg. Test (↑) | Acc. (10) (↑) | Acc. (15) (↑) | Acc. (25) (↑) | Acc. (35) (↑) | Acc. (40) (↑) |
|---|---|---|---|---|---|---|---|
| GPS | ✗ | **0.970** ±0.001 | **0.948** ±0.001 | **0.938** ±0.001 | **0.873** ±0.006 | **0.770** ±0.013 | **0.688** ±0.009 |
|  | ✅ | 0.969 ±0.001 | 0.946 ±0.003 | 0.937 ±0.003 | 0.869 ±0.003 | 0.769 ±0.012 | 0.679 ±0.014 |
| GPS (Pretrained) | ✅ | 0.967 ±0.002 | 0.945 ±0.004 | 0.934 ±0.005 | 0.864 ±0.009 | 0.759 ±0.010 | 0.674 ±0.002 |

Table 7: **Calibration of GPS Variants on RotatedMNIST.** We focus on the calibration results for GPS variants on rotated MNIST dataset. Across the board, we see improvements from using G-ΔUQ , with our strategy of applying it to a pretrained model doing best.

| MODEL | G-ΔUQ | Avg.ECE (↓) | ECE (10) (↓) | ECE (15) (↓) | ECE (25) (↓) | ECE (35) (↓) | ECE (40) (↓) |
|---|---|---|---|---|---|---|---|
| GPS-TS | ✗ | 0.024 ±0.001 | 0.041 ±0.001 | 0.049 ±0.001 | 0.102 ±0.006 | 0.188 ±0.012 | 0.261 ±0.008 |
| GPS | ✗ | 0.026 ±0.001 | 0.044 ±0.001 | 0.052 ±0.156 | 0.108 ±0.006 | 0.197 ±0.012 | 0.273 ±0.008 |
|  | ✅ | 0.022 ±0.001 | 0.037 ±0.005 | 0.044 ±0.133 | 0.091 ±0.008 | 0.165 ±0.018 | 0.239 ±0.018 |
| GPS (Pretrained) | ✅ | **0.021** ±0.001 | **0.032** ±0.003 | **0.039** ±0.116 | **0.083** ±0.002 | **0.153** ±0.007 | **0.217** ±0.012 |

Table 8: **MNIST Feature Shifts**. G-$\Delta$UQ improves calibration and maintains competitive or even improved accuracy across varying levels of feature distribution shift.

| MODEL | LPE? | G-$\Delta$UQ? | Calibration | STD = 0.1 | | STD = 0.2 | | STD = 0.3 | | STD = 0.4 | |
|---|---|---|---|---|---|---|---|---|---|---|---|
| | | | | Accuracy ($\uparrow$) | ECE ($\downarrow$) | Accuracy ($\uparrow$) | ECE ($\downarrow$) | Accuracy ($\uparrow$) | ECE ($\downarrow$) | Accuracy ($\uparrow$) | ECE ($\downarrow$) |
| GatedGCN | ✗ | ✗ | ✗ | 0.742±0.005 | 0.186±0.018 | 0.481±0.015 | 0.414±0.092 | 0.293±0.074 | 0.606±0.147 | 0.197±0.092 | 0.71±0.178 |
| | ✗ | ✓ | ✗ | **0.773**±0.053 | **0.075**±0.032 | 0.536±0.010 | **0.160**±0.087 | **0.356**±0.101 | 0.422±0.083 | **0.249**±0.074 | **0.529**±0.047 |
| | ✓ | ✗ | ✗ | 0.751±0.02 | 0.176±0.014 | 0.519±0.004 | 0.348±0.03 | 0.345±0.032 | 0.485±0.096 | 0.233±0.043 | 0.581±0.142 |
| | ✓ | ✓ | ✗ | 0.745±0.026 | 0.100±0.036 | **0.541**±0.040 | 0.235±0.067 | 0.355±0.062 | **0.408**±0.116 | 0.242±0.063 | 0.539±0.139 |

## A.5 STOCHASTIC CENTERING ON THE EMPIRICAL NTK OF GRAPH NEURAL NETWORKS

Using a simple grid-graph dataset and 4 layer GIN model, we compute the Fourier spectrum of the NTK. As shown in Fig. 7, we find that shifts to the node features can induce systematic changes to the spectrum.

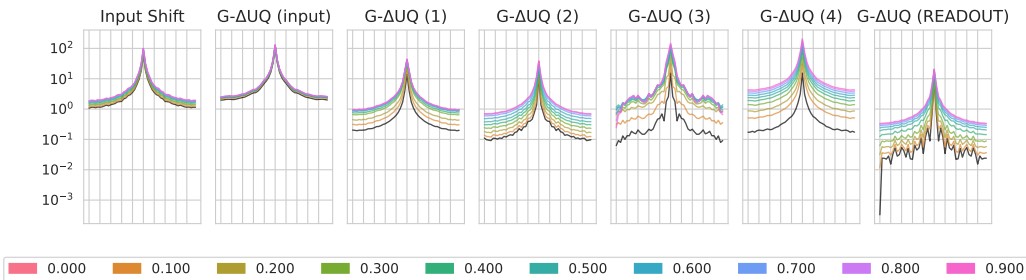

Figure 7: **Stochastic Centering with the empirical GNN NTK.** We find that performing constant shifts at intermediate layers introduces changes to a GNN's NTK. We include a vanilla GNN NTK in black for reference. Further, note the shape of the spectrum should not be compared across subplots as each subplot was created with a different random initialization.

### A.6   SIZE-GENERALIZATION DATASET STATISTICS

The statistics for the size generalization experiments (see Sec. 5.1) are provided below in Table 9.

Table 9: **Size Generalization Dataset Statistics:** This table is directly reproduced from (Buffelli et al., 2022), who in turn used statistics from (Yehudai et al., 2021; Bevilacqua et al., 2021).

| | NCI1 | | | NCI109 | | |
|---|---|---|---|---|---|---|
| | ALL | SMALLEST 50% | LARGEST 10% | ALL | SMALLEST 50% | LARGEST 10% |
| CLASS A | 49.95% | 62.30% | 19.17% | 49.62% | 62.04% | 21.37% |
| CLASS B | 50.04% | 37.69% | 80.82% | 50.37% | 37.95% | 78.62% |
| # OF GRAPHS | 4110 | 2157 | 412 | 4127 | 2079 | 421 |
| AVG GRAPH SIZE | 29 | 20 | 61 | 29 | 20 | 61 |

| | PROTEINS | | | DD | | |
|---|---|---|---|---|---|---|
| | ALL | SMALLEST 50% | LARGEST 10% | ALL | SMALLEST 50% | LARGEST 10% |
| CLASS A | 59.56% | 41.97% | 90.17% | 58.65% | 35.47% | 79.66% |
| CLASS B | 40.43% | 58.02% | 9.82% | 41.34% | 64.52% | 20.33% |
| # OF GRAPHS | 1113 | 567 | 112 | 1178 | 592 | 118 |
| AVG GRAPH SIZE | 39 | 15 | 138 | 284 | 144 | 746 |

### A.7   GOOD BENCHMARK EXPERIMENTAL DETAILS

For our experiments in Sec. 5.2, we utilize the in/out-of-distribution covariate and concept splits provided by Gui et al. (2022). Furthermore, we use the suggested models and architectures provided by their package. In brief, we use GIN models with virtual nodes (except for GOODMotif) for training, and average scores over 3 seeds. When performing stochastic anchoring at a particular layer, we double the hidden representation size for that layer. Subsequent layers retain the original size of the vanilla model.

When performing stochastic anchoring, we use 10 fixed anchors randomly drawn from the in-distribution validation dataset. We also train the G-ΔUQ for an additional 50 epochs to ensure that models are able to converge. Please see our code repository for the full details.

We also include results on additional node classification benchmarks featuring distribution shift in Table 12. In Table 13, we compare models without G-ΔUQ to the use of G-ΔUQ with randomly sampled anchors at the first or second layer.

| Dataset | Shift | Train | ID validation | ID test | OOD validation | OOD test | Train | OOD validation | ID validation | ID test | OOD test |
|---|---|---|---|---|---|---|---|---|---|---|---|
| | | | | Length | | | | | | | |
| GOOD-SST2 | covariate | 24744 | 5301 | 5301 | 17206 | 17490 | | | | | |
| | concept | 27270 | 5843 | 5843 | 15142 | 15944 | | | | | |
| | | | | Color | | | | | | | |
| GOOD-CMNIST | covariate | 42000 | 7000 | 7000 | 7000 | 7000 | | | | | |
| | concept | 29400 | 6300 | 6300 | 14000 | 14000 | | | | | |
| | no shift | 42000 | 14000 | 14000 | - | - | | | | | |
| | | | | Base | | | | | Size | | |
| GOOD-Motif | covariate | 18000 | 3000 | 3000 | 3000 | 3000 | 18000 | 3000 | 3000 | 3000 | 3000 |
| | concept | 12600 | 2700 | 2700 | 6000 | 6000 | 12600 | 2700 | 2700 | 6000 | 6000 |
| | | | | Word | | | | | Degree | | |
| GOOD-Cora | covariate | 9378 | 1979 | 1979 | 3003 | 3454 | 8213 | 1979 | 1979 | 3841 | 3781 |
| | concept | 7273 | 1558 | 1558 | 3807 | 5597 | 7281 | 1560 | 1560 | 3706 | 5686 |
| | | | | University | | | | | | | |
| GOOD-WebKB | covariate | 244 | 61 | 61 | 125 | 126 | | | | | |
| | concept | 282 | 60 | 60 | 106 | 109 | | | | | |
| | | | | Color | | | | | | | |
| GOOD-CBAS | covariate | 420 | 70 | 70 | 70 | 70 | | | | | |
| | concept | 140 | 140 | 140 | 140 | 140 | | | | | |

Table 10: Number of Graphs/Nodes per dataset.

| Dataset | Model | # Model layers | Batch Size | # Max Epochs | # Iterations per epoch | Initial LR | Node Feature Dim |
|---|---|---|---|---|---|---|---|
| GOOD-SST2 | GIN-Virtual | 3 | 32 | 200/100 | – | 1e-3 | 768 |
| GOOD-CMNIST | GIN-Virtual | 5 | 128 | 500 | – | 1e-3 | 3 |
| GOOD-Motif | GIN | 3 | 32 | 200 | – | 1e-3 | 4 |
| GOOD-Cora | GCN | 3 | 4096 | 100 | 10 | 1e-3 | 8710 |
| GOOD-WebKB | GCN | 3 | 4096 | 100 | 10 | 1e-3/5e-3 | 1703 |
| GOOD-CBAS | GCN | 3 | 1000 | 200 | 10 | 3e-3 | 8 |

Table 11: Model and hyperparameters for GOOD datasets.

## A.8 GOOD DATASET ADDITIONAL RESULTS

We also include results on additional node classification benchmarks featuring distribution shift in Table 12. In Table 13, we compare models without G-ΔUQto the use of G-ΔUQwith randomly sampled anchors at the first or second layer.

Table 12: **Additional Node Classification Benchmarks.** Here, we compare accuracy and calibration error of G-ΔUQ and "no G-ΔUQ " (vanilla) models on 4 node classification benchmarks across concept and covariate shifts. First, we note that across all our evaluations, without any posthoc calibration, G-ΔUQ is superior to the vanilla model on nearly every benchmark for better or same accuracy (8/8 benchmarks) and better calibration error (7/8), often with a significant gain in calibration performance. However, due to the challenging nature of these shifts, achieving state-of-the-art calibration performance often requires the use of post-hoc calibration methods – so we also evaluate how these posthoc methods can be elevated when combined with G-ΔUQ (versus the vanilla variant). When combined with popular posthoc methods, we highlight that performance improves across the board, when combined with G-ΔUQ (including in WebKB and CBAS-Concept). For example, on WebKB – across the 9 calibration methods considered, "G-ΔUQ + calibration method" improves or maintains the calibration performance of the analogous "no G-ΔUQ + calibration method" in 7/9 (concept) and 6/9 (covariate). In CBAS, calibration is improved or maintained as the no-G-ΔUQ version on 5/9 (concept) and 9/9 (covariate). In all cases, this is achieved with little or no compromise on classification accuracy (often improving over "no G-ΔUQ" variant). We also emphasize that, across all the 8 evaluation sets (4 datasets x 2 shift types) in Table 10, the best performance is almost always obtained with a GDUQ variant: (accuracy: 8/8) as well as best calibration (6/8) or second best (2/8).

| | | | Shift: Concept | | | | Shift: Covariate | | | |
| | | | Accuracy (↑) | | ECE (↓) | | Accuracy (↑) | | ECE (↓) | |
| Dataset | Domain | Calibration | No G-Δ UQ | G-Δ UQ | No G-Δ UQ | G-Δ UQ | No G-Δ UQ | G-Δ UQ | No G-Δ UQ | G-Δ UQ |
|---|---|---|---|---|---|---|---|---|---|---|
| WebKB | University | × | 0.253±0.003 | **0.281±0.009** | 0.67±0.061 | 0.593±0.025 | 0.122±0.029 | 0.115±0.041 | 0.599±0.091 | 0.525±0.033 |
| | | CAGCN | 0.253±0.005 | 0.268±0.008 | 0.452±0.14 | 0.473±0.12 | 0.122±0.018 | 0.092±0.161 | 0.355±0.227 | 0.396±0.161 |
| | | Dirichlet | 0.229±0.018 | 0.22±0.022 | 0.472±0.06 | 0.472±0.03 | 0.244±0.105 | **0.295±0.044** | **0.299±0.092** | 0.328±0.044 |
| | | ETS | 0.253±0.005 | 0.273±0.012 | 0.64±0.06 | 0.575±0.019 | 0.121±0.021 | 0.084±0.027 | 0.539±0.112 | 0.499±0.02 |
| | | GATS | 0.253±0.005 | 0.273±0.01 | 0.608±0.008 | 0.485±0.02 | 0.122±0.018 | 0.079±0.029 | 0.455±0.057 | 0.376±0.029 |
| | | IRM | 0.251±0.005 | 0.266±0.011 | **0.342±0.017** | 0.349±0.006 | 0.097±0.04 | 0.046±0.013 | 0.352±0.037 | 0.422±0.013 |
| | | Orderinvariant | 0.253±0.005 | 0.27±0.01 | 0.628±0.026 | 0.564±0.024 | 0.122±0.018 | 0.106±0.065 | 0.545±0.079 | 0.47±0.065 |
| | | Spline | 0.237±0.012 | 0.257±0.023 | 0.436±0.029 | 0.386±0.034 | 0.122±0.013 | 0.171±0.056 | 0.472±0.031 | 0.39±0.056 |
| | | VS | 0.253±0.005 | 0.275±0.011 | 0.67±0.009 | 0.588±0.011 | 0.122±0.018 | 0.095±0.014 | 0.602±0.044 | 0.507±0.014 |
| Cora | Degree | × | 0.581±0.003 | 0.595±0.003 | 0.307±0.009 | 0.13±0.011 | 0.47±0.002 | 0.518±0.014 | 0.348±0.032 | 0.141±0.008 |
| | | CAGCN | 0.581±0.003 | **0.597±0.002** | 0.135±0.009 | 0.128±0.025 | 0.47±0.002 | 0.522±0.025 | 0.256±0.08 | 0.231±0.025 |
| | | Dirichlet | 0.534±0.007 | 0.551±0.004 | 0.12±0.004 | 0.196±0.003 | 0.414±0.007 | 0.449±0.01 | 0.163±0.002 | 0.356±0.01 |
| | | ETS | 0.581±0.003 | 0.596±0.004 | 0.301±0.009 | 0.116±0.018 | 0.47±0.002 | **0.523±0.003** | 0.31±0.077 | 0.141±0.003 |
| | | GATS | 0.581±0.003 | 0.596±0.004 | 0.185±0.018 | 0.229±0.039 | 0.47±0.002 | 0.521±0.011 | 0.211±0.004 | 0.308±0.011 |
| | | IRM | 0.582±0.002 | 0.597±0.002 | 0.125±0.001 | 0.102±0.002 | 0.469±0.001 | 0.522±0.004 | 0.194±0.005 | 0.13±0.004 |
| | | Orderinvariant | 0.581±0.003 | 0.592±0.002 | 0.226±0.024 | 0.213±0.049 | 0.47±0.002 | 0.498±0.027 | 0.318±0.042 | 0.196±0.027 |
| | | Spline | 0.571±0.003 | 0.595±0.003 | 0.080±0.004 | **0.068±0.004** | 0.459±0.003 | 0.52±0.004 | 0.158±0.01 | **0.098±0.004** |
| | | VS | 0.581±0.003 | 0.596±0.004 | 0.306±0.004 | 0.127±0.002 | 0.47±0.001 | 0.522±0.005 | 0.345±0.005 | 0.146±0.005 |
| Cora | Word | × | 0.607±0.003 | 0.628±0.001 | 0.284±0.009 | 0.111±0.013 | 0.603±0.004 | 0.633±0.031 | 0.263±0.004 | 0.118±0.019 |
| | | CAGCN | 0.607±0.002 | 0.628±0.002 | 0.138±0.011 | 0.236±0.019 | 0.603±0.004 | 0.634±0.035 | 0.129±0.009 | 0.253±0.035 |
| | | Dirichlet | 0.579±0.007 | 0.588±0.006 | 0.105±0.011 | 0.168±0.005 | 0.562±0.007 | 0.578±0.007 | 0.095±0.006 | 0.269±0.007 |
| | | ETS | 0.607±0.002 | 0.628±0.002 | 0.282±0.002 | 0.11±0.003 | 0.603±0.004 | 0.634±0.013 | 0.243±0.023 | 0.106±0.013 |
| | | GATS | 0.607±0.002 | 0.628±0.002 | 0.166±0.009 | 0.261±0.028 | 0.603±0.004 | 0.635±0.037 | 0.16±0.015 | 0.293±0.037 |
| | | IRM | 0.608±0.001 | 0.63±0.002 | 0.115±0.002 | 0.088±0.003 | 0.602±0.003 | 0.635±0.004 | 0.106±0.002 | 0.098±0.004 |
| | | Orderinvariant | 0.607±0.002 | 0.624±0.002 | 0.174±0.024 | 0.201±0.061 | 0.603±0.004 | 0.621±0.076 | 0.154±0.022 | 0.202±0.076 |
| | | Spline | 0.598±0.005 | 0.629±0.002 | 0.073±0.002 | **0.062±0.005** | 0.591±0.002 | 0.635±0.004 | 0.063±0.006 | **0.053±0.004** |
| | | VS | 0.607±0.001 | **0.63±0.002** | 0.283±0.003 | 0.111±0.003 | 0.603±0.004 | **0.636±0.003** | 0.261±0.005 | 0.119±0.003 |
| CBAS | Color | × | 0.83±0.014 | 0.829±0.011 | 0.169±0.013 | 0.151±0.014 | 0.703±0.015 | 0.746±0.027 | 0.266±0.02 | 0.169±0.018 |
| | | CAGCN | 0.83±0.013 | 0.83±0.013 | 0.137±0.011 | 0.143±0.022 | 0.703±0.019 | 0.749±0.033 | 0.25±0.021 | 0.186±0.017 |
| | | Dirichlet | 0.801±0.02 | 0.806±0.008 | 0.161±0.012 | 0.17±0.01 | 0.671±0.018 | 0.771±0.03 | 0.241±0.029 | 0.217±0.017 |
| | | ETS | 0.83±0.013 | 0.827±0.014 | 0.146±0.013 | 0.164±0.007 | 0.703±0.019 | 0.76±0.037 | 0.28±0.023 | 0.176±0.019 |
| | | GATS | 0.83±0.013 | 0.83±0.021 | 0.16±0.009 | 0.173±0.021 | 0.703±0.019 | 0.751±0.016 | 0.236±0.039 | 0.16±0.015 |
| | | IRM | 0.829±0.013 | 0.839±0.015 | 0.142±0.009 | **0.133±0.006** | 0.72±0.009 | 0.803±0.04 | 0.207±0.035 | **0.158±0.017** |
| | | Orderinvariant | 0.83±0.013 | 0.803±0.008 | 0.174±0.006 | 0.173±0.009 | 0.703±0.019 | 0.766±0.045 | 0.261±0.017 | 0.194±0.031 |
| | | Spline | 0.82±0.016 | 0.824±0.011 | 0.159±0.009 | 0.16±0.014 | 0.683±0.019 | 0.786±0.038 | 0.225±0.034 | 0.179±0.035 |
| | | VS | 0.829±0.012 | **0.840±0.011** | 0.166±0.011 | 0.146±0.012 | 0.717±0.019 | **0.809±0.008** | 0.242±0.019 | 0.182±0.014 |

Table 13: **Layerwise Anchoring for Node Classification Datasets with Intermediate Representation Distributions.** Here, we provide preliminary results for performing layerwise anchoring when performing node classification. We fit a gaussian distribution over the representations (similar to node feature anchoring) and then sample anchors from this distribution. We fit a gaussian distribution over the representations (similar to node feature anchoring) and then sample anchors from this distribution. We see that these alternative strategies do provide benefits in some cases, but overall, our original input node feature anchoring strategy is more performant.

| | | | Shift: Concept | | | | | | Shift: Covariate | | | | | |
| | | | | Accuracy (↑) | | | ECE (↓) | | | Accuracy (↑) | | | ECE (↓) | |
| Dataset | Domain | Calibration | No G-Δ UQ | Random 1 | Random 2 | No G-Δ UQ | Random 1 | Random 2 | No G-Δ UQ | Random 1 | Random 2 | No G-Δ UQ | Random 1 | Random 2 |
|---|---|---|---|---|---|---|---|---|---|---|---|---|---|---|
| CBAS | Color | Dirichlet | 0.801±0.02 | 0.765±0.012 | 0.839±0.023 | 0.161±0.012 | 0.301±0.018 | 0.234±0.027 | 0.671±0.018 | 0.74±0.023 | 0.689±0.032 | 0.241±0.029 | 0.349±0.04 | 0.381±0.029 |
| | | ETS | 0.83±0.013 | 0.819±0.012 | 0.82±0.088 | 0.146±0.013 | 0.23±0.017 | 0.257±0.021 | 0.703±0.019 | 0.638±0.051 | 0.686±0.026 | 0.28±0.023 | 0.347±0.037 | 0.334±0.028 |
| | | IRM | 0.829±0.013 | 0.821±0.019 | 0.885±0.026 | 0.142±0.009 | 0.219±0.012 | 0.206±0.066 | 0.72±0.019 | 0.617±0.084 | 0.693±0.026 | 0.207±0.035 | 0.363±0.03 | 0.299±0.036 |
| | | Orderinvariant | 0.83±0.013 | 0.813±0.015 | 0.819±0.028 | 0.174±0.006 | 0.255±0.015 | 0.236±0.006 | 0.703±0.019 | 0.831±0.008 | 0.636±0.026 | 0.261±0.017 | 0.286±0.039 | 0.303±0.062 |
| | | Spline | 0.82±0.016 | 0.814±0.022 | 0.839±0.035 | 0.159±0.009 | 0.235±0.017 | 0.196±0.036 | 0.683±0.019 | 0.621±0.052 | 0.757±0.026 | 0.225±0.034 | 0.312±0.026 | 0.331±0.024 |
| | | VS | 0.829±0.012 | 0.817±0.017 | 0.91±0.006 | 0.166±0.011 | 0.251±0.012 | 0.259±0.021 | 0.717±0.019 | 0.593±0.038 | 0.695±0.051 | 0.242±0.019 | 0.38±0.037 | 0.359±0.02 |
| Cora | Degree | Dirichlet | 0.534±0.007 | 0.483±0.014 | 0.423±0.007 | 0.12±0.004 | 0.355±0.004 | 0.347±0.004 | 0.414±0.007 | 0.466±0.073 | 0.425±0.005 | 0.163±0.002 | 0.315±0.042 | 0.345±0.007 |
| | | ETS | 0.581±0.003 | 0.562±0.01 | 0.496±0.002 | 0.301±0.009 | 0.297±0.009 | 0.289±0.006 | 0.47±0.002 | 0.498±0.119 | 0.511±0.005 | 0.34±0.076 | 0.511±0.005 | 0.329±0.008 |
| | | IRM | 0.582±0.002 | 0.567±0.011 | 0.492±0.003 | 0.125±0.001 | 0.072±0.003 | 0.116±0.006 | 0.469±0.001 | 0.499±0.117 | 0.508±0.005 | 0.194±0.005 | 0.094±0.009 | 0.105±0.006 |
| | | Orderinvariant | 0.581±0.003 | 0.566±0.004 | 0.495±0.002 | 0.226±0.024 | 0.151±0.015 | 0.14±0.008 | 0.47±0.002 | 0.499±0.107 | 0.108±0.034 | 0.318±0.042 | 0.506±0.005 | 0.093±0.009 |
| | | Spline | 0.571±0.003 | 0.561±0.011 | 0.493±0.005 | 0.080±0.004 | 0.11±0.01 | 0.119±0.005 | 0.459±0.003 | 0.499±0.12 | 0.508±0.006 | 0.158±0.01 | 0.105±0.03 | 0.127±0.012 |
| | | VS | 0.581±0.003 | 0.571±0.002 | 0.279±0.009 | 0.306±0.004 | 0.493±0.008 | 0.272±0.009 | 0.47±0.001 | 0.511±0.091 | 0.51±0.002 | 0.345±0.005 | 0.347±0.051 | 0.323±0.007 |
| Cora | Word | Dirichlet | 0.579±0.007 | 0.581±0.004 | 0.504±0.004 | 0.105±0.011 | 0.271±0.011 | 0.285±0.002 | 0.562±0.007 | 0.586±0.009 | 0.497±0.01 | 0.095±0.006 | 0.264±0.022 | 0.275±0.007 |
| | | ETS | 0.607±0.002 | 0.641±0.003 | 0.575±0.003 | 0.282±0.002 | 0.352±0.012 | 0.328±0.007 | 0.603±0.004 | 0.633±0.003 | 0.567±0.004 | 0.243±0.023 | 0.377±0.023 | 0.374±0.006 |
| | | IRM | 0.608±0.001 | 0.642±0.002 | 0.574±0.003 | 0.115±0.002 | 0.106±0.004 | 0.154±0.005 | 0.602±0.003 | 0.635±0.004 | 0.569±0.003 | 0.106±0.002 | 0.136±0.012 | 0.173±0.007 |
| | | Orderinvariant | 0.607±0.002 | 0.642±0.004 | 0.573±0.004 | 0.174±0.024 | 0.109±0.011 | 0.107±0.01 | 0.603±0.004 | 0.638±0.004 | 0.566±0.004 | 0.154±0.022 | 0.087±0.006 | 0.073±0.004 |
| | | Spline | 0.598±0.005 | 0.641±0.002 | 0.576±0.004 | 0.073±0.002 | 0.076±0.004 | 0.068±0.007 | 0.591±0.002 | 0.632±0.002 | 0.568±0.003 | 0.063±0.006 | 0.066±0.005 | 0.077±0.004 |
| | | VS | 0.607±0.001 | 0.639±0.003 | 0.583±0.005 | 0.283±0.003 | 0.345±0.007 | 0.335±0.012 | 0.603±0.004 | 0.637±0.004 | 0.579±0.004 | 0.261±0.005 | 0.396±0.028 | 0.384±0.005 |
| WebKB | University | Dirichlet | 0.229±0.018 | 0.214±0.000 | 0.228±0.012 | 0.472±0.06 | 0.56±0.000 | 0.552±0.041 | 0.244±0.105 | | 0.347±0.012 | 0.299±0.092 | | 0.429±0.05 |
| | | ETS | 0.253±0.005 | 0.279±0.000 | 0.234±0.01 | 0.64±0.06 | 0.437±0.000 | 0.33±0.022 | 0.121±0.021 | | 0.225±0.013 | 0.539±0.112 | | 0.258±0.028 |
| | | IRM | 0.251±0.005 | 0.251±0.000 | 0.232±0.009 | 0.342±0.017 | 0.379±0.000 | 0.459±0.01 | 0.097±0.04 | | 0.187±0.021 | 0.352±0.037 | | 0.294±0.018 |
| | | Orderinvariant | 0.253±0.005 | 0.279±0.000 | 0.237±0.01 | 0.628±0.026 | 0.568±0.000 | 0.53±0.049 | 0.122±0.018 | | 0.221±0.026 | 0.545±0.079 | | 0.321±0.061 |
| | | Spline | 0.237±0.012 | 0.237±0.000 | 0.233±0.008 | 0.436±0.029 | 0.467±0.000 | 0.483±0.041 | 0.122±0.013 | | 0.205±0.01 | 0.472±0.031 | | 0.329±0.035 |
| | | VS | 0.253±0.005 | 0.279±0.000 | 0.234±0.01 | 0.67±0.009 | 0.49±0.000 | 0.344±0.02 | 0.122±0.018 | | 0.201±0.011 | 0.602±0.044 | | 0.256±0.014 |

Table 14: **Layerwise Anchoring for Node Classification Datasets with Random Shuffling.** Here, we provide preliminary results for performing layerwise anchoring when performing node classification. We use random shuffling (similar to the proposed hidden layer strategy) to create the interemediate representations. We see that these alternative strategies do provide benefits.

| | | | Shift: Concept | | | | | | Shift: Covariate | | | | | |
| | | | | Accuracy (↑) | | | ECE (↓) | | | Accuracy (↑) | | | ECE (↓) | |
| Dataset | Domain | Calibration | No G-Δ UQ | Batch 1 | Batch 2 | No G-Δ UQ | Batch 1 | Batch 2 | No G-Δ UQ | Batch 1 | Batch 2 | No G-Δ UQ | Batch 1 | Batch 2 |
|---|---|---|---|---|---|---|---|---|---|---|---|---|---|---|
| CBAS | Color | Dirichlet | 0.801±0.02 | 0.757±0.045 | 0.58±0.046 | 0.161±0.012 | 0.309±0.059 | 0.431±0.033 | 0.671±0.018 | 0.548±0.035 | 0.629±0.019 | 0.241±0.029 | 0.48±0.03 | 0.407±0.01 |
| | | ETS | 0.83±0.013 | 0.699±0.036 | 0.637±0.014 | 0.146±0.013 | 0.265±0.013 | 0.258±0.015 | 0.703±0.019 | 0.562±0.087 | 0.507±0 | 0.28±0.023 | 0.37±0.021 | 0.333±0.02 |
| | | IRM | 0.829±0.013 | 0.711±0.031 | 0.724±0.029 | 0.142±0.009 | 0.284±0.032 | 0.291±0.02 | 0.72±0.019 | 0.59±0.079 | 0.657±0.037 | 0.207±0.035 | 0.336±0.032 | 0.268±0.037 |
| | | Orderinvariant | 0.83±0.013 | 0.788±0.007 | 0.574±0.051 | 0.174±0.006 | 0.268±0.023 | 0.208±0.055 | 0.703±0.019 | 0.61±0.011 | 0.5±0.019 | 0.261±0.017 | 0.334±0.035 | 0.249±0.037 |
| | | Spline | 0.82±0.016 | 0.695±0.039 | 0.652±0.042 | 0.159±0.009 | 0.279±0.018 | 0.236±0.013 | 0.683±0.019 | 0.49±0.124 | 0.6±0.032 | 0.225±0.034 | 0.364±0.034 | 0.308±0.054 |
| | | VS | 0.829±0.012 | 0.73±0.043 | 0.693±0.051 | 0.166±0.011 | 0.264±0.009 | 0.197±0.033 | 0.717±0.019 | 0.429±0.083 | 0.607±0.042 | 0.242±0.019 | 0.478±0.042 | 0.312±0.014 |
| Cora | Degree | Dirichlet | 0.534±0.007 | 0.515±0.003 | 0.442±0.012 | 0.12±0.004 | 0.304±0.01 | 0.315±0.004 | 0.414±0.007 | 0.507±0.004 | 0.419±0.006 | 0.163±0.002 | 0.28±0.006 | 0.338±0.004 |
| | | ETS | 0.581±0.003 | 0.576±0.011 | 0.516±0.013 | 0.301±0.009 | 0.317±0.018 | 0.285±0.007 | 0.47±0.002 | 0.563±0.003 | 0.496±0.005 | 0.31±0.077 | 0.373±0.009 | 0.311±0.006 |
| | | IRM | 0.582±0.002 | 0.579±0.009 | 0.523±0.008 | 0.125±0.001 | 0.076±0.004 | 0.129±0.004 | 0.469±0.001 | 0.562±0.004 | 0.494±0.004 | 0.194±0.005 | 0.088±0.011 | 0.098±0.003 |
| | | Orderinvariant | 0.581±0.003 | 0.582±0.003 | 0.518±0.005 | 0.226±0.024 | 0.134±0.023 | 0.126±0.012 | 0.47±0.002 | 0.561±0.004 | 0.496±0.004 | 0.318±0.042 | 0.091±0.014 | 0.096±0.007 |
| | | Spline | 0.571±0.003 | 0.58±0.006 | 0.518±0.011 | 0.080±0.004 | 0.093±0.007 | 0.092±0.007 | 0.459±0.003 | 0.565±0.004 | 0.496±0.005 | 0.158±0.01 | 0.091±0.009 | 0.128±0.012 |
| | | VS | 0.581±0.003 | 0.581±0.005 | 0.529±0.005 | 0.306±0.004 | 0.313±0.006 | 0.294±0.004 | 0.47±0.001 | 0.562±0.005 | 0.498±0.008 | 0.345±0.005 | 0.368±0.016 | 0.308±0.003 |
| Cora | Word | Dirichlet | 0.579±0.007 | 0.575±0.004 | 0.491±0.013 | 0.105±0.011 | 0.28±0.007 | 0.282±0.012 | 0.562±0.007 | 0.586±0.009 | 0.507±0.006 | 0.095±0.006 | 0.264±0.022 | 0.249±0.007 |
| | | ETS | 0.607±0.002 | 0.636±0.003 | 0.562±0.006 | 0.282±0.002 | 0.359±0.02 | 0.311±0.006 | 0.603±0.004 | 0.633±0.003 | 0.561±0.005 | 0.243±0.023 | 0.377±0.023 | 0.365±0.005 |
| | | IRM | 0.608±0.001 | 0.632±0.004 | 0.562±0.006 | 0.115±0.002 | 0.124±0.006 | 0.16±0.005 | 0.602±0.003 | 0.635±0.004 | 0.557±0.006 | 0.106±0.002 | 0.136±0.012 | 0.176±0.007 |
| | | Orderinvariant | 0.607±0.002 | 0.639±0.003 | 0.561±0.006 | 0.174±0.024 | 0.111±0.008 | 0.095±0.006 | 0.603±0.004 | 0.638±0.004 | 0.56±0.004 | 0.154±0.022 | 0.087±0.006 | 0.076±0.006 |
| | | Spline | 0.598±0.005 | 0.633±0.004 | 0.561±0.007 | 0.073±0.002 | 0.077±0.005 | 0.069±0.004 | 0.591±0.002 | 0.632±0.002 | 0.56±0.006 | 0.063±0.006 | 0.066±0.005 | 0.08±0.004 |
| | | VS | 0.607±0.001 | 0.633±0.006 | 0.574±0.007 | 0.283±0.003 | 0.368±0.009 | 0.32±0.005 | 0.603±0.004 | 0.637±0.004 | 0.573±0.008 | 0.261±0.005 | 0.396±0.028 | 0.373±0.006 |
| WebKB | University | Dirichlet | 0.229±0.018 | 0.231±0.015 | 0.234±0.007 | 0.472±0.06 | 0.562±0.014 | 0.534±0.022 | 0.244±0.105 | 0.242±0.166 | 0.298±0.077 | 0.299±0.092 | 0.468±0.092 | 0.483±0.055 |
| | | ETS | 0.253±0.005 | 0.277±0.007 | 0.234±0.003 | 0.64±0.06 | 0.421±0.017 | 0.327±0.015 | 0.121±0.021 | 0.128±0.017 | 0.101±0.033 | 0.539±0.112 | 0.437±0.032 | 0.293±0.01 |
| | | IRM | 0.251±0.005 | 0.265±0.019 | 0.232±0.014 | 0.342±0.017 | 0.377±0.015 | 0.438±0.015 | 0.097±0.04 | 0.118±0.033 | 0.093±0.034 | 0.352±0.037 | 0.482±0.02 | 0.435±0.016 |
| | | Orderinvariant | 0.253±0.005 | 0.268±0.01 | 0.231±0.01 | 0.628±0.026 | 0.513±0.071 | 0.431±0.025 | 0.122±0.018 | 0.122±0.018 | 0.1±0.029 | 0.545±0.079 | 0.475±0.049 | 0.38±0.069 |
| | | Spline | 0.237±0.012 | 0.242±0.01 | 0.228±0.014 | 0.436±0.029 | 0.415±0.042 | 0.484±0.035 | 0.122±0.013 | 0.129±0.024 | 0.097±0.013 | 0.472±0.031 | 0.478±0.033 | 0.425±0.013 |
| | | VS | 0.253±0.005 | 0.279±0.007 | 0.232±0.005 | 0.67±0.009 | 0.441±0.021 | 0.323±0.015 | 0.122±0.018 | 0.132±0.01 | 0.101±0.033 | 0.602±0.044 | 0.455±0.041 | 0.297±0.008 |

Table 15: **Alternative Anchoring Strategies.** Here, we consider an alternative anchoring formulation for graph classification. Namely, instead of shuffling features across the batch (denoted Batch in the table), we perform READOUT anchoring by fitting a normal distribution over the hidden representations. We then randomly sample from this distribution to create anchors. Conceptually, this is similar to the node feature anchoring strategy. One potential direction of future work that is permitted by this formulation is to optimize the parameters of this distribution given a signal from an appropriate auxiliary task or loss. For example, we could perform an alternating optimization where the GNN is trained to minimize the loss, and the mean and variance of the anchoring distribution are optimized to minimize the expected calibration error on a separate calibration dataset. While a rigorous formulation is left to future work, we emphasize that the potential for improving the anchoring distribution, and thus controlling corresponding hypothesis diversity, is in fact a unique benefit of G-$\Delta$UQ.

| Shift Type | Method | Test Acc MPNN | Test Acc Batch | Test Acc Random | Test Cal MPNN | Test Cal Batch | Test Cal Random | OOD Acc MPNN | OOD Acc Batch | OOD Acc Random | OOD Cal MPNN | OOD Cal Batch | OOD Cal Random |
|---|---|---|---|---|---|---|---|---|---|---|---|---|---|
| *GoodMotif, basis, concept* | Dirichlet | $0.995 \pm 0.0007$ | $0.994 \pm 0.0002$ | $0.996 \pm 0.0009$ | $0.040 \pm 0.0037$ | $0.036 \pm 0.0016$ | $0.035 \pm 0.0058$ | $0.924 \pm 0.0069$ | $0.923 \pm 0.0117$ | $0.942 \pm 0.0034$ | $0.080 \pm 0.0153$ | $0.102 \pm 0.0071$ | $0.062 \pm 0.0086$ |
| | ETS | $0.995 \pm 0.0007$ | $0.995 \pm 0.0005$ | $0.996 \pm 0.0007$ | $0.035 \pm 0.0034$ | $0.036 \pm 0.0101$ | $0.032 \pm 0.0052$ | $0.925 \pm 0.0095$ | $0.926 \pm 0.009$ | $0.7935 \pm 0.0068$ | $0.095 \pm 0.0098$ | $0.096 \pm 0.0128$ | $0.087 \pm 0.01451$ |
| | IRM | $0.9954 \pm 0.0007$ | $0.9957 \pm 0.0009$ | $0.9965 \pm 0.0004$ | $0.0198 \pm 0.0089$ | $0.0229 \pm 0.0105$ | $0.0225 \pm 0.0038$ | $0.9251 \pm 0.0096$ | $0.9301 \pm 0.0123$ | $0.9462 \pm 0.0024$ | $0.0873 \pm 0.0176$ | $0.0966 \pm 0.0103$ | $0.0907 \pm 0.0276$ |
| | OrderInvariant | $0.995 \pm 0.0007$ | $0.995 \pm 0.0005$ | $0.995 \pm 0.0005$ | $0.033 \pm 0.0094$ | $0.028 \pm 0.0047$ | $0.032 \pm 0.0009$ | $0.925 \pm 0.0095$ | $0.928 \pm 0.0104$ | $0.935 \pm 0.0027$ | $0.090 \pm 0.0092$ | $0.093 \pm 0.0070$ | $0.0754 \pm 0.0029$ |
| | Spline | $0.995 \pm 0.0007$ | $0.995 \pm 0.0007$ | $0.9962 \pm 0.0005$ | $0.034 \pm 0.0002$ | $0.035 \pm 0.0090$ | $0.032 \pm 0.0048$ | $0.924 \pm 0.0098$ | $0.926 \pm 0.0092$ | $0.937 \pm 0.0030$ | $0.091 \pm 0.0084$ | $0.089 \pm 0.0123$ | $0.083 \pm 0.0065$ |
| | VS | $0.995 \pm 0.0007$ | $0.995 \pm 0.0005$ | $0.996 \pm 0.000$ | $0.035 \pm 0.0034$ | $0.036 \pm 0.0087$ | $0.033 \pm 0.0098$ | $0.925 \pm 0.0094$ | $0.926 \pm 0.0095$ | $0.936 \pm 0.0053$ | $0.094 \pm 0.0096$ | $0.095 \pm 0.0133$ | $0.082 \pm 0.009$ |
| *GoodMotif,basis, covariate* | Dirichlet | $0.999 \pm 0.0003$ | $0.999 \pm 0.0004$ | $0.999 \pm 0.0002$ | $0.017 \pm 0.0054$ | $0.017 \pm 0.0019$ | $0.014 \pm 0.0004$ | $0.685 \pm 0.0504$ | $0.650 \pm 0.0450$ | $0.698 \pm 0.0139$ | $0.336 \pm 0.0667$ | $0.371 \pm 0.0474$ | $0.320 \pm 0.0140$ |
| | ETS | $0.9997 \pm 0.0004$ | $0.999 \pm 0.0005$ | $0.999 \pm 0.0002$ | $0.0095 \pm 0.0091$ | $0.017 \pm 0.0064$ | $0.017 \pm 0.0056$ | $0.690 \pm 0.0434$ | $0.649 \pm 0.0476$ | $0.686 \pm 0.0226$ | $0.313 \pm 0.0413$ | $0.3739 \pm 0.0485$ | $0.334 \pm 0.0167$ |
| | IRM | $0.9997 \pm 0.0004$ | $0.999 \pm 0.0006$ | $0.999 \pm 0.0003$ | $0.0085 \pm 0.0032$ | $0.010 \pm 0.0032$ | $0.014 \pm 0.0042$ | $0.690 \pm 0.0434$ | $0.647 \pm 0.0472$ | $0.692 \pm 0.0226$ | $0.315 \pm 0.0505$ | $0.354 \pm 0.0450$ | $0.328 \pm 0.0211$ |
| | OrderInvariant | $0.9997 \pm 0.0004$ | $0.999 \pm 0.0005$ | $0.999 \pm 0.0003$ | $0.014 \pm 0.0028$ | $0.020 \pm 0.0090$ | $0.013 \pm 0.0081$ | $0.690 \pm 0.0434$ | $0.649 \pm 0.0450$ | $0.689 \pm 0.0170$ | $0.320 \pm 0.0501$ | $0.358 \pm 0.0410$ | $0.328 \pm 0.0218$ |
| | Spline | $0.9997 \pm 0.0004$ | $0.999 \pm 0.0005$ | $0.999 \pm 0.0003$ | $0.016 \pm 0.0049$ | $0.017 \pm 0.0053$ | $0.017 \pm 0.0052$ | $0.690 \pm 0.0434$ | $0.649 \pm 0.0476$ | $0.6923 \pm 0.0199$ | $0.324 \pm 0.0548$ | $0.3733 \pm 0.0507$ | $0.327 \pm 0.0105$ |
| | VS | $0.9998 \pm 0.0001$ | $0.999 \pm 0.0003$ | $0.999 \pm 0.0002$ | $0.011 \pm 0.0053$ | $0.014 \pm 0.0034$ | $0.012 \pm 0.0016$ | $0.682 \pm 0.0561$ | $0.650 \pm 0.0546$ | $0.682 \pm 0.0251$ | $0.325 \pm 0.0568$ | $0.371 \pm 0.0591$ | $0.337 \pm 0.0264$ |
| *GOODSST2,length,concept* | Dirichlet | $0.938 \pm 0.0019$ | $0.939 \pm 0.0056$ | $0.942 \pm 0.00180$ | $0.189 \pm 0.01989$ | $0.165 \pm 0.0179$ | $0.187 \pm 0.0256$ | $0.694 \pm 0.0193$ | $0.693 \pm 0.0020$ | $0.687 \pm 0.0027$ | $0.146 \pm 0.0196$ | $0.133 \pm 0.015$ | $0.169 \pm 0.0168$ |
| | ETS | $0.938 \pm 0.0020$ | $0.939 \pm 0.0060$ | $0.941 \pm 0.0017$ | $0.389 \pm 0.0018$ | $0.390 \pm 0.0022$ | $0.393 \pm 0.0007$ | $0.6940 \pm 0.0193$ | $0.692 \pm 0.0019$ | $0.687 \pm 0.0034$ | $0.214 \pm 0.0098$ | $0.216 \pm 0.0033$ | $0.220 \pm 0.0057$ |
| | IRM | $0.939 \pm 0.0016$ | $0.939 \pm 0.0058$ | $0.941 \pm 0.0018$ | $0.326 \pm 0.0011$ | $0.326 \pm 0.0013$ | $0.327 \pm 0.0017$ | $0.693 \pm 0.0185$ | $0.692 \pm 0.0026$ | $0.685 \pm 0.0026$ | $0.240 \pm 0.0017$ | $0.232 \pm 0.0050$ | $0.242 \pm 0.0053$ |
| | OrderInvariant | $0.938 \pm 0.0020$ | $0.939 \pm 0.0060$ | $0.941 \pm 0.0022$ | $0.314 \pm 0.0014$ | $0.315 \pm 0.0029$ | $0.315 \pm 0.0012$ | $0.6940 \pm 0.0193$ | $0.692 \pm 0.0019$ | $0.687 \pm 0.0033$ | $0.224 \pm 0.0010$ | $0.222 \pm 0.0030$ | $0.223 \pm 0.0054$ |
| | Spline | $0.938 \pm 0.0026$ | $0.938 \pm 0.0044$ | $0.941 \pm 0.0010$ | $0.329 \pm 0.0021$ | $0.329 \pm 0.0019$ | $0.328 \pm 0.0012$ | $0.6940 \pm 0.0193$ | $0.692 \pm 0.0022$ | $0.687 \pm 0.0035$ | $0.234 \pm 0.0052$ | $0.231 \pm 0.0044$ | $0.243 \pm 0.0034$ |
| | VS | $0.938 \pm 0.0027$ | $0.939 \pm 0.0057$ | $0.941 \pm 0.0018$ | $0.290 \pm 0.2099$ | $0.484 \pm 0.0008$ | $0.487 \pm 0.0007$ | $0.693 \pm 0.0184$ | $0.693 \pm 0.0018$ | $0.687 \pm 0.0031$ | $0.331 \pm 0.0484$ | $0.375 \pm 0.0022$ | $0.382 \pm 0.0048$ |
| *GOODSST2,length,covariate* | Dirichlet | $0.896 \pm 0.0029$ | $0.893 \pm 0.0009$ | $0.895 \pm 0.00095$ | $0.196 \pm 0.0155$ | $0.172 \pm 0.0091$ | $0.1797 \pm 0.0109$ | $0.825 \pm 0.0037$ | $0.827 \pm 0.0066$ | $0.805 \pm 0.0150$ | $0.163 \pm 0.0198$ | $0.141 \pm 0.0087$ | $0.142 \pm 0.0122$ |
| | ETS | $0.8966 \pm 0.0023$ | $0.894 \pm 0.0011$ | $0.894 \pm 0.0006$ | $0.357 \pm 0.0013$ | $0.359 \pm 0.0004$ | $0.362 \pm 0.0019$ | $0.826 \pm 0.0036$ | $0.828 \pm 0.0065$ | $0.806 \pm 0.0117$ | $0.309 \pm 0.0050$ | $0.314 \pm 0.0076$ | $0.300 \pm 0.0070$ |
| | IRM | $0.895 \pm 0.0019$ | $0.893 \pm 0.0003$ | $0.894 \pm 0.0007$ | $0.307 \pm 0.0004$ | $0.307 \pm 0.0003$ | $0.306 \pm 0.0020$ | $0.826 \pm 0.0040$ | $0.828 \pm 0.0065$ | $0.809 \pm 0.0152$ | $0.276 \pm 0.0046$ | $0.277 \pm 0.0061$ | $0.265 \pm 0.0078$ |
| | OrderInvariant | $0.896 \pm 0.0023$ | $0.894 \pm 0.0011$ | $0.894 \pm 0.0008$ | $0.288 \pm 0.0008$ | $0.285 \pm 0.0008$ | $0.284 \pm 0.0013$ | $0.826 \pm 0.0036$ | $0.828 \pm 0.0065$ | $0.806 \pm 0.0106$ | $0.244 \pm 0.0022$ | $0.241 \pm 0.0037$ | $0.225 \pm 0.0054$ |
| | Spline | $0.894 \pm 0.0016$ | $0.890 \pm 0.0009$ | $0.892 \pm 0.0040$ | $0.309 \pm 0.0024$ | $0.307 \pm 0.0009$ | $0.307 \pm 0.0022$ | $0.822 \pm 0.0026$ | $0.822 \pm 0.0092$ | $0.801 \pm 0.0110$ | $0.275 \pm 0.0043$ | $0.276 \pm 0.0063$ | $0.264 \pm 0.0063$ |
| | VS | $0.8963 \pm 0.0028$ | $0.893 \pm 0.0008$ | $0.894 \pm 0.0007$ | $0.291 \pm 0.1833$ | $0.460 \pm 0.0011$ | $0.465 \pm 0.0010$ | $0.821 \pm 0.0053$ | $0.827 \pm 0.0071$ | $0.806 \pm 0.0119$ | $0.299 \pm 0.1395$ | $0.431 \pm 0.0061$ | $0.429 \pm 0.0054$ |

### A.9 Post-hoc Calibration Strategies

Several post hoc strategies have been developed for calibrating the predictions of a model. These have the advantage of flexibility, as they operate only on the outputs of a model and do not require that any changes be made to the model itself. Some methods include:

- **Temperature scaling (TS)** (Guo et al., 2017) simply scales the logits by a temperature parameter $T > 1$ to smooth the predictions. The scaling parameter $T$ can be tuned on a validation set.

- **Ensemble temperature scaling (ETS)** (Zhang et al., 2020) learns an ensemble of temperature-scaled predictions with uncalibrated predictions ($T = 1$) and uniform probabilistic outputs ($T = \infty$).

- **Vector scaling** (VS) Guo et al. (2017) scales the entire output vector of class probabilities, rather than just the logits.

- **Multi-class isotonic regression (IRM)** (Zhang et al., 2020) is a multiclass generalization of the famous isotonic regression method (Zadrozny & Elkan, 2002)): it ensembles predictions and labels, then learns a monotonically increasing function to map transformed predictions to labels.

- **Order-invariant calibration** (Rahimi et al., 2020) uses a neural network to learn an intra-order-preserving calibration function that can preserve a model's top-k predictions.

- **Spline** calibration instead uses splines to fit the calibration function (Gupta et al., 2021).

- **Dirichlet calibration** (Kull et al., 2019) models the distribution of outputs using a Dirichlet distribution, using simple log-transformation of the uncalibrated probabilities which are then passed to a regularized fully connected neural network layer with softmax activation.

For node classification, some graph-specific post-hoc calibration methods have been proposed. **CaGCN** (Wang et al., 2021) uses the graph structure and an additional GCN to produce node-wise temperatures. GATS (Hsu et al., 2022) extends this idea by using graph attention to model the influence of neighbors' temperatures when learning node-wise temperatures. We use the post hoc calibration baselines provided by Hsu et al. in our experiments.

All of the above methods, and others, may be applied to the output of any model including one using G-$\Delta$UQ. As we have shown, applying such post hoc methods to the outputs of the calibrated models may improve uncertainty estimates even more. Notably, calibrated models are expected to produce confidence estimates that match the true probabilities of the classes being predicted (Naeini et al., 2015; Guo et al., 2017; Ovadia et al., 2019). While poorly calibrated CIs are over/under confident in their predictions, calibrated CIs are more trustworthy and can also improve performance on other safety-critical tasks which implicitly require reliable prediction probabilities (see Sec. 5). We report the top-1 label expected calibration error (ECE) (Kumar et al., 2019; Detlefsen et al., 2022). Formally, let $p_i$ be the top-1 probability, $c_i$ be the predicted confidence, $b_i$ a uniformly sized bin in $[0, 1]$. Then,

$$ECE := \sum_{i}^{N} b_i \|(p_i - c_i)\|$$

.

## A.10 Details on Generalization Gap Prediction

Accurate estimation of the expected generalization error on unlabeled datasets allows models with unacceptable performance to be pulled from production. To this end, generalization error predictors (GEPs) (Garg et al., 2022; Ng et al., 2022; Jiang et al., 2019; Trivedi et al., 2023a; Guillory et al., 2021) which assign sample-level scores, $S(x_i)$ which are then aggregated into dataset-level error estimates, have become popular. We use maximum softmax probability and a simple thresholding mechanism as the GEP (since we are interested in understanding the behavior of confidence indicators), and report the error between the predicted and true target dataset accuracy: $GEPError := ||\text{Acc}_{target} - \frac{1}{|X|}\sum_i \mathbb{I}(\text{S}(\bar{\text{x}}_i; \text{F}) > \tau)||$ where $\tau$ is tuned by minimizing GEP error on the validation dataset. We use the confidences obtained by the different baselines as sample-level scores, $\text{S}(\text{x}_i)$ corresponding to the model's expectation that a sample is correct. The MAE between the estimated error and true error is reported on both in- and out-of -distribution test splits provided by the GOOD benchmark.

## A.11 Results on Generalization Error Prediction

**GEP Experimental Setup.** GEPs (Garg et al., 2022; Ng et al., 2022; Jiang et al., 2019; Trivedi et al., 2023a; Guillory et al., 2021) aggregate sample-level scores capturing a model's uncertainty about the correctness of a prediction into dataset-level error estimates. Here, we use maximum softmax probability for scores and a thresholding mechanism as the GEP. (See Appendix A.10 for more details.) We consider READOUT anchoring with both pretrained and end-to-end training, and report the mean absolute error between the predicted and true target dataset accuracy on the OOD test split.

**GEP Results**. As shown in Table 16, both pretrained and end-to-end G-$\Delta$UQ outperform the vanilla model on 7/8 datasets. Notably, we see that pretrained G-$\Delta$UQ is particularly effective as it obtains the best performance across 6/8 datasets. This not only highlights its utility as a flexible, light-weight strategy for improving uncertainty estimates without sacrificing accuracy, but also emphasizes that importance of structure, in lieu of full stochasticity, when estimating GNN uncertainties.

Table 16: **GOOD-Datasets, Generalization Error Prediction Performance**. The MAE between the predicted and true test error on the OOD test split is reported. G-$\Delta$UQ variants outperform vanilla models on 7/8 datasets (GOODMotif(Basis,Covariate) being the exception). Pretrained G-$\Delta$UQ is particularly effective at this task as it achieves the best performance overall on 6/8 datasets. Promisingly, we see that regular G-$\Delta$UQ improves performance over the vanilla model on 6/8 datasets (even if it is not the best overall). We further observe that performing generalization error prediction is more challenging under covariate shift than concept shift on the GOODCMNIST, GOODMotif(Basis) and GOODMotif(Size) datasets. On these datasets, the MAE is almost twice as large than their respective concept shift counterparts, across methods. GOODSST2 is the exception, where concept shift is in fact more challenging. To the best our knowledge, we are the first to investigate generalization error prediction on GNN-based tasks under distribution shift. Understanding this behavior further is an interesting direction of future work.

| Method | CMNIST (Color) | | MotifLPE (Basis) | | MotifLPE (Size) | | SST2 | |
|---|---|---|---|---|---|---|---|---|
| | Concept($\downarrow$) | Covariate ($\downarrow$) | Concept($\downarrow$) | Covariate($\downarrow$) | Concept($\downarrow$) | Covariate($\downarrow$) | Concept($\downarrow$) | Covariate($\downarrow$) |
| Vanilla | $0.200 \pm 0.009$ | $0.510 \pm 0.089$ | $0.045 \pm 0.003$ | $0.570 \pm 0.012$ | $0.324 \pm 0.018$ | $0.537 \pm 0.146$ | $0.117 \pm 0.006$ | $0.056 \pm 0.044$ |
| G-$\Delta$UQ | $0.190 \pm 0.010$ | $0.493 \pm 0.072$ | $0.023 \pm 0.003$ | $0.572 \pm 0.019$ | $0.317 \pm 0.007$ | $0.528 \pm 0.189$ | $0.124 \pm 0.016$ | $0.054 \pm 0.043$ |
| Pretr. G-$\Delta$UQ | $0.192 \pm 0.005$ | $0.387 \pm 0.048$ | $0.018 \pm 0.012$ | $0.573 \pm 0.004$ | $0.307 \pm 0.016$ | $0.356 \pm 0.143$ | $0.114 \pm 0.004$ | $0.030 \pm 0.026$ |

## A.12 ADDITIONAL STUDY ON PRETRAINED G-ΔUQ

For the datasets and data shifts on which we reported out-of-distribution calibration error of pretrained vs. in-training G-ΔUQ earlier in Fig. 4, we now report additional results for in-distribution and out-of distribution accuracy as well as calibration error. We also include results for the additional GOODMotif-basis benchmark for completeness, noting that the methods provided by the original benchmark (Gui et al., 2022) generalized poorly to this split (which may be related to why G-ΔUQ methods offer little improvement over the vanilla model.) Fig. 8 shows these extended results. By these additional metrics, we again see the competitiveness of applying G-ΔUQ to a pretrained model versus using it in end-to-end training.

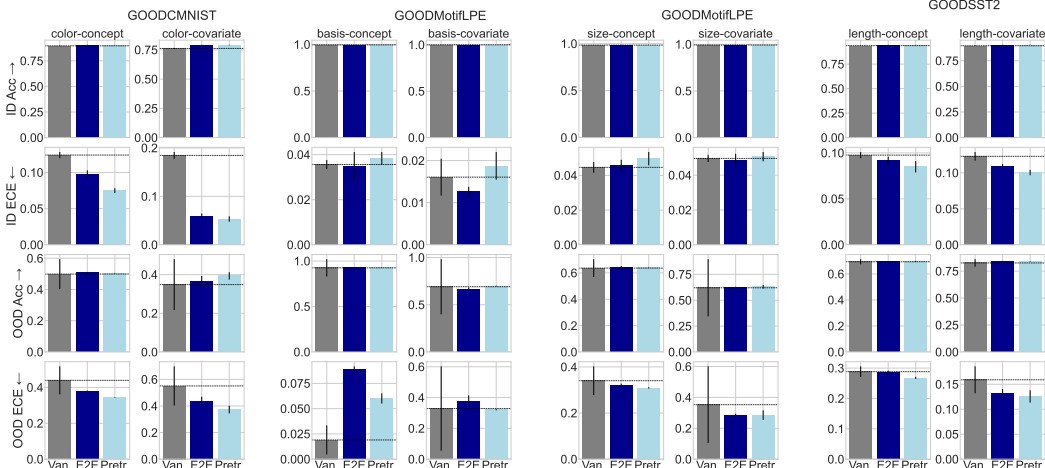

Figure 8: **Evaluating Pretrained G-ΔUQ.** Here, we report the performance of pretrained G-ΔUQ models vs. end-to-end and vanilla models with respect to in-distribution and out-of-distribution accuracy as well as expected calibration error. With the exception of the GOODMotif (basis) dataset, pretrained G-ΔUQ improves the OOD ECE over both the vanilla model and end-to-end G-ΔUQ at comparable or improved OOD accuracy on 7/8 datasets. Furthermore, pretrained G-ΔUQ also improves the ID ECE on all but the GOODMotif (size) datasets (6/8), where it performs comparably to the vanilla model, and maintains the ID accuracy. (We note that all methods are comparably better calibrated on the GOODMotif ID data than GOODCMIST/GOODSST2 ID data; we suspect this is because there may exist simple shortcuts available in the GOODMotif dataset that can be used on the ID test set effectively.) Overall, these results clearly demonstrate that pretrained G-ΔUQ does offer some performance advantages over end-to-end G-ΔUQ and does so at reduced training times (see Table. A.13). For example, on GOODCMNIST (covariate shift), pretrained G-ΔUQ is not only 50% faster than end-to-end G-ΔUQ , it also improves OOD accuracy and OOD ECE over both the vanilla and end-to-end G-ΔUQmodels.

## A.13 RUNTIME TABLE

Table 17: **Runtimes.** We include the runtimes of both training per epoch (in seconds) and performing calibration. Reducing stochasticity can help reduce computation (L1 → L3). Cost can also be reduced by using a pretrained model.

| | GOODCMNIST | | GOODSST2 | | GOODMotifLPE | |
|---|---|---|---|---|---|---|
| Dataset | Training (S) | Inference (S) | Training (S) | Inference (S) | Training (S) | Inference (S) |
| Vanilla | 18.5 | 25.8 | 10.8 | 18.5 | 3.8 | 4.5 |
| Temp. Scaling | 18.5 | 23.5 | 10.8 | 13.4 | 3.8 | 5.3 |
| DEns (Ens Size=3) | 18.456 x Ens Size | 59.4 | 10.795 x Ens Size | 29.0 | 3.8 x Ens Size | 11.8 |
| G-$\Delta$UQ (L1, 10 anchors) | 22.1 | 181.5 | 15.9 | 17.1 | 5.8 | 15.5 |
| G-$\Delta$UQ (L2, 10) | 22.4 | 148.6 | 12.7 | 15.5 | 5.8 | 11.8 |
| G-$\Delta$UQ (HiddenRep, 10) | 18.5 | 28.0 | 13.8 | 19.6 | 3.9 | 6.5 |
| G-$\Delta$UQ (Pretr. HiddenRep, 10) | 8.6 | 27.8 | 6.8 | 16.0 | 2.5 | 6.4 |

## A.14 MEAN AND VARIANCE OF NODE FEATURE GAUSSIANS

Table 18: **Mean and Variance of Node Feature Anchoring Gaussians.** We report the mean and variance of the Gaussian distributions fitted to the input node features. Because the input node features vary in size, we report aggregate statistics over the mean and variance corresponding to each dimension. For example, Min(Mu) indicates that we are reports the minimum mean over the $d$-dim set of means.

| Dataset | Domain | Shift | Min (Mu) | Max (Mu) | Mean (Mu) | Std (Mu) | Min (Std) | Max (Std) | Mean (Std) | Std (Std) |
|---|---|---|---|---|---|---|---|---|---|---|
| GOODSST2 | length | concept | -4.563 | 0.69 | -0.011 | 0.278 | 0.163 | 0.803 | 0.242 | 0.049 |
| GOODSST2 | length | covariate | -4.902 | 0.684 | -0.01 | 0.3 | 0.175 | 0.838 | 0.255 | 0.05 |
| GOODCMNIST | color | concept | 0.117 | 0.133 | 0.127 | 0.008 | 0.092 | 0.097 | 0.095 | 0.003 |
| GOODCMNIST | color | covariate | 0.087 | 0.131 | 0.102 | 0.025 | 0.108 | 0.109 | 0.108 | 0 |
| GOODMotifLPE | size | covariate | 0.003 | 0.021 | 0.011 | 0.008 | 0.835 | 1.728 | 1.248 | 0.377 |
| GOODMotifLPE | size | concept | -0.006 | 0 | -0.002 | 0.003 | 0.542 | 1.114 | 0.783 | 0.242 |
| GOODMotifLPE | basis | concept | -0.011 | 0.015 | 0.001 | 0.011 | 0.721 | 1.464 | 1.09 | 0.304 |
| GOODMotifLPE | basis | covariate | -0.007 | -0.002 | -0.004 | 0.002 | 0.808 | 1.913 | 1.251 | 0.469 |
| GOODWebKB | university | concept | 0 | 0.95 | 0.049 | 0.099 | 0.001 | 0.5 | 0.168 | 0.095 |
| GOODWebKB | university | covariate | 0 | 0.934 | 0.05 | 0.104 | 0.001 | 0.5 | 0.164 | 0.098 |
| GOODCora | degree | concept | 0 | 0.507 | 0.007 | 0.017 | 0.001 | 0.5 | 0.061 | 0.051 |
| GOODCora | degree | covariate | 0 | 0.518 | 0.007 | 0.017 | 0.001 | 0.5 | 0.061 | 0.052 |
| GOODCBAS | color | covariate | 0.394 | 0.591 | 0.471 | 0.093 | 0.142 | 0.492 | 0.403 | 0.174 |
| GOODCBAS | color | concept | 0.23 | 0.569 | 0.4 | 0.144 | 0.168 | 0.495 | 0.39 | 0.152 |

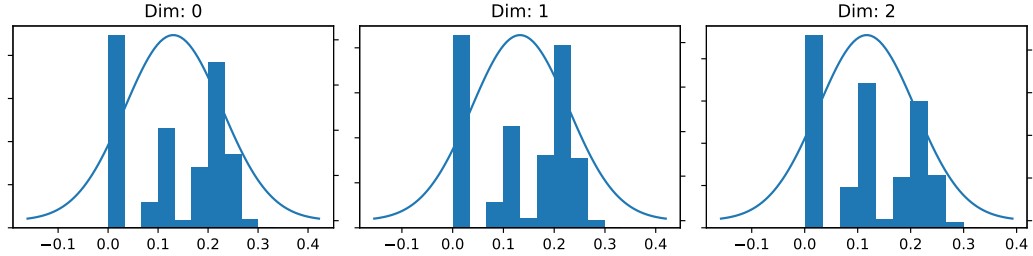

Figure 9: **GOODCMNIST, Concept, Anchoring Distribution.** We plot the mean and variance of the fitted anchoring distribution vs. the true feature distribution for each input dimension. We observe there is a mismatch between the empircal distribution and the fitted Gaussian. However, we did not find this mismatch to harm the effectiveness of G-ΔUQ.

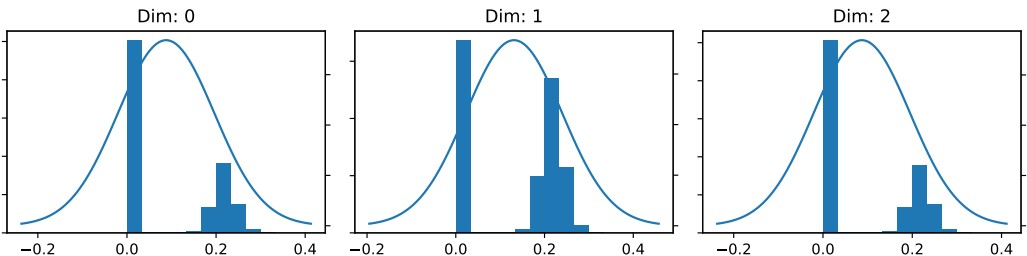

Figure 10: **GOODCMNIST, Covariate, Anchoring Distribution.** We plot the mean and variance of the fitted anchoring distribution vs. the true feature distribution for each input dimension. We observe there is a mismatch between the empircal distribution and the fitted Gaussian. However, we did not find this mismatch to harm the effectiveness of G-ΔUQ.

Table 19: **Number of Parmeters per Model.** We provide the number of parameters in the vanilla and modified parameter as follows. Note, that the change in parameters is architecture and input dimension dependent. For example, GOODCMNIST, and GOODSST2 use GIN MPNN layers. Therefore, when changing the layer dimension, we are changing the dimension of its internal MLP. It is not an error that intermediate layer G-$\Delta$UQhave the same number of parameters, this is due to the architecture: these layers are the same size in the vanilla model. Likewise, GOODCora's input features have dimension is 8701, so doubling the input layer's dimension appears to add a signficant number of parameters. We do not believe this

| Dataset | GOODCMNIST | GOODMotif | GOODSST2 | GOODCORA | GOODWebKB | GOODCBAS |
|---|---|---|---|---|---|---|
| Baseline | 2001310 | 911403 | 1732201 | 2816770 | 695105 | 185104 |
| G-$\Delta$UQ(NFA) | 2003110 | 913803 | 2193001 | 5429770 | 1206005 | 186304 |
| G-$\Delta$UQ(L1) | 2360110 | 1633203 | 2091001 | | | |
| G-$\Delta$UQ(L2) | 2360110 | 1633203 | 2091001 | | | |
| G-$\Delta$UQ(L3) | 2360110 | | | | | |
| G-$\Delta$UQ(L4) | 2360110 | | | | | |
| G-$\Delta$UQ(Readout) | 2004310 | 912303 | 1732501 | | | |

### A.15 EXPANDED DISCUSSION ON ANCHORING DESIGN CHOICES

Below, we expand upon some of the design choices for the proposed anchoring strategies.

**When performing node featuring anchoring, how does fitting a Gaussian distribution to the input node features help manage the combinatorial stochasticity induced by message passing?**

Without loss of generality, consider a node classification setting, where every sample is assigned a unique anchor. Then, due to message passing, after $l$ hops, a given node's representation will have aggregated information from its $l$ hop neighborhood. However, since each node in this neighborhood has a unique anchor, we see that any given node's representation is not only stochastic due to its own anchor but also that of its neighbors. For example, if any of its neighbors are assigned a different anchor, then the given node's representation will change, even if its own anchor did not. Since this behavior holds true for all nodes and each of their respective neighborhoods, we loosely refer to this phenomenon having combinatorial complexity, as effectively marginalizing out the anchoring distribution would require handling any and all changes to all $l$-hop neighbors. In contrast, when performing anchored image classification, the representation of a sample is only dependent on its unique, corresponding anchor, and is not influenced by the anchors of other samples. To this end, using the fitted Gaussian distribution helps manage this complexity, since changes to the anchors of a node's $l$-hop neighborhood are simpler to model as they require only learning to marginalize out a Gaussian distribution (instead of the training distribution). Indeed, for example, if we were to assume simplified model where message passing only summed node neighbors, the anchoring distribution would remain Gaussian after $l$ rounds of message passing since the sum of Gaussian is still Gaussian (the exact parameters of the distribution would depend on the normalization used however).

