# A  APPENDIX

## A.1 PSEUDOCODE FOR G-ΔUQ

```
class GNN:
    def __init__():
        self.make_layers()
        self.MPNN1 = MPNN(InputDim,HDim1)
        self.MPNN2 = MPNN(HDim1,HDim2)
        self.MPNN3 = MPNN(HDim2, HDim3)
        self.READOUT = READOUTLayer()
        self.classifier = Linear(HDim3, NumClass)

    def forward(self, X,Adj):
        h = self.MPNN1(X,Adj) # Out: (B,N,HDim1)
        h = self.MPNN2(h,Adj) # Out: (B, N,HDim2)
        h = self.MPNN3(h,Adj) # Out: (B,N,HDim3)
        g = self.READOUT(h,Batch) # Out: (B, HDim3)
        pred = self.classifier(g) # Out: (B, NumClass)
        return pred
```

(a) Vanilla GNN

```
class GNN_NodeFeatureAnc:
    def __init__(GNN):
        self.MPNN1 = MPNN(InputDim * 2,HDim1)
        self.MPNN2 = MPNN(HDim1,HDim2)
        self.MPNN3 = MPNN(HDim2, HDim3)
        self.READOUT = READOUTLayer()
        self.classifier = Linear(HDim3, NumClass)
        self.make_anchor_dist(AggNodeFeats)

    def make_anchor_dist(NodeFeats):
        mu = MEAN(AggNodeFeats,dim=0)
        std = STD(AggNodeFeats,dim=0)
        #d-dim Gaussian
        self.AnchorDist = Normal(mean=mu, std=std)

    def forward(X,Adj):
        # (N,InputDim)
        C = self.AnchorDist.sample(X.shape[0])
        # (N,InputDim x 2)
        AncRep = CONCAT([X - C, C], dim=1)
        h = self.MPNN1(AncRep,Adj) #Out: (N,HDim1)
        h = self.MPNN2(h,Adj)
        h = self.MPNN3(h,Adj)
        g = self.READOUT(h,Batch)
        pred = self.classifier(g)
        return pred
```

(b) G-ΔUQ with Node Feature Anchoring

```
class GNN_HiddnRepAnc:
    def __init__(GNN):
        self.MPNN1 = MPNN(InputDim,HDim1)
        # Corres. to GDUQLayer!
        self.MPNN2 = MPNN(HDim1 * 2,HDim2)
        self.MPNN3 = MPNN(HDim2, HDim3)
        self.READOUT = READOUTLayer()
        self.classifier = Linear(HDim3, NumClass)

    def forward(X,Adj):
        h = self.MPNN1(X,Adj)
        C = SHUFFLE(h, dim=0)
        hAnc = CONCAT([h-C, C],dim=1)
        h = self.MPNN2(hAnc,Adj) #Out: (N, HDim2)

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

| --- | --- | --- | --- | --- | --- | --- | --- | --- | --- | --- | --- | --- | --- |
| | | MPNN | Batch | Random | MPNN | Batch | Random | MPNN | Batch | Random | MPNN | Batch | Random |
| GoodMotif, basis, concept | Dirichlet | 0.995 ± 0.0007 | 0.994 ± 0.0002 | 0.996 ± 0.0009 | 0.040 ± 0.0037 | 0.036 ± 0.0016 | 0.035 ± 0.0058 | 0.924 ± 0.0069 | 0.923 ± 0.0117 | 0.942 ± 0.0034 | 0.080 ± 0.0153 | 0.102 ± 0.0071 | 0.062 ± 0.0086 |
| | ETS | 0.995 ± 0.0007 | 0.995 ± 0.0005 | 0.996 ± 0.0007 | 0.035 ± 0.0034 | 0.036 ± 0.0101 | 0.032 ± 0.0052 | 0.925 ± 0.0095 | 0.926 ± 0.009 | 0.935 ± 0.0068 | 0.095 ± 0.0098 | 0.096 ± 0.0128 | 0.087 ± 0.01451 |
| | IRM | 0.9954 ± 0.0007 | 0.9957 ± 0.0009 | 0.9965 ± 0.0004 | 0.0198 ± 0.0089 | 0.0229 ± 0.0105 | 0.0225 ± 0.0038 | 0.9251 ± 0.0096 | 0.9301 ± 0.0123 | 0.9462 ± 0.0024 | 0.0873 ± 0.0176 | 0.0966 ± 0.0103 | 0.0907 ± 0.0276 |
| | OrderInvariant | 0.995 ± 0.0007 | 0.995 ± 0.0005 | 0.995 ± 0.0005 | 0.033 ± 0.0094 | 0.028 ± 0.0047 | 0.032 ± 0.0009 | 0.925 ± 0.0095 | 0.928 ± 0.0104 | 0.935 ± 0.0027 | 0.090 ± 0.0092 | 0.093 ± 0.0070 | 0.0754 ± 0.0029 |
| | Spline | 0.995 ± 0.0007 | 0.995 ± 0.0007 | 0.9962 ± 0.0005 | 0.034 ± 0.0002 | 0.035 ± 0.0090 | 0.032 ± 0.0048 | 0.924 ± 0.0098 | 0.926 ± 0.0092 | 0.937 ± 0.0030 | 0.091 ± 0.0084 | 0.089 ± 0.0123 | 0.083 ± 0.0065 |
| | VS | 0.995 ± 0.0007 | 0.995 ± 0.0005 | 0.996 ± 0.000 | 0.035 ± 0.0034 | 0.036 ± 0.0087 | 0.033 ± 0.0098 | 0.925 ± 0.0094 | 0.926 ± 0.0095 | 0.936 ± 0.0053 | 0.094 ± 0.0096 | 0.095 ± 0.0133 | 0.082 ± 0.009 |
| GoodMotif,basis, covariate | Dirichlet | 0.999 ± 0.0003 | 0.999 ± 0.0004 | 0.999 ± 0.0002 | 0.017 ± 0.0054 | 0.017 ± 0.0019 | 0.014 ± 0.0004 | 0.685 ± 0.0504 | 0.650 ± 0.0450 | 0.698 ± 0.0139 | 0.336 ± 0.0667 | 0.371 ± 0.0474 | 0.320 ± 0.0140 |
| | ETS | 0.9997 ± 0.0004 | 0.999 ± 0.0005 | 0.999 ± 0.0002 | 0.0095 ± 0.0091 | 0.017 ± 0.0064 | 0.017 ± 0.0056 | 0.690 ± 0.0434 | 0.649 ± 0.0476 | 0.686 ± 0.0226 | 0.313 ± 0.0413 | 0.3739 ± 0.0485 | 0.334 ± 0.0167 |
| | IRM | 0.9997 ± 0.0004 | 0.999 ± 0.0006 | 0.999 ± 0.0003 | 0.0085 ± 0.0032 | 0.010 ± 0.0032 | 0.014 ± 0.0042 | 0.690 ± 0.0434 | 0.647 ± 0.0472 | 0.692 ± 0.0226 | 0.315 ± 0.0505 | 0.354 ± 0.0450 | 0.328 ± 0.0211 |
| | OrderInvariant | 0.9997 ± 0.0004 | 0.999 ± 0.0005 | 0.999 ± 0.0003 | 0.014 ± 0.0028 | 0.020 ± 0.0090 | 0.013 ± 0.0081 | 0.690 ± 0.0434 | 0.649 ± 0.0450 | 0.689 ± 0.0170 | 0.320 ± 0.0501 | 0.358 ± 0.0410 | 0.328 ± 0.0218 |
| | Spline | 0.9997 ± 0.0004 | 0.999 ± 0.0005 | 0.999 ± 0.0003 | 0.016 ± 0.0049 | 0.017 ± 0.0053 | 0.017 ± 0.0052 | 0.690 ± 0.0434 | 0.649 ± 0.0476 | 0.6923 ± 0.0199 | 0.324 ± 0.0548 | 0.3733 ± 0.0507 | 0.327 ± 0.0105 |
| | VS | 0.9998 ± 0.0001 | 0.999 ± 0.0003 | 0.999 ± 0.0002 | 0.011 ± 0.0053 | 0.014 ± 0.0034 | 0.012 ± 0.0016 | 0.682 ± 0.0561 | 0.650 ± 0.0546 | 0.682 ± 0.0251 | 0.325 ± 0.0568 | 0.371 ± 0.0591 | 0.337 ± 0.0264 |
| GOODSST2,length,concept | Dirichlet | 0.938 ± 0.0019 | 0.939 ± 0.0056 | 0.942 ± 0.00180 | 0.189 ± 0.01989 | 0.165 ± 0.0179 | 0.187 ± 0.0256 | 0.694 ± 0.0193 | 0.693 ± 0.0020 | 0.687 ± 0.0027 | 0.146 ± 0.0196 | 0.133 ± 0.015 | 0.169 ± 0.0168 |
| | ETS | 0.938 ± 0.0020 | 0.939 ± 0.0060 | 0.941 ± 0.0017 | 0.389 ± 0.0018 | 0.390 ± 0.0022 | 0.393 ± 0.0007 | 0.6940 ± 0.0193 | 0.692 ± 0.0019 | 0.687 ± 0.0034 | 0.214 ± 0.0098 | 0.216 ± 0.0033 | 0.220 ± 0.0057 |
| | IRM | 0.939 ± 0.0016 | 0.939 ± 0.0058 | 0.941 ± 0.0018 | 0.326 ± 0.0011 | 0.326 ± 0.0013 | 0.327 ± 0.0017 | 0.693 ± 0.0185 | 0.692 ± 0.0026 | 0.685 ± 0.0026 | 0.240 ± 0.0017 | 0.232 ± 0.0050 | 0.242 ± 0.0053 |
| | OrderInvariant | 0.938 ± 0.0020 | 0.939 ± 0.0060 | 0.941 ± 0.0022 | 0.314 ± 0.0014 | 0.315 ± 0.0029 | 0.315 ± 0.0012 | 0.6940 ± 0.0193 | 0.692 ± 0.0019 | 0.687 ± 0.0033 | 0.224 ± 0.0010 | 0.222 ± 0.0030 | 0.223 ± 0.0054 |
| | Spline | 0.938 ± 0.0026 | 0.938 ± 0.0044 | 0.941 ± 0.0010 | 0.329 ± 0.0021 | 0.329 ± 0.0019 | 0.328 ± 0.0012 | 0.692 ± 0.0190 | 0.692 ± 0.0022 | 0.687 ± 0.0035 | 0.234 ± 0.0052 | 0.231 ± 0.0044 | 0.243 ± 0.0034 |
| | VS | 0.938 ± 0.0027 | 0.939 ± 0.0057 | 0.941 ± 0.0018 | 0.290 ± 0.2099 | 0.484 ± 0.0008 | 0.487 ± 0.0007 | 0.693 ± 0.0184 | 0.693 ± 0.0018 | 0.687 ± 0.0031 | 0.331 ± 0.0484 | 0.375 ± 0.0022 | 0.382 ± 0.0048 |
| GOODSST2,length,covariate | Dirichlet | 0.896 ± 0.0029 | 0.893 ± 0.0009 | 0.895 ± 0.00095 | 0.196 ± 0.0155 | 0.172 ± 0.0091 | 0.1797 ± 0.0109 | 0.825 ± 0.0037 | 0.827 ± 0.0066 | 0.805 ± 0.0150 | 0.163 ± 0.0198 | 0.141 ± 0.0087 | 0.142 ± 0.0122 |
| | ETS | 0.8966 ± 0.0023 | 0.894 ± 0.0011 | 0.894 ± 0.0006 | 0.357 ± 0.0013 | 0.359 ± 0.0004 | 0.362 ± 0.0019 | 0.826 ± 0.0036 | 0.828 ± 0.0065 | 0.806 ± 0.0117 | 0.309 ± 0.0050 | 0.314 ± 0.0076 | 0.300 ± 0.0070 |
| | IRM | 0.895 ± 0.0019 | 0.893 ± 0.0003 | 0.894 ± 0.0007 | 0.307 ± 0.0004 | 0.307 ± 0.0003 | 0.306 ± 0.0020 | 0.826 ± 0.0040 | 0.826 ± 0.0036 | 0.809 ± 0.0152 | 0.276 ± 0.0046 | 0.277 ± 0.0061 | 0.265 ± 0.0078 |
| | OrderInvariant | 0.896 ± 0.0023 | 0.894 ± 0.0011 | 0.894 ± 0.0008 | 0.288 ± 0.0008 | 0.285 ± 0.0008 | 0.284 ± 0.0013 | 0.826 ± 0.0036 | 0.828 ± 0.0065 | 0.806 ± 0.0106 | 0.244 ± 0.0022 | 0.241 ± 0.0037 | 0.225 ± 0.0054 |
| | Spline | 0.894 ± 0.0016 | 0.890 ± 0.0009 | 0.892 ± 0.0040 | 0.309 ± 0.0024 | 0.307 ± 0.0009 | 0.307 ± 0.0022 | 0.822 ± 0.0026 | 0.822 ± 0.0092 | 0.801 ± 0.0110 | 0.275 ± 0.0043 | 0.276 ± 0.0063 | 0.264 ± 0.0063 |
| | VS | 0.8963 ± 0.0028 | 0.893 ± 0.0008 | 0.894 ± 0.0007 | 0.291 ± 0.1833 | 0.460 ± 0.0011 | 0.465 ± 0.0010 | 0.821 ± 0.0053 | 0.827 ± 0.0071 | 0.806 ± 0.0119 | 0.299 ± 0.1395 | 0.431 ± 0.0061 | 0.429 ± 0.0054 |

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

| G-ΔUQ(L2) | 2360110 | 1633203 | 2091001 | | | |
| G-ΔUQ(L3) | 2360110 | | | | | |
| G-ΔUQ(L4) | 2360110 | | | | | |
| G-ΔUQ(Readout) | 2004310 | 912303 | 1732501 | | | |

### A.15 EXPANDED DISCUSSION ON ANCHORING DESIGN CHOICES

Below, we expand upon some of the design choices for the proposed anchoring strategies.

**When performing node featuring anchoring, how does fitting a Gaussian distribution to the input node features help manage the combinatorial stochasticity induced by message passing?**

Without loss of generality, consider a node classification setting, where every sample is assigned a unique anchor. Then, due to message passing, after $l$ hops, a given node's representation will have aggregated information from its $l$ hop neighborhood. However, since each node in this neighborhood has a unique anchor, we see that any given node's representation is not only stochastic due to its own anchor but also that of its neighbors. For example, if any of its neighbors are assigned a different anchor, then the given node's representation will change, even if its own anchor did not. Since this behavior holds true for all nodes and each of their respective neighborhoods, we loosely refer to this phenomenon having combinatorial complexity, as effectively marginalizing out the anchoring distribution would require handling any and all changes to all $l$-hop neighbors. In contrast, when performing anchored image classification, the representation of a sample is only dependent on its unique, corresponding anchor, and is not influenced by the anchors of other samples. To this end, using the fitted Gaussian distribution helps manage this complexity, since changes to the anchors of a node's $l$-hop neighborhood are simpler to model as they require only learning to marginalize out a Gaussian distribution (instead of the training distribution). Indeed, for example, if we were to assume simplified model where message passing only summed node neighbors, the anchoring distribution would remain gaussian after $l$ rounds of message passing since the sum of gaussians is still gaussian (the exact parameters of the distribution would depend on the normalization used however).