# OpenReview forum: "Accurate and Scalable Estimation of Epistemic Uncertainty for Graph Neural Networks"
_ICLR.cc/2024/Conference — ICLR 2024 poster_

### Official Review · Reviewer_Cjvm · 2023-10-31

**Soundness:** 3 good
**Presentation:** 3 good
**Contribution:** 2 fair
**Rating:** 6
**Confidence:** 3

**Summary:**

This paper extends the ∆-UQ model to graph learning to enhance intrinsic GNN uncertainty estimates. It proposes several training techniques to introduce partial stochasticity for node and graph-level classification tasks. Extensive experimental results validate the effectiveness of the proposed method. Interestingly, the graph-level training method can be applied to pre-trained graph models.

**Strengths:**

1. Extensive experimental results on various datasets and setups demonstrate the effectiveness of the proposed method.

2. The introduced method is fairly simple and easy to understand. It seems that it can be applied to many existing models in a plug-and-play manner, making it flexible and extendable.

3. The paper also explores the potential of applying this method to pre-trained models, which is an interesting aspect.

**Weaknesses:**

1. Some notations and definitions in the paper are unclear, such as the symbols used for samples and distributions, and the shape of tensors during concatenation.

2. The experiments conducted focus solely on classification tasks. Consideration for regression tasks seems to be missing.

3. There are limited performance gains observed with the use of pre-trained models.

These points are further detailed below.

**Questions:**

1. In Section 2 (Preliminaries), the notation $\\mathbf{X}$ is somewhat unclear. It seems to represent some distributions, but there is also the concatenation $[\\mathbf{X} - \\mathbf{C}, \\mathbf{C}]$, which is described as "channel-wise concatenating two images". Please clarify the notations regarding distributions, samples and training (test) sets.

2. On page 3 (Section 3.1 Node Feature Anchoring), where are $\\mathbf{A}_c$ and $Y_c$ defined?

3. On page 4, "the anchor/query node feature pair" is defined as $[\\mathbf{X}_i − \\mathbf{C}|| \\mathbf{X}_i]$, where $\\mathbf{C}^{N \\times d} \\sim \\mathcal{N} (\\mu, \\sigma^2)$. How are the shapes of $\\mathbf{X}_i$ and $\\mathbf{C}$ aligned?

4. For the node classification task, anchoring is sampled from a learned Gaussian. The notations used seem unclear to the reviewer. Could you elaborate on the learning procedures, the loss function used, and the dimensions of the Gaussians? During inference, stochasticity arises from sampling from such Gaussians. Is it possible to display qualitative results of the learned Gaussians, including details such as variance and mean?


5. a) For the graph classification task, there seems to be no extra learnable component to create the anchors, but the subsequent MPNN layers are modified to accept features of $d_r \\times 2$ dimensions. How does this subtle change affect the number of trainable parameters?\
b) The stochasticity appears to arise from the random shuffle of the node features across the entire batch, and the subsequent selection of anchoring $\\mathbf{c}^{1 \\times d}$. Does creating anchors from a Gaussian distribution work in this case?\
c) Conversely, is it feasible to use intermediate representations to construct anchors for the node classification task? Technically, it seems like these could substitute for the learned Gaussian.\
d) Furthermore, when randomly shuffling the node features across the entire batch (potentially involving cross-graph shuffling), are there observable impacts attributed to the size of the mini-batch?

6. Can this method be generalized to graph/node/edge level regression tasks?

7. The design of shuffling across batches in this paper appears conceptually somewhat similar to the mixup method [1] in the feature space. A comparison between them would be interesting.

8. In Fig. 6, Tables 4/5, the effects of applying G-∆UQ on top of the pretrained model (GPS) seem very marginal.

[1] mixup: Beyond Empirical Risk Minimization, ICLR 2018.

---

> ### Author Response · Authors · 2023-11-20
> **Response to Reviewer Cjvm (Part 1)**
>
> Thank you for thoughtful comments and questions! We are pleased you found our proposed method “flexible and extendable” as well as “effective” and our experimental evaluation extensive. We address comments/questions below.
>
> > ***(W1/Q1-3) Some notations and definitions in the paper are unclear, such as the symbols used for samples and distributions, and the shape of tensors during concatenation.***
>
> Thanks for the comments. We’ve updated the main paper with better notation. Please see sections 2 and 3 for the updated notations, highlighted in blue. We summarize our anchoring strategies with the updated notation below.
>
> Let a graph classification GNN consisting of  $( 1 \dots \ell)$ message passing layers $(\text{MPNN})$, a graph-level readout function ($\text{READOUT}$), and classifier head ($\text{MLP}$) is defined as follows: $\mathbf{X}^{\ell} = \text{MPNN}^{\ell} \left(\mathbf{X}^{\ell - 1}, \mathbf{A} \right)$, $\mathcal{G}=  \text{READOUT} \left(\mathbf{X}^{\ell} \right)$, and $\hat{Y} =\text{MLP}\left( \mathcal{G} \right)$ where $\mathbf{X}^{\ell} \in \mathbb{R}^{N  \times d_{\ell}}$ is the intermediate node representation at layer $\ell$, $\mathcal{G} \in\mathbb{R}^{1 \times d_{\ell}}$ is the graph representation, and $\hat{Y} \in \{0,1\}^q$ is the predicted label. When performing node classification, the $\text{READOUT}$ layer is not included, and instead node-level predictions are outputted: $\hat{Y}=\text{MLP}\left( \mathbf{X}^{\ell}\right)$. Then, we can define the anchoring strategies as follows. *(Note, we show an abrdiged version here, please see the text for full details.)*
>
> - **Node feature anchoring:** $\mathbf{X}^{1} = \text{MPNN}^{1}(\mathbf{X}^0 - \mathbf{C} || \mathbf{C}, \mathbf{A})$, where the dimension of the $\text{MPNN}^1$ has been changed to accommodate $d \times 2$ dimensional input.
>
> - **Hidden layer anchoring:** Let $r > 2$ be the layer at which anchoring is applied. Then, $\mathbf{X}^{r-1} = \text{MPNN}^{r-1}(\mathbf{X}^{r-2}, \mathbf{A})$ and $\mathbf{C}=\text{SHUFFLE}(\mathbf{X}^{r-1}, dim=0)$ , where  $\mathbf{X}^{r} = \text{MPNN}^{r}(\mathbf{X}^{r-1} - \mathbf{C} || \mathbf{C}), \mathbf{A})$ is the anchored representation. Here, $\text{MPNN}^r$ ‘s input dimension is changed to $d_{r-1} \times 2$
>
> - **READOUT anchoring:** $\mathbf{G} = \text{READOUT}(\text{MPNN}^{1 \ldots \ell}(\mathbf{X}^0, A),BatchVec)$, where $\mathbf{C}=\text{SHUFFLE}(\mathbf{G}, dim=0)$ and the anchored representation is defined as  $\mathbf{Y} = \text{MLP}(\mathbf{G} - \mathbf{C} || \mathbf{C})$.
>
> > ***(W2/Q6) The experiments conducted focus solely on classification tasks. Can this method be generalized to graph/node/edge level regression tasks?***
>
> Applying G-ΔUQ to regression tasks is a natural extension and our model formulation does conceptually permit it. However, since regression tasks require different definitions of uncertainty to account for real-valued outputs, we suspect that our inference procedure and calibration strategy will need to be changed. This is definitely part of our future work; thanks for the suggestion! Lastly, we briefly note that while we focus on graph and node classification tasks in the paper due to the readily available distribution shift benchmarks, G-ΔUQ can in fact be easily extended to link prediction by using a similar formulation to node feature anchoring. The main difference would be that both the “head” and “tail” nodes of the predicted link would be anchored, prior to being fed into the binary link-predictor head.

---

> ### Author Response · Authors · 2023-11-20
> **Response to Reviewer Cjvm (Part 2)**
>
> > ***(W3/Q8) There are limited performance gains observed with the use of pre-trained models.***
>
> Thanks for the feedback. We’ve expanded our discussion in the paper to make the benefits of the pre-trained G-ΔUQ clearer, and discuss key takeaways here.
>
> - **Comments about Fig 6.** With the exception of the GOODMotif (basis) dataset, *pretrained G-ΔUQ improves the OOD ECE over both the vanilla model and end-to-end G-ΔUQ at comparable or improved OOD accuracy (6/8 datasets)*. Furthermore, pretrained G-ΔUQ also improves the *ID ECE on all but the GOODMotif (size) datasets (6/8)*, where it performs comparably to the vanilla model, and maintains the ID accuracy.  This demonstrates that pretrained G-ΔUQ does offer some performance advantages over end-to-end G-ΔUQ, though we agree the absolute benefits may appear a bit small.
>
>     However, this assessment did not take into account the computational benefits and the flexibility of being able to use a pre-trained backbone. Indeed, only an anchored classifier is trained for the pretraining setting; this substantially reduces training time relative to end-to-end G-ΔUQ, *where training time is ~40% less than vanilla or node-feature G-ΔUQ training (We’ve added new runtime results in App., Table 17 to illustrate this.)* Moreover, as larger, pretrained models become widely available, pretrained G-ΔUQ offers an effective, lightweight solution for improving uncertainty estimates (as evidenced by the improvements over the Vanilla model in Fig 6.). We highlight that this is a unique benefit of our stochastic centering formulation.
>
> - **Comments about GPS:**  Tables 5 and 6 contain several axes (Architecture, G-ΔUQ vs. Vanilla, LPE vs w/o LPE, Ensemble vs. Single Model, Pretrained vs. End-to-End) that makes it difficult to immediately parse the benefits of Pretraining with respect to GPS. Thus, *we have added a new table (Tables 6/7 in the updated appendix) where we pull out the comparable rows and see that pretrained G-ΔUQ clearly improves the calibration at comparable accuracy relative to the Vanilla GPS model, especially as the distribution shift becomes larger.* We’ve updated the paper text to better discuss these axes.
>
> > ***(Q4.a) Could you elaborate on the learning procedures, the loss function used, and the dimensions of the Gaussians?***
>
> Thanks for the comments! We *updated the paper with Pytorch-Style pseudo-code (Figures 1,2 and Appendix A.1) to more intuitively explain the learning procedures, and clarified the notations to make the text easier to follow (Sec 3).*
>
> In brief, G-∆UQ does not change the loss function when training. Thus, we use the CrossEntropyLoss to train both anchored and vanilla models. Anchors are uniquely re-sampled for training samples at each epoch (each sample gets a new anchor).  At inference, we average predictions over $K$ random anchors, and use the prediction variance over the $K$-predictions to estimate uncertainty (as shown in Sec 2.). When performing node feature anchoring, the dimension of the Gaussian is determined by the dataset’s node feature dimension. We’ve updated Table 9 in the appendix to include this information.
>
> > ***(Q4.b) During inference, stochasticity arises from sampling from such Gaussians. Is it possible to display qualitative results of the learned Gaussians, including details such as variance and mean?***
>
>
> We wish to clarify that the Gaussians obtained from the node features are fixed at initialization and are not updated (or learned) during training. For example, when performing node classification, let $\mathbf{X}^{N \times d}$ be the set of training node features. We compute the mean, $\mu$, and standard deviation, $\sigma$ for each dimension $d$ independently, and then initialize a $d$ dimensional Normal with mean $= [\mu_1 \dots \mu _d]$ and scale $= [\sigma_1 \dots \sigma_d]$.  We can then straight-forwardly sample from this distribution when training to obtain anchors.
>
> For graph classification, this procedure is slightly modified since each sample (graph) has its own set of node features. Therefore, we first create the set of node features in the dataset by aggregating node features from individual graph samples. In detail, let $\mathcal{D} = {\mathcal{G}_1 \times \mathcal{G}_t}$ be the training dataset, where each $\mathcal{G}_i$ has $\mathbf{X}^{N \times d}$ node features. The aggregated set of node features would be of dimension of $(\sum^t N_i \times d)$. Given this set, the same procedure as above is used to calculate the normal.
>
> Since the dimension of the node feature anchoring Gaussians can vary per dataset, we provide summary statistics computed over the per-dimension means and standard deviations, and have included them in the appendix (Table 18).

---

> ### Author Response · Authors · 2023-11-20
> **Response to Reviewer Cjvm (Part 3)**
>
> > ***(Q5.a) For the graph classification task, there seems to be no extra learnable component to create the anchors, but the subsequent MPNN layers are modified to accept features of dimensions. How does this affect the number of trainable parameters?***
>
> You are correct that there are no extra learnable components for creating the anchors. Only the dimension of the layer at which anchoring is applied is modified, namely by doubling its input dimension to accommodate the anchor and residual sample. For example, if anchoring is applied to a 4-layer GNN at the 2nd MPNN layer, we modify MPNN_2 ’s input dimension, but $\text{MPNN}_{1,3,4}$ and $\text{MPNN}_2$ output dimension remain the same.
>
> The exact change to the number of parameters depends on the underlying architecture, the selected intermediate layer, and input dimension . For example, changing the dimension of a GIN layer will require changing the dimension of its MLP, while READOUT anchoring only changes the dimension of the classifier (a single linear layer in our experiments).  *We include this information in the appendix (Table 19). Note that while the number of parameters increases, the task has also increased a bit in difficulty, so we do not consider this increase as affecting the model’s capacity.*
>
> > ***(Q5.b) The stochasticity appears to arise from the random shuffle of the node features across the entire batch, and the subsequent selection of anchoring. Does creating anchors from a Gaussian distribution work in this case?***
>
> It is indeed conceptually possible to create a Gaussian distribution over the intermediate representations that can be used to perform hidden representation anchoring on graph classification tasks. *We’ve added some preliminary results to Tables 15 in the appendix evaluating this formulation.* We find that the performance is generally comparable with respect to our original shuffling-based formulation and indeed, there are some improvements. Performing automatic anchoring strategy selection is an interesting direction of future research.
>
> > ***(Q5.c) Conversely, is it feasible to use intermediate representations to construct anchors for the node classification task? Technically, it seems like these could substitute for the learned Gaussian.***
>
> It is indeed feasible to use intermediate representations to construct anchors when performing node classification in a manner similar to Sec. 3.2. *Below and in the updated appendix, we provide some preliminary results demonstrating this is possible and that G-∆UQ is able to provide partially stochastic models for node classification.* We note that we focused on using node featuring anchoring for node classifications tasks as this formulation is most closely related to the original stochastic centering formulation [1].
> In brief, [1], which focuses on image classification tasks, uses input-space transformations (image-image subtraction, and channel wise concatenation) to create anchored representations and fully stochastic models, where each anchored image constitutes a single sample. Similarly, with node classification, each node can be considered a “sample” since the loss is computed with respect to its predicted label. Then, by using node feature anchoring, we can also perform input-space transformations to create per-sample anchored representations and fully stochastic GNNs.
>
> [1] Single Model Uncertainty Estimation via Stochastic Data Centering, Thiagarajan et al. NeurIPS'22
>
> > ***(Q5.d) Furthermore, when randomly shuffling the node features across the entire batch (potentially involving cross-graph shuffling), are there observable impacts attributed to the size of the mini-batch?***
>
> We used the same batch size as indicated by the original benchmark code-bases (GOODBenchmark,GPS, SizeShiftReg) to ensure fair comparison with a well-tuned vanilla baseline. *We do not anticipate negative impacts caused by cross-graph shuffling or observable impacts attributed to the size of mini-batch.* Indeed, in our present formulation, cross-graph shuffling is already occurring as we use PytorchGeometric style mini-batching (which stacks the node-features of different graphs). Reducing the batch-size would lead to less cross-graph shuffling. However, there is also increased risk of training instability due to noisier gradient updates, and cross-graph shuffling is not in fact harmful to G-ΔUQ’s performance.

---

> ### Author Response · Authors · 2023-11-20
> **Response to Reviewer Cjvm (Part 4)**
>
> > ***(Q7) The design of shuffling across batches in this paper appears conceptually somewhat similar to the mixup method [1] in the feature space. A comparison between them would be interesting.***
>
> Mixup, which trains models on convex combinations of pairs of samples and labels, regularize neural networks to prefer simple linear behavior, which can help generalization, robustness and create smoother uncertainty estimates. While G-ΔUQ also conceptually combines pairs of samples, our overarching goals and formulation are fundamentally different. Importantly, mixup uses label information, whereas G-ΔUQ is only an input transformation that does not use the label.
>
> Our objective is to provide reliable uncertainty estimates that can then be used in various uncertainty based tasks. *To this end, we use anchoring as a mechanism for sampling different functional hypotheses from a single model.* Sampling different hypotheses allows us to obtain reliable epistemic uncertainty estimates that can then be used to improve confidence estimates on a variety of downstream uncertainty based tasks (Calibration, OOD Detection, Generalization Gap Prediction). *In contrast, mixup is a data augmentation strategy, so it does not offer a mechanism for sampling different hypotheses at inference time, and can only regularize the behavior of a single functional hypothesis.* We note that it is conceptually possible to combine G-ΔUQ and mixup to support both the sampling of different functions for obtaining epistemic uncertainty estimates, while also ensuring that these hypotheses exhibit more linear behavior. We thank the reviewer for their suggestion and will conduct this experiment.
>
> [1] mixup: Beyond Empirical Risk Minimization, Zhang et al. ICLR 2018.

---

> > ### Author Response · Authors · 2023-11-22
> > **Thanks for the Feedback/Request for Response**
> >
> > Thank you for your helpful comments! We have [modified the paper](https://openreview.net/forum?id=ZL6yd6N1S2&noteId=whZnC2XK4J) to incorporate your feedback and have provided responses to your questions above.
> >
> > As the discussion period is closing, we wanted to check if you had additional questions that we can address before it ends. If you find our responses satisfactory, we humbly request that you raise the score and to help champion our paper.

---

> > > ### Comment · Reviewer_Cjvm · 2023-11-23
> > >
> > > Thank you for your comprehensive response. The pseudocode explanation was especially beneficial in helping me understand the implementation process, and I found it very insightful. I would like to adjust my rating to "6: marginally above the acceptance threshold".

---

### Official Review · Reviewer_bgJx · 2023-10-31

**Soundness:** 2 fair
**Presentation:** 2 fair
**Contribution:** 2 fair
**Rating:** 5
**Confidence:** 3

**Summary:**

The paper proposed a new training framework inspired by the stochastic anchored training in computer vision. In the framework two (partial) stochastic anchoring techniques (node feature anchoring and hidden layer anchoring) are designed for sake of the better uncertainty estimation and calibration in GNNs. The experiments on node classification and graph classificaiton validate the effectiveness of the framework.

**Strengths:**

1. The idea that introducing the anchoring training in GNNs is new and interesting.

2. The paper conducted comprehensive evalutions on the calibration of GNNs under different settings. Specifically, the evaluation on calibration of GNNs under distribution shift is new and significant. The experimental results show that on most tasks the proposed method can achieve lower calibration error.

**Weaknesses:**

1.Since the paper focused on calibration and uncertainty estimation of GNNs. The concepts of calibration and uncertainty estimation of GNNs should be introduced in the paper.

2.The paper doesn't provide sufficient discussion or theories to justify the methods provided in the paper. For instance,  in node feature anchoring why authors sample the anchors from the Gaussian Distribution? How to determine the value of $u$ and $\sigma$?

3.The paper claimed that the framework can improve the uncertainty estimation. However, how the method can improve the uncertainty estimation is still unclear.

4.The organization of the experiment part is messy. More setup details should be clarified.

5.Some typos in the paper and the title of Table 2 is missing.

**Questions:**

See weakness

---

> ### Author Response · Authors · 2023-11-20
> **Response to Reviewer bgJx (Part 1)**
>
> Thank you for your helpful comments and questions! We are pleased you found the proposed method “new and interesting” and our evaluation “new and significant.” Comments and questions are addressed below.
>
> >***(W1) Since the paper focused on calibration and uncertainty estimation of GNNs. The concepts of calibration and uncertainty estimation of GNNs should be introduced in the paper.***
>
> Thanks for the feedback. We have *updated the introduction* so uncertainty estimation and calibration are presented earlier to the reader. (We note that Appendix A.9 also contains information on different post-hoc calibration methods and defines the expected calibration error.)
>
> > ***(W2.a). The paper doesn't provide sufficient discussion or theories to justify the methods provided in the paper.***
>
> Thanks for the questions. We have (i) added *additional discussion in Sec 3*. to explain the goals of the anchoring distribution/strategy to provide context for our design decisions, (ii) improved the description of the proposed strategies, and (iii) added pytorch-style pseudocode *to improve clarity (Fig 1, Fig 2, Fig 5).*
>
> Conceptually, anchoring provides a mechanism for sampling multiple hypotheses from a single model such that their variability can be aggregated to provide reliable epistemic uncertainty estimates. *Our introduced anchoring strategies are specifically designed to (i) handle discrete, variable-sized inputs and (ii) induce diverse hypotheses with controlled stochasticity.* In particular, node feature anchoring is proposed to provide a fully stochastic formulation that is compatible with graph data and message passing neural networks (MPNNs), while Hidden Layer and Readout Anchoring not only support graph data/MPNNs, but also induce partially stochastic models that are able to provide functionally diverse hypotheses. Empirically, we find that balancing functional diversity and stochasticity leads to better performance.
>
> **Hidden Layer and Readout Anchoring:** While the success of model ensembles has been attributed to their ability to leverage diverse hypotheses, it was recently demonstrated that partial stochasticity can perform comparably to fully stochastic networks at significantly less cost [1]. This suggests that in addition to the *amount of diversity*, the *effective or functional diversity* of the sampled hypotheses is also critical for performance. Since hidden layer and readout anchoring are performed after multiple rounds of message passing, the corresponding representations capture both node-feature and topological information. *By defining the residual transformation and anchoring distribution with respect to these intermediate representations, we are able to sample hypotheses from a latent space containing higher-level, disentangled semantics.* Thus, even though the layers prior to anchoring are deterministic, the sampled hypotheses are functionally diverse (e.g., they sample decision rules that rely upon different higher-order features). We empirically find, through extensive experiments across datasets, shifts, tasks and architectures, that this improves performance. Note that hidden layer and readout anchoring are, by design, able to handle discrete and variable-sized input. However, to support fully stochastic networks, we proposed *node feature anchoring.*
>
> [1] Do Bayesian Neural Networks Need To Be Fully Stochastic? Sharma et al. AISTATS'23
>
> > ***(W2.b1) How are the values of the node feature anchoring gaussian distribution determined?***
>
> *Please see Fig 2 for pseudocode explaining this process.* In brief, when performing node classification, let $\mathbf{X}^{N \times d}$ be the set of training node features. We compute the mean, $\mu$, and standard deviation, $\sigma$ for each dimension $d$ independently, and then initialize a $d$ dimensional Normal with mean $= [\mu_1 \dots \mu _d]$ and scale $= [\sigma_1 \dots \sigma_d]$.  We can then straight-forwardly sample from this distribution to obtain anchors.
>
> For graph classification, this procedure is slightly modified since each sample (graph) has its own set of node features. Therefore, we first create the set of node features in the dataset by aggregating node features from individual graph samples. In detail, let $\mathcal{D} = {\mathcal{G}_1 \dots \mathcal{G}_t}$ be the training dataset,  where each $\mathcal{G}_i$ has $\mathbf{X}^{N \times d}$ node features. The aggregated set of node features would be of dimension of $((\sum^t N_i) \times d))$. Given this set, the same procedure as above is used to calculate the normal.
>
> *Since the dimension of the node feature anchoring Gaussians can vary per dataset, we provide summary statistics over the per dimension means and standard deviations, and have included them in appendix A.14.*

---

> ### Author Response · Authors · 2023-11-20
> **Response to Reviewer bgJx (Part 2)**
>
> > ***(W2.b2) Why is a Gaussian distribution used?***
>
> [1] motivated stochastic centering through the lens of neural tangent kernel (NTK), where they showed that constant shifts to the input $(\mathbf{X}-\mathbf{C})$ induce analogous shifts to the effective NTK, i.e. different hypotheses. They then demonstrated that a single (anchored) model can replicate this behavior if the input bias (e.g. the anchor, $\mathbf{C}$)  is treated as a random variable during training (see Fig. 1) and marginalized out during inference. While [1] used the training samples as anchors and computed $(\mathbf{X}-\mathbf{C})$ for image classification, this is not applicable to graph datasets Subtracting two adjacency matrices will introduce connectivity artifacts and subtracting input node features is also not advisable because features are not IID and may belong to different sized graphs. *By using the proposed Gaussian anchoring distribution, we are able to sample anchors for arbitrarily sized graphs, without introducing connectivity artifacts or drastically deviating from the original training landscape (evidenced by similar ID accuracies).*
>
> **Controlling Stochasticity.** Without loss of generality, consider a node classification setting, where every sample is assigned a unique anchor. Then, due to message passing, after $\ell$ hops, a given node’s representation will have aggregated information from its $\ell$ hop neighborhood. *As a result, any given node’s representation is stochastic not only due to its own anchor but also that of its neighbors. Since this behavior holds true for all nodes and each of their respective neighborhoods, we loosely refer to this phenomenon as having combinatorial complexity*, as effectively marginalizing out the anchoring distribution would require handling any and all changes to all $\ell$-hop neighbors. In contrast, when performing anchored image classification, the representation of a sample is only dependent on its own anchor, and is not influenced by the other samples. To this end, using the fitted Gaussian distribution helps manage this complexity, *since changes to the anchors of a node’s $\ell$-hop neighborhood are simpler to model as they require only learning to marginalize out a Gaussian distribution (instead of the training distribution).*
>
> [1] Single Model Uncertainty Estimation via Stochastic Data Centering, Thiagarajan et al. NeurIPS'22
>
> > ***(W3) The paper claimed that the framework can improve the uncertainty estimation. However, how the method can improve the uncertainty estimation is still unclear.***
>
> Thanks for the comment. Epistemic uncertainty, or reducible uncertainty arising from the lack of information regarding a model’s fit to data distribution, can be estimated by aggregating prediction variability over potential model hypotheses. Indeed, on OOD or challenging data, these hypotheses are more likely to diverge, indicating higher uncertainty; while on ID or easier samples, these hypotheses are more likely to agree, indicating lower uncertainty. *Our proposed anchoring strategies enable us to sample multiple, diverse hypotheses from a single model. By computing the mean prediction and variance over $K$ different anchors, we can estimate the epistemic uncertainty around a particular sample.* We then use these estimates to rescale the predicted mean logits to reflect the expectation that a model should be relatively less (or more) confident when its uncertainty is high (or low) (see Sec 2. and Fig 1).  Using these rescaled logits leads to improvements across shifts (covariate, size and concept), datasets, and tasks (calibration, generalization gap prediction and OOD detection).
>
> > ***(W4) The organization of the experiment part is messy. More setup details should be clarified.***
>
> Thanks for the feedback. We will provide more set-up details in the expanded appendix and will open-source a code-base for replicating our experiments. (While a working anonymous github repo link has been provided; we will clean this before officially releasing). Further, since we are focused on evaluating the benefits of G-ΔUQ on various distribution shift benchmarks and tasks, we adhere to published experimental protocols to ensure fair comparison.
>
> > ***(W5) Some typos in the paper and the title of Table 2 is missing.***
>
> Thanks for the catch! We’ve updated the paper after some additional editing.

---

> ### Author Response · Authors · 2023-11-22
> **Thanks for the Feedback/Request for Response**
>
> Thank you for your helpful comments! We have [modified the paper](https://openreview.net/forum?id=ZL6yd6N1S2&noteId=whZnC2XK4J) to incorporate your feedback and have provided responses to your questions above.
>
> As the discussion period is closing, we wanted to check if you had additional questions that we can address before it ends. If you find our responses satisfactory, we humbly request that you raise the score and to help champion our paper.

---

### Official Review · Reviewer_b8ZS · 2023-11-02

**Soundness:** 3 good
**Presentation:** 3 good
**Contribution:** 3 good
**Rating:** 6
**Confidence:** 4

**Summary:**

The paper proposes a training framework for GNNs that is designed to improve the intrinsic uncertainty estimates. The adopted strategy is to adapt the principle of stochastic data centering to graph data. This involves introducing novel graph anchoring techniques. The paper demonstrates that the methodology can support partially stochastic GNNs. Experimental results in the paper suggest that the partial stochasticity is sufficient; it also has the advantage of providing a mechanism for incorporating pre-trained models. The paper reports experiments investigating the impact of covariate, concept and graph size shifts and demonstrates that the proposed technique leads to better calibrated GNNs, both for node and graph classification. Additional experiments illustrate how the approach performs for out-of-distribution detection and generalization gap estimation.

**Strengths:**

S1. The paper introduces a novel approach for improving the intrinsic uncertainty estimates of GNNs by translating the stochastic centering strategy to the graph domain. This is non-trivial, both for node and graph classification.

S2. The paper reports on experiments exploring (i) node classification under distribution shift (concept and covariate shifts); (ii) calibration under distribution shift for graph classification; (iii) how the approach impacts the calibration of more expressive models such as graph transformers. The experiments are thorough and examine multiple interesting questions.

S3. The experiments support the interesting observation that the network does not need to be fully stochastic in order to provide improved uncertainty estimates. This paves the way to combine the p

**Weaknesses:**

W1.	Some of the methods employed to translate stochastic centering to the graph domain appear somewhat heuristic, or are at least the text describing the methodology does not provide sufficient detail to perceive the design principles. For example, the node feature anchoring fits a Gaussian distribution, but there is no explanation as to why a Gaussian is selected and no discussion as to whether the mismatch between the fitted anchor distribution and the feature distribution has a negative impact or could be a concern. The text states that the introduction helps to “manage the combinatorial stochasticity induced by message passing” but it does not elaborate on this to explain why or how. For the graph anchoring, there is a random shuffling of the node features over the entire batch. There is no discussion of this design choice – it doesn’t seem obvious to me that this is the only thing one could choose to do (or the optimal).

W2.	Distilling the methodological contributions, we see that they involve: (i) node anchoring via fitting a Gaussian distribution and drawing an anchor from this fitted distribution; (ii) hidden layer anchoring by randomly shuffling the node features after the r-th layer. After these steps to construct appropriate anchors in the graph domain, there is effectively a standard application of the stochastic centering approach. The technical methodological contribution is thus not particularly substantial. On the other hand, the experiments are thorough and provide a good balancing contribution.

W3.	Some of the results are not presented in a particularly helpful manner and are not described or discussed in much detail. For example, the observations for node classification essentially boil down to “our method works better”. The table contains interesting elements such as the proposed method failing to improve (WebKB, CBAS – Concept) or substantially increasing (Cora) the ECE when combined with Dirichlet calibration. But there is no discussion of this. In general there is not a significant attempt to draw detailed conclusions from the obtained results – similar comments apply to Section 5.3 where again the conclusions are “both pretrained and end-to-end G-∆UQ outperform the vanilla model on 7/8 datasets” and “G-∆UQ variants improve OOD detection performance over the vanilla baseline on 6/8 datasets”. Insights beyond “works better” make a paper much stronger and more insightful.

There is room for improvement in some of the figures and the explanations of how they are being interpreted. Figure 2 is a particular case in point – L1, L2, L3, N/A are not clearly defined. The text states that “READOUT anchoring generally performs well” but does not explain how we should interpret the figure to come to this conclusion. It’s not obvious what “performs well” means – what is an acceptable deterioration in performance. The behaviour over datasets and architectures differs considerably and should be discussed.

In terms of assessing the variability of performance, the paper reports standard deviations over a few trials, but does not make any attempt to assess the statistical significance of the results or to specify confidence intervals on the reported means.

**Questions:**

Q1. Why does the proposed method appear to work relatively poorly (at least in terms of often making ECE worse and sometimes considerably worse) when used in conjunction with Dirichlet calibration in the node classification setting?
Q2. Figure 2 is challenging to understand. Insufficient detail is provided. What is the meaning of L1, L2, L3, N/A? What does it mean “generally performs well across datasets and architectures” – which bars am I comparing to draw this conclusion? The accuracy and ECE behaviour seems quite inconsistent across different datasets and architectures – sometimes increasing, sometimes decreasing, sometimes going up then down. How do I determine when something “performs well”?
Q3. “We emphasize that this distribution is only used for anchoring and does not assume that the dataset’s node features are normally distributed.” This sentence seems to imply that it does not matter if the anchoring distribution matches the data distribution or not. But then why fit the Gaussian distribution to the training data? The fitting procedure implies that there is a need to have an anchor distribution that matches (at least to some degree) the feature distribution. So how close does it need to be to a Gaussian distribution?

---

> ### Author Response · Authors · 2023-11-20
> **Response to Reviewer: b8ZS  (Part 1)**
>
> Thank you for the  thoughtful comments! We appreciate that you found the proposed approach “novel” and “non-trivial” and our experiments to be “thorough and examine multiple interesting questions.” Comments/questions are answered below.
>
> > ***(W1) The text describing the methodology does not provide sufficient detail to perceive the design principles.***
>
> We’ve expanded and moved some of our experiments to the appendix so that we can discuss this in more detail. In particular, we’ve *added some additional discussion to Sec 3 and Sec 2* that conceptually explains the goals of the anchoring and strategy for distribution design to provide context for our design decisions. *We’ve also updated the appendix to include Pytorch-style pseudo code (App.1) to more clearly explain the proposed variants.*
>
> > ***(W1.1/Q3.1) Is it problematic if the fitted anchor distribution and the feature distribution do not match?***
>
> The anchoring distribution is used to sample different functional hypotheses from the model, but does not directly carry information that can be used to learn features. Indeed, the anchored representation $[\mathbf{X}-\mathbf{C} ||\mathbf{C}]$ contains both the residual $(\mathbf{X}-\mathbf{C})$ and the anchor ($\mathbf{C}$) so information is not in fact lost during stochastically centering, as the original sample can be recovered as ($\mathbf{X}- \mathbf{C} + \mathbf{C} = \mathbf{X}$). Therefore, **mismatch between the fitted anchor distribution and the feature distribution is not a major concern**. The bigger concern is if the anchoring distribution leads to an untenable amount of stochasticity where the model is in fact unable to effectively marginalize out the anchoring distribution when training. We elaborate on this below, and note this is partially why we proposed the Gaussian feature distribution. *Lastly, to provide some empirical evidence that potential mismatch is not a concern, we’ve added additional results to the appendix which plot the true distribution and fitted distribution. (App. A.14).* Despite the mismatch, G-ΔUQ still outperforms the vanilla model (Fig. 9, Fig. 10).

---

> ### Author Response · Authors · 2023-11-20
> **Response to Reviewer: b8ZS (part 2)**
>
> > ***(W1.2/Q3.2/Q3.3) Why is a Gaussian distribution selected when performing node featuring anchoring and how does this help manage the combinatorial stochasticity induced by message passing?***
>
> As discussed above, our central objective is to ensure that anchoring distribution induces a useful amount of stochasticity and, on the practical side, is able to handle graph inputs. *By using the fitted distribution, we are able to efficiently sample anchors for (1) arbitrarily-sized, discrete inputs and (2) control the exploding stochasticity that arises from message passing.*
>
> **Anchoring with graph data.** [1] motivated stochastic centering through the lens of neural tangent kernel (NTK), where they showed that constant shifts to the input $(\mathbf{X}-\mathbf{C})$ induce analogous shifts to the effective NTK, i.e. different hypotheses. They then demonstrated that a single (anchored) model can replicate this behavior if the input bias (e.g. the anchor, $\mathbf{C}$)  is treated as a random variable during training (see Fig. 1) and marginalized out during inference. While [1] used the training samples as anchors and computed $(\mathbf{X}-\mathbf{C})$ for image classification, this is not applicable to graph datasets. Subtracting two adjacency matrices will introduce connectivity artifacts and subtracting input node features is also not advisable because features are not IID. *By using the proposed Gaussian anchoring distribution, we are able to sample anchors for arbitrarily sized graphs, without introducing connectivity artifacts or drastically deviating from the original training landscape (evidenced by similar ID accuracies).*
>
> **Controlling Stochasticity.** Without loss of generality, consider a node classification setting, where every sample is assigned a unique anchor. Then, due to message passing, after $\ell$ hops, a given node’s representation will have aggregated information from its $\ell$ hop neighborhood. As a result, any given node’s representation is stochastic not only due to its own anchor but also that of its neighbors. Since this behavior holds true for all nodes and each of their respective neighborhoods, we loosely refer to this phenomenon as having combinatorial complexity, as effectively marginalizing out the anchoring distribution would require handling any and all changes to all $\ell$-hop neighbors. In contrast, when performing anchored image classification, the representation of a sample is only dependent on its own anchor, and is not influenced by the other samples. *Using the fitted Gaussian distribution helps manage this complexity, since changes to the anchors of a node’s $\ell$-hop neighborhood are simpler to model as they require only learning to marginalize out a Gaussian distribution (instead of the training distribution).* For example, if we were to assume a simplified message passing that only summed node neighbors, the anchoring distribution would remain gaussian after $\ell$ rounds of message passing since the sum of gaussians is still gaussian (the exact parameters of the distribution would depend on the normalization used however).
>
> [1] Single Model Uncertainty Estimation via Stochastic Data Centering, Thiagarajan et al. NeurIPS'22
>
> > ***(W1.4/Q3.4) The fitting procedure implies that there is a need to have an anchor distribution that matches (at least to some degree) the feature distribution. So how close does it need to be to a Gaussian distribution?***
>
> You are correct that we desire that “the anchored distribution matches (at least to some degree) the feature distribution.” As discussed above, this is because the anchoring distribution can be seen as a random variable which the model must learn to (partially) marginalize for convergence. By keeping the anchoring distribution closer to input features (even imperfectly), it is a bit *easier on the model to train, since the model does not have to handle vastly different input ranges* or drastically diverge from the training manifold. *We’ve added summary statistics of the fitted distribution to the appendix in Table 18.*

---

> ### Author Response · Authors · 2023-11-20
> **Response to Reviewer: b8ZS (part 3)**
>
> > ***(W1.5) For the graph anchoring, there is a random shuffling of the node features over the entire batch. There is no discussion of this design choice – it doesn’t seem obvious to me that this is the only thing one could choose to do (or the optimal).***
>
> Directly optimizing over the anchoring distribution/strategy is difficult because we cannot know a priori (i) what hypotheses different anchors will induce (*diversity*) and (ii) if the variability arising from the sampled hypotheses is reflective of the model’s uncertainty (*effective* diversity). So, we agree that shuffling is not the only solution, nor likely the most optimal. We use random shuffling *primarily as an empirical mechanism for enabling the sampling of many different hypotheses, while also remaining close enough to the training manifold in order to support convergence (as evidenced by ID accuracy).* We emphasize that difficulties of sampling “optimal” hypotheses for ensemble-style methods is not unique to G-ΔUQ and empirically does not seem to harm G-ΔUQ’s performance. Indeed, similar arguments can be made about DeepEns (e.g., is changing the random seed the optimal way of obtaining hypotheses?). *Moreover, our extensive experiments demonstrate that our proposed strategies outperform baselines across 11 datasets, 3 shifts and 3 tasks; we suspect efforts into more sophisticated anchoring strategies would only improve this performance.*
>
> **Alternative Strategies:** *We conduct additional experiments (see Table. 15)*  where we define a Gaussian distribution over the hidden representations at the anchoring layer, and then sample from this distribution to create anchors (similar to node feature anchoring). We find that this strategy generally maintains or improves performance over the vanilla model, and does in fact improve on calibration on some datasets/shifts relative to the shuffling formulation (see Table 15 for details). An interesting direction of future work permitted by this formulation is to optimize the parameters of the anchoring distribution given a signal from an appropriate auxiliary task or loss.
>
>
> > ***(W2) The technical methodological contribution is thus not particularly substantial. On the other hand, the experiments are thorough and provide a good balancing contribution.***
>
> Thanks for finding our experiments thorough and a balancing contribution. While we agree our proposed anchoring strategies may seem simple at first due to their straightforward implementations, they are in fact carefully motivated and address key gaps in the original stochastic centering formulation. Indeed, *our hidden layer anchoring experiments demonstrate that it is possible to trade off the “amount” of diversity (as induced by the number of stochastic layers) and the “effective” or functional diversity (as induced by sampling hypotheses that rely upon different high-level semantic features)*; this results in improved performance over input-space anchoring in many cases. Importantly, *our proposed strategies are also able to support pretrained models*, which is particularly valuable in the era of large models and was not considered by [1]. Furthermore, we note that our proposed strategies support the dual tasks of node-classification and graph classification.
>
> Lastly, as we discussed above, managing the effects of message passing when using input space anchoring is non-trivial due to compounding stochasticity. While our solution is simple, it is indeed effective (as demonstrated by our extensive experiments) and provides interesting insights for future work that seeks to optimize this distribution.
>
> > ***(W3). Some of the results are not presented in a particularly helpful manner and are not described or discussed in much detail.***
>
> Thanks for the comments. We’ve moved the generalization error prediction experiments to the appendix so that we can expand upon the presented results and *summarize the interesting insights below. We have updated the paper with this expanded discussion. Changes are shown in blue.*

---

> ### Author Response · Authors · 2023-11-20
> **Response to Reviewer: b8ZS (part 4)**
>
> - ***(W3) The table contains interesting elements such as the proposed method failing to improve (WebKB, CBAS – Concept).***
>
> Our goal in Table 1 (and in Table 10) is to compare accuracy and ECE of G-ΔUQ and non-G-ΔUQ (vanilla) models on 4 node classification benchmarks evaluated across concept and covariate shift. First, we note that across all our evaluations, without any posthoc calibration, **G-ΔUQ is superior to the vanilla model on nearly every benchmark for better or same accuracy (16/16 benchmarks) and better ECE (15/16).** However, due to the challenging nature of these shifts, achieving SOTA calibration performance often requires the use of post-hoc calibration methods – so we also evaluate how such methods can be elevated when combined with G-ΔUQ.
>
> + When combined with posthoc methods, performance generally improves across the board. For example, on **WebKB – across the 9 calibration methods considered, “G-ΔUQ + <calibration method>” improves or maintains the calibration performance of “no G-ΔUQ + <calibration method>” in 7/9 (concept) and 6/9 (covariate)**. On **CBAS**, calibration is improved or similar to the no-G-ΔUQ version on **5/9 (concept) and 9/9 (covariate) cases.** Note, this is achieved with little or no compromise on classification accuracy (often improving over “no G-ΔUQ” variant).
>
> + We also emphasize that, across all the 8 evaluation sets (4 datasets x 2 shift types) in Table 10, the best performance is almost always obtained with a **G-ΔUQ variant: (accuracy: 8/8) as well as best calibration (6/8) or second best (2/8).**
>
> The reviewer is correct to note that there is a discrepancy in the performance across calibration methods, and a more detailed study of how different assumptions made by these techniques interact with stochastic centering is required, but is out of scope of the current study. We will address these issues in our expanded discussion section.
>
> - ***(W3/ Q1) Why does the proposed method appear to work relatively poorly in terms of ECE when used with Dirichlet calibration in the node classification setting?***
>
> Table 12 contains many different axes, so while it appears that our proposed method performs worse with Dirichlet, this is not a shortcoming of our method. *Instead, it is a shortcoming of Dirichlet as a post hoc calibration strategy. Indeed, with the exception of (WebKB, covariate shift), applying Dirichlet non-trivially decreases the accuracy and calibration on 7 ⁄ 8 datasets with respect to both the vanilla and G-ΔUQ models (where no post-hoc strategy has been applied).* This clearly indicates that Dirichlet is not a robust post hoc calibration strategy under distribution shifts. One explanation for this behavior is that Dirichlet fits a distribution to the uncalibrated probabilities on the in-distribution validation data using a regularized MLP, and the resulting distribution does not align with OOD data. Indeed, the failure of Dirichlet in this setting highlights the importance of evaluating calibration and other uncertainty based tasks on the realistic setting of OOD data. Furthermore, it supports the development of methods like G-ΔUQ which produce intrinsically better uncertainty estimates, even when additional validation/calibration datasets are not available.
>
> - ***(W3) Expanded Discussion on Pretrained (PT) vs. End-to-End (E2) G-ΔUQ.***
>
> We’ve expanded our discussion of (A.12; Fig. 8) to go into detail about the benefits of PT G-ΔUQ across individual metrics (Accuracy, ECE) and specific datasets. We also added runtimes to (A.13, Table 17) to highlight that PT G-ΔUQ is able to improve performance, across datasets and shifts, at a significantly reduced complexity relative to training an model E2E.
>
> - ***(W3) Expanded Discussion on OOD Detection.***
>
> We expanded our discussion to include *how PT G-ΔUQ and E2E G-ΔUQ perform separately, and to also discuss how type of shift does not play a substantial role for a method’s performance on this task.* We end our discussion by noting that future work could use more sophisticated ood detection scores on top of G-ΔUQ for improved performance.
>
> - ***(W3) Expanded Discussion on Generalization Error Prediction.***
>
> We’ve expanded our discussion of (A.11, Table 16) to analyze the performance of individual methods in more detail and discuss the effect of types of shifts (covariate vs. concept) on task difficulty. For example, while *G-ΔUQ variants perform best on 7/8 datasets, (PT) G-ΔUQ in fact performs the best on 6/8 datasets, and G-ΔUQ performs second best on 6/8.* Given its strong performance on improving OOD calibration, these results suggest that PT G-ΔUQ is an effective, light-weight strategy. We also observe that on 3/4 datasets predicting generalization error is more difficult when faced with covariate shift rather than concept shift. While we are, to the best of our knowledge, the first to investigate GEP for GNNs, studying this shift-dependent behavior further is an important direction for future work.

---

> ### Author Response · Authors · 2023-11-20
> **Response to Reviewer: b8ZS (part 5)**
>
> >  ***(Q4) Figure 2 is challenging to understand. Insufficient detail is provided.***
>
> Thanks for bringing this up. We will clarify the text around the figure – this figure corresponds to an ablation study that illustrates the impact of “partial stochasticity” on accuracy and calibration – i.e., only a part of the network is stochastic while the rest is deterministic. G-ΔUQ is uniquely capable of inducing stochasticity in only a part of the network. *As such, L1, L2, and L3 correspond to performing anchoring (and thereby making the remainder of the network stochastic) at the first, second or third/READOUT layer. The bar in gray corresponds to the vanilla (no G-ΔUQ) model for comparison.*
> Since we are doing intermediate layer anchoring here, the best performance is highly dependent on the network architecture and the task. We see this reflected in the plots, where different layers are optimal across different networks and tasks. *However, a consistent trend is that across all datasets and architectures the best accuracy and ECE is always one of the G-ΔUQ variants.*
> Additionally, we note that we are the first to empirically study the impact of partial stochasticity on accuracy and calibration, and our results suggest that it can give big boosts in performance, and enable the use of pre-trained models (for e.g. L3).

---

> > ### Comment · Reviewer_b8ZS · 2023-11-22
> > **Acknowledgement of response**
> >
> > Thank you for the extremely thorough response to my review. The methodology has definitely been improved, with design choices now being considerably clearer. The presentation of the results has also improved substantially, and the additional discussion makes the results easier to understand. The response concerning the Dirichlet post-hoc calibration was helpful. Overall, the experiments and results still have more of a flavour of “our proposed method works better” rather than providing helpful insight into the often more interesting questions of why and how, but the results are certainly comprehensive, thorough, and demonstrate outperformance across a variety of settings. It is impressive experimental work. I do not have any remaining questions and do not need further clarification from the authors. I am currently reflecting on the revised paper, the response to my review, and the other reviews and responses. This will potentially lead to me increasing my rating of the paper. My hesitation is mainly a concern about the extent of the technical contribution. The methodology in the paper is brief (Sections 3.1 and 3.2 occupy less than 1.5 pages). While the idea is novel, and the paper does present elegant solutions to the challenges in translating stochastic anchoring to the graph domain, as well as introducing the intriguing approach of anchoring at hidden layers, the overall technical development is somewhat limited.

---

### Author Response · Authors · 2023-11-20
**Summary of Rebuttal Revision**

We thank the reviewers for their helpful comments. Below, we list the updates made to the paper to incorporate this feedback. ***Changes are shown in BLUE in the revised pdf.***

- **Introduction:** We have updated the introduction so calibration and uncertainty estimation are introduced sooner.

- **Notations:** We have updated Sec. 2 (Background on Anchoring) and Sec. 3 in response to reviewer feedback. We have clarified the anchoring formulation (Sec 3.1-3.2) with respect to the updated notation and included additional details.

- **Sec 3: “What are the Goals of Anchoring”:** We have added a conceptual discussion that helps contextualize our anchoring design choices.

- **Results Discussion (Sec 4,5.1,5.2,6.2):** We have expanded our original analysis to provide additional empirical insights.

- **New Pseudocode Figures:** We have added Figs 1, 2, and Fig 5 to the paper, which contain, respectively pseudocode for training/inference with anchored models, creating node feature anchors, and intuitively demonstrating the difference between vanilla and GDUQ models.

The following updates have been made to the appendix, including several new experiments.

- **Table 14/15:** We added additional results for performing intermediate anchoring for node classification tasks.
- **Table 16:** We added additional results for alternative anchoring strategies when performing graph classification tasks.
- **A.11 (Results on Generalization Error Prediction):** We have moved the generalization error prediction results to appendix and expanded our discussion accordingly.
- **A.12 (Additional Study On Pretrained G-∆UQ):** We have expanded our discussion of pretrained GDUQ performance on GOODDatasets.
- **A.13 (Runtime Table):** We’ve added a runtimes table to contextualize the benefits of pretrained GDUQ.
- **A.14 (Mean And Variance Of Node Feature Gaussians):** We include the mean and variance of the distributions used for node feature anchoring.
- **Table 19:** We include the number of parameters of the models used in the GOODDataset experiments.
- **A.15 (Expanded Discussion on Anchoring Design Choices):** We added some discussion explaining how our proposed node feature anchoring helps support fully stochastic GNNs.

---

### Meta-Review · Area_Chair_TuQt · 2023-12-09

**Metareview:**

The paper presents a method for improving uncertainty estimates in graph neural networks in distributional shift settings. It does so by adapting the principle of stochastic data centering to graph data through novel graph anchoring strategies and is able to support partially stochastic GNNs.  Experiments are done in node and graph classification tasks.

I thank the authors and the reviewers for the discussions. Overall, all reviewers agree that the paper presents an interesting novel idea and that the experiments are comprehensive. Some of the main issues highlighted were: (i) unjustified heuristics/design decisions such as the use of Gaussian for node feature anchoring; (ii) not may insights in experiments beyond “it works better” and clarity of the paper.

The authors have provided a comprehensive rebuttal to each of the reviewers’ concerns and, indeed, the length of methodological section should not be factor for assessing the quality of paper. Overall, I think the contribution of this paper is mainly conceptual (that required some methodological development) that that the empirical results are strong. Hence, I recommend acceptance.

**Justification For Why Not Higher Score:**

Perhaps not enough methodological contribution to be a spotlight but good conceptual innovation with strong results.

**Justification For Why Not Lower Score:**

Good conceptual innovation with strong result.

---

### Decision · Program_Chairs · 2024-01-16

Accept (poster)